# A spatial human thymus cell atlas mapped to a continuous tissue axis

Nadav Yayon[1,2,13,19], Veronika R. Kedlian[1,13,19], Lena Boehme[3,19], Chenqu Suo[1,4], Brianna T. Wachter[5], Rebecca T. Beuschel[6], Oren Amsalem[7], Krzysztof Polanski[1], Simon Koplev[1], Elizabeth Tuck[1], Emma Dann[1], Jolien Van Hulle[3], Shani Perera[1], Tom Putteman[3], Alexander V. Predeus[1], Monika Dabrowska[1], Laura Richardson[1], Catherine Tudor[1], Alexandra Y. Kreins[8,9], Justin Engelbert[10], Emily Stephenson[1,10], Vitalii Kleshchevnikov[1], Fabrizio De Rita[11], David Crossland[11], Marita Bosticardo[5], Francesca Pala[5], Elena Prigmore[1], Nana-Jane Chipampe[1], Martin Prete[1], Lijiang Fei[1], Ken To[1], Roger A. Barker[12,13], Xiaoling He[12,13], Filip Van Nieuwerburgh[14,15], Omer Ali Bayraktar[1], Minal Patel[1], E Graham Davies[8,9], Muzlifah A. Haniffa[1,10,16], Virginie Uhlmann[2], Luigi D. Notarangelo[5], Ronald N. Germain[6✉], Andrea J. Radtke[6✉], John C. Marioni[2,17✉], Tom Taghon[3,15✉] & Sarah A. Teichmann[1,13,18✉]

T cells develop from circulating precursor cells, which enter the thymus and migrate through specialized subcompartments that support their maturation and selection[1]. In humans, this process starts in early fetal development and is highly active until thymic involution in adolescence. To map the microanatomical underpinnings of this process in pre- and early postnatal stages, we established a quantitative morphological framework for the thymus—the Cortico-Medullary Axis—and used it to perform a spatially resolved analysis. Here, by applying this framework to a curated multimodal single-cell atlas, spatial transcriptomics and high-resolution multiplex imaging data, we demonstrate establishment of the lobular cytokine network, canonical thymocyte trajectories and thymic epithelial cell distributions by the beginning of the the second trimester of fetal development. We pinpoint tissue niches of thymic epithelial cell progenitors and distinct subtypes associated with Hassall's corpuscles and identify divergence in the timing of medullary entry between CD4 and CD8 T cell lineages. These findings provide a basis for a detailed understanding of T lymphocyte development and are complemented with a holistic toolkit for cross-platform imaging data analysis, annotation and OrganAxis construction (TissueTag), which can be applied to any tissue.

The thymus is a highly specialized organ dedicated to supporting T cell generation, which begins to develop in embryonic post-conception week (p.c.w.) 8 from the bilateral thymic primordia[1]. Precursors of thymic epithelial cells (TECs), vascular and mesenchymal cells establish the microenvironment for T cell development, while lymphoid precursors colonize the thymic lobes and begin their continuous differentiation into conventional and unconventional T cell types that are released into the periphery from p.c.w. 12–14 onwards[1–3].

The thymus is organized into macroscopic compartments, the medulla and cortex, which are associated with distinct stages of T cell development. Bone-marrow-derived precursors enter the thymus in the medulla or around the corticomedullary junction (CMJ) and migrate to the cortex, where the CD4−CD8− double-negative (DN) thymocytes commit to the T lineage and undergo a first wave of V(D)J recombination of their T cell receptors (TCRs). After a proliferative burst in the early CD4+CD8+ double-positive (DP) stage, a second wave of TCR rearrangement at the *TRA* locus is followed by positive selection of cells with a functional TCR through interactions with cortical TECs (cTECs). This prompts differentiation into the major conventional lineages of CD4+ and CD8+ single-positive (SP) thymocytes, which

[1]Cellular Genetics, Wellcome Sanger Institute, Cambridge, UK. [2]European Bioinformatics Institute, European Molecular Biology Laboratory, Cambridge, UK. [3]Department of Diagnostic Sciences, Ghent University, Ghent, Belgium. [4]Department of Paediatrics, Cambridge University Hospitals, Cambridge, UK. [5]Laboratory of Clinical Immunology and Microbiology, Division of Intramural Research, National Institute of Allergy and Infectious Diseases, National Institutes of Health (NIH), Bethesda, MD, USA. [6]Laboratory of Immune System Biology, Lymphocyte Biology Section and Center for Advanced Tissue Imaging, National Institute of Allergy and Infectious Diseases (NIH), Bethesda, MD, USA. [7]Division of Endocrinology, Diabetes and Metabolism, Harvard Medical School, Beth Israel Deaconess Medical Center, Boston, MA, USA. [8]Department of Immunology and Gene Therapy, Great Ormond Street Hospital for Children NHS Foundation Trust, London, UK. [9]Infection, Immunity and Inflammation Research & Teaching Department, UCL Great Ormond Street Institute of Child Health, London, UK. [10]Faculty of Medical Sciences, Biosciences Institute, Newcastle University, Newcastle upon Tyne, UK. [11]Department of Adult Congenital Heart Disease and Paediatric Cardiology/Cardiothoracic Surgery, Freeman Hospital, Newcastle upon Tyne, UK. [12]Department of Clinical Neurosciences, John van Geest Centre for Brain Repair, University of Cambridge, Cambridge, UK. [13]Cambridge Stem Cell Institute, University of Cambridge, Cambridge, UK. [14]Laboratory of Pharmaceutical Biotechnology, Ghent University, Ghent, Belgium. [15]Cancer Research Institute Ghent (CRIG), Ghent, Belgium. [16]Department of Dermatology and NIHR Newcastle Biomedical Research Centre, Newcastle Hospitals NHS Foundation Trust, Newcastle upon Tyne, UK. [17]University of Cambridge, Cancer Research UK, Cambridge, UK. [18]Department of Medicine, University of Cambridge, Cambridge, UK. [19]These authors contributed equally: Nadav Yayon, Veronika R. Kedlian, Lena Boehme. ✉e-mail: rgermain@niaid.nih.gov; andrea.radtke@nih.gov; marioni@ebi.ac.uk; tom.taghon@ugent.be; sat1003@cam.ac.uk

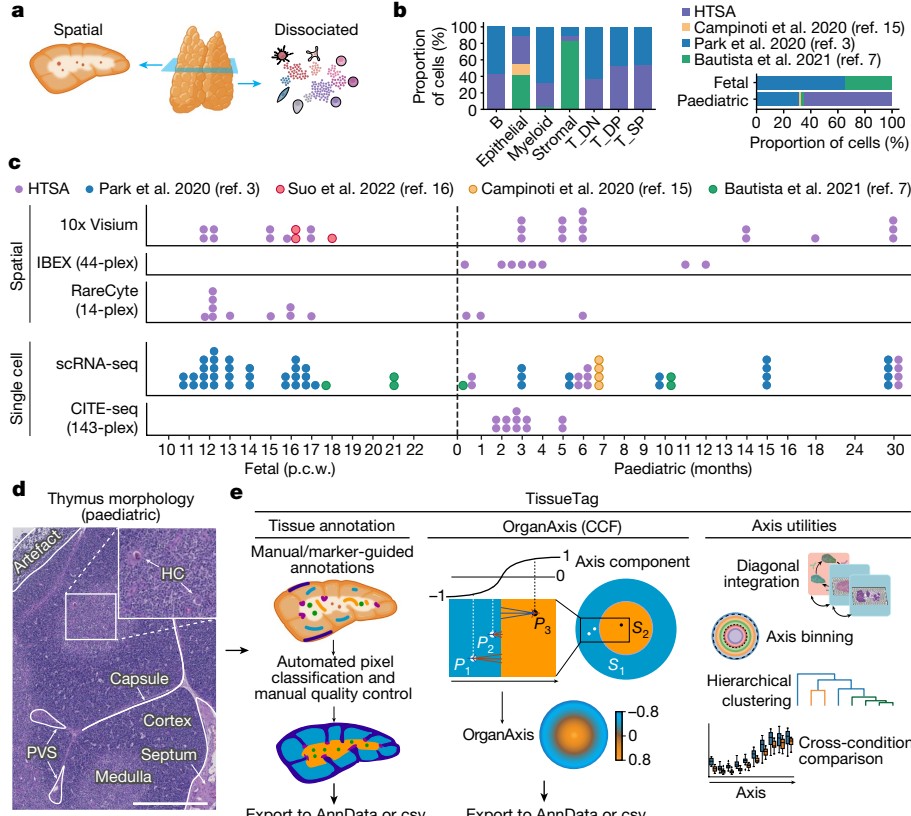

**Fig. 1 | Human thymus spatial atlas data composition and methodology.** **a**, Schematic of the combined use of spatial and dissociated datasets. **b**, The proportional contributions of different studies to main cell types and age groups, n = 29 donors. HTSA, human thymus spatial atlas. **c**, The composition of dissociated and spatial datasets containing both newly generated and previously published data spanning fetal and early paediatric human life. Each dot represents a sample and the stacked dots within a technology panel represent samples from the same donor. The dot colour indicates data source. Further information is provided in Supplementary Tables 1 and 2 and Extended Data Fig. 1a. **d**, Representative H&E image of a paediatric thymus (7 days old) showing the major anatomical compartments. Scale bar, 0.5 mm. **e**, Overview of the functionalities available in the TissueTag software. Details of histology annotations and derivation of the OrganAxis framework are provided in Supplementary Notes 1 and 2.

migrate into the thymic medulla. There they undergo a pruning process termed negative selection, whereby medullary TECs (mTECs) and other antigen-presenting cells, such as dendritic cells (DCs) and B cells, delete or convert autoreactive thymocytes into suppressive regulatory T ($T_{reg}$) cells to enforce tolerance to self[4]. Beyond the coarse division into cortex and medulla, the thymus has secondary morphological structures, such as medullary Hassall's corpuscles (HCs), and less clearly defined regions, such as the highly vascularized region at the CMJ, often referred to as perivascular space (PVS). Finally, the thymus is surrounded by a capsule, which exhibits extensions of connective tissue, called trabeculae or septa, that separate individual thymic lobules[5].

Recent efforts by the Human Cell Atlas Community, the Human BioMolecular Atlas Program (HuBMAP) and others have identified a high degree of diversity both in developing T cells and in resident haematopoietic and stromal lineages in the prenatal and postnatal human thymus[3,6–12]. Yet, questions remain about the fine-grained organization of these cell types within the thymic lobules and tissue niches that drive the maturation and migration of thymocytes. These points have been difficult to address using low-throughput spatial technologies, but emerging methods for spatial genomics and highly multiplexed RNA/protein imaging[13] now provide sufficient resolution to construct a comprehensive spatial and molecular atlas of the human thymus. Despite these advances, intersample comparisons, especially for human samples, have been impeded by the variability in tissue morphology and sampling approaches, which represents a

substantial hurdle for data integration and cross-study comparisons. Moreover, the lack of a common coordinate framework (CCF)[14] limits spatial annotations to discrete structures or morphological compartments, precluding the assessment of intraregion variance and intrinsic molecular gradients. To overcome these obstacles and obtain a more holistic portrait of this critical immunological organ throughout early human development, we describe a mathematical model for a continuous scale- and rotation-invariant morphological CCF for the human thymus, which we term the Cortico-Medullary Axis (CMA).

We used this CCF to integrate a comprehensive dataset consisting of two types of spatial omics data and, to our knowledge, the largest multimodal single-cell annotation reference of the human thymus to date. This enabled us to describe the organization of the thymus at a resolution beyond the typically annotated morphological compartments. We show that most cytokine/chemokine gradients and the canonical T lineage maturation trajectory are already established by the beginning of the second trimester, whereas TEC progenitor niches partially differ between prenatal and postnatal thymus. Furthermore, we find specific cell types and genes associated with HCs and show that thymocytes developing along the CD4 and CD8 lineages exhibit divergent timing of corticomedullary migration.

## Building a multimodal spatial thymus atlas

To robustly map and characterize human thymic cell types throughout development, we built a comprehensive spatial multimodal thymus

cell atlas by combining single-cell sequencing and spatial data from fetal (p.c.w. 11–21) and paediatric (neonate to 3 years) thymus samples (Fig. 1a). We first integrated three large publicly available datasets[3,7,15] (20 donors, 266,551 cells) with in-house-generated cellular indexing of transcriptomes and epitopes by sequencing (CITE-seq) data (5 donors, 146,352 cells) and additional stroma-enriched single-cell RNA-sequencing (scRNA-seq) data (4 donors, 69,748 cells) (Fig. 1b,c, Extended Data Fig. 1a,b and Supplementary Table 1). This permitted annotation of T cell developmental states and increased representation of epithelial, stromal and resident immune subsets in the thymus (Fig. 1b and Extended Data Fig. 1c–e). To provide a spatial context, we generated Visium spatial transcriptomics data (10x Genomics) from both fetal (9 samples from 5 donors) and paediatric (16 samples from 6 donors) tissue and integrated previously published fetal Visium data (3 samples from 2 donors)[16] (Fig. 1c and Supplementary Table 2). Moreover, we performed 44-plex IBEX cyclic protein imaging[17] of paediatric tissue samples (Supplementary Table 3). IBEX nucleus signals were segmented in three-dimensional (3D) space to extract channel mean levels per cell, yielding 1,101,631 nuclei from 8 samples/donors (Methods). Finally, we generated 14-plex RareCyte protein imaging data for validation purposes (Fig. 1c and Supplementary Table 4).

To integrate spatial modalities and biological samples, we developed the TissueTag computational framework and Python package (see Code availability). TissueTag uses high-resolution haematoxylin and eosin (H&E)-stained images for (semi)automatic tissue annotation to extract key histological features of tissues, such as the thymic cortex and medulla (Fig. 1d,e (left) and Supplementary Note 1). These annotations can subsequently be used to construct a CCF (OrganAxis) based on distance measurements between two or more histological landmarks (Fig. 1e (centre)). This unified OrganAxis enables various downstream analyses, such as integration of data from different modalities and cross-condition comparison of gene expression and protein abundance along the OrganAxis (Fig. 1e (right)).

## Constructing a CCF for the human thymus

To establish a CCF for the human thymus and accurately model the spatial variability within a thymic lobule, we produced resolution-matched tissue annotations for IBEX and 10x Visium data using TissueTag. For each image, we used a pixel classifier to distinguish between the cortex and medulla and unbiasedly call the border between them. We next corrected annotations where applicable and defined tissue edge (capsule), HC, PVS and vessel regions manually (Fig. 2a and Extended Data Fig. 2a,b; details on TissueTag-based annotation are provided in Supplementary Note 1). Finally, we used TissueTag to position any spatial coordinate $P$ (such as a Visium spot or the centroid of a segmented IBEX nucleus) through the OrganAxis framework using the distances to the cortex, medulla and edge (Fig. 2b, Extended Data Fig. 2c and Supplementary Note 2). First, a pair of these distances (for example, distance to cortex and distance to medulla) was used to compute a nonlinear distance metric $H$ (signed and normalized) to the boundary separating the two regions (such as the cortex and medulla). $H$ is a sigmoid-shaped function, engineered for increased sensitivity to boundary-proximal spatial changes. We next combined two $H$ functions between (1) the edge and cortex and (2) the cortex and medulla to better capture the relative position across the entire thymus lobule. This enabled us to construct a thymus-specific OrganAxis, which we termed the CMA (Fig. 2b,c and Extended Data Fig. 2d). As the OrganAxis is calculated on the basis of morphological landmarks alone, the approach is universally applicable to any 2D and 3D datasets acquired by distinct spatial technologies, enabling quantitative comparison across samples and modalities.

On integrated UMAP embeddings, discrete annotations broadly clustered together but did not capture heterogeneity across space (Fig. 2d), while the CMA captured local and global features consistently

(Fig. 2e). All other technical co-factors were independent (Extended Data Fig. 3a–c), highlighting the ability of the CMA to capture biological variance in integrated space. To quantify this, we performed a principal component analysis (PCA) of the full feature space for each spatial modality (genes for Visium, protein targets for IBEX). We then derived a score to assess the correlation of the principal component (PC) cumulative explained variance with the CMA or with the total gene or marker coverage per Visium spot or IBEX segmentation, respectively. We found that the CMA explained substantial and similar degrees of variance in both the fetal and paediatric Visium dataset (Fig. 2f), indicating consistency in the representation of transcriptomic diversity across both developmental stages despite clear morphological differences. In IBEX data, the association with the CMA was lower, probably owing to lack of information on cell composition variance that is missed by single-nucleus segmentations. Another factor is likely to be the composition of our antibody panel, which was deliberately chosen on the premise of cell identification and not for highly variable features across the CMA (Fig. 2f). Deriving the cumulative explained variance on individual samples from all datasets demonstrated no clear association with age or donor in both developmental time windows (Extended Data Fig. 3d).

Overall, the CMA captures spatial variance both at the level of the individual thymus section as well as across sections, independent of tissue size, donor age or batch (Fig. 2c–e and Extended Data Fig. 3a–d). A detailed description of the axis formulation, rationale, data simulation and implementation is provided in Supplementary Note 2 and the accompanying online tutorial (see Code availability).

## T lineage trajectories across ages

The thymus undergoes substantial architectural changes during early human development, underscoring the need for a sophisticated approach to locate equivalent regions in fetal and paediatric samples. To aid with data integration, we binned the CMA into anatomically informed levels (capsular and subcapsular region, cortical and medullary CMJ, and three cortical and medullary levels) using the same CMA cut-off values for all of the samples (Fig. 3a,b and Methods). Next, to compare cell type spatial localization patterns between fetal and paediatric thymus, we performed cell type deconvolution of Visium data using our newly integrated fetal and paediatric single-cell datasets and mapped the annotated cell types to the CMA (Fig. 3c,d). This enabled us to spatially position a broad range of stromal and immune cell types, including TECs, fibroblasts, vascular cells, B cells and myeloid cells (Supplementary Note 3). Moreover, we were able to map various differentiation stages of the T cell lineage and establish their localization in the fetal and paediatric thymus (Fig. 3e and Extended Data Fig. 4a; details on T lineage annotation and mapping are provided in Supplementary Note 4).

We found a highly conserved distribution of canonical αβ T lineage thymocytes along the CMA across development (Fig. 3e), which complements our previous observation that the cellular abundances of the main thymocyte differentiation stages are generally constant from p.c.w. 12 until 3 years of age[3]. Visium-derived CMA prediction pinpointed the early T cell progenitor (ETP) stage to the medulla in both fetal and paediatric tissue (Fig. 3e). This is consistent with previous reports of thymic entry of lymphoid progenitors in the medulla or CMJ[18,19], and also with demonstrations of an extensive vascular network in the thymic medulla[20]. The subsequent DN(early) stage showed substantial migratory activity as reflected by its wide distribution across thymic layers. While it was mostly detected in the subcapsular region in both the fetal and paediatric thymus, additional enrichment of this stage was observed in the paediatric medulla (Fig. 3e). We found that the subsequent maturation stages followed the conventional circular migration path[21], with (sub)capsular localization of DN thymocytes and a more distributed cortical pattern in the proliferating DP(P) and quiescent

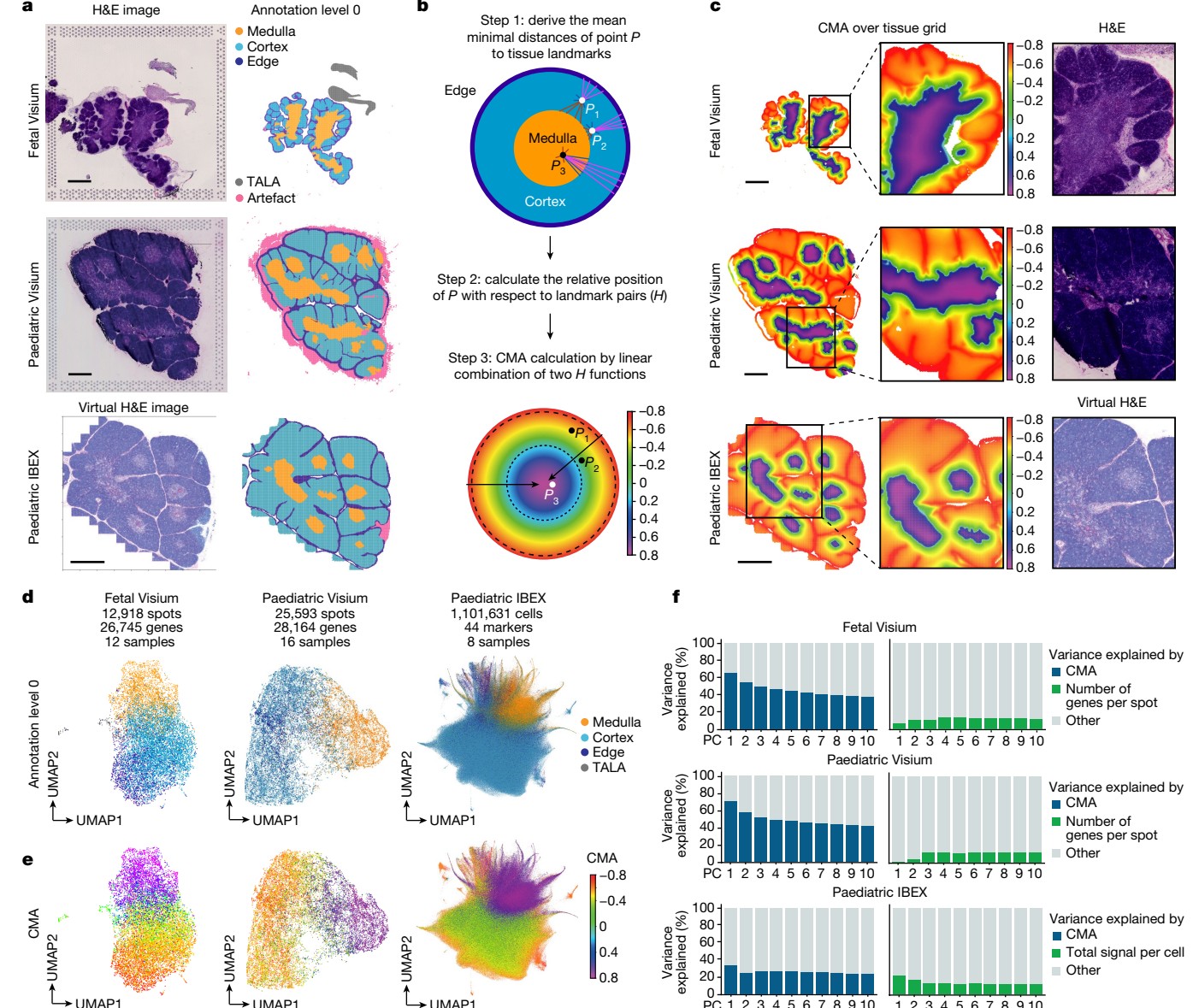

**Fig. 2 | Multimodal data integration using tissue landmarks and a continuous CMA CCF. a**, Representative H&E sections (left) of fetal (p.c.w. 15) and paediatric (3 months old) Visium data, and virtual H&E for paediatric IBEX (7 days old). Corresponding discrete annotations (right) curated with TissueTag for cortex, medulla, edge (capsule + septa), tissue artefacts and fetal thymus-associated lymphoid aggregates (TALA). **b**, Illustration of the CMA derivation. *P*, spot in space. Details of OrganAxis construction are provided in Supplementary Note 2. **c**, CMA mapping for sections shown in **a** and a magnified region with the corresponding (virtual) H&E image. Axis parameters: *r* = 15 μm,

*K* = 10. **d**, Batch-corrected UMAP embeddings of all Visium and IBEX samples in the study coloured by tissue annotations presented in **a**. Each dot corresponds to a Visium spot or IBEX cell. The total numbers of spots or cells, detected genes or markers and samples per dataset are indicated. **e**, UMAP embedding from **d** coloured according to CMA values. **f**, The contributions of CMA, technical factor (number of genes per spot or total signal per cell) and other sources of variability to the cumulative variance explained by first ten PCs in each spatial dataset: fetal Visium (12 samples), paediatric Visium (16 samples) and paediatric IBEX (8 samples). For **a** and **c**, scale bars, 1 mm.

DP(Q) thymocytes. αβT(entry) cells undergoing positive selection were still cortically located, whereas mature thymocytes committed to the CD4 or CD8 T lineage were enriched in the medulla (Fig. 3e).

To elucidate possible driving forces governing this migration process, we examined cytokine and chemokine expression in our Visium dataset. To this end, we calculated the relative distribution across the binned CMA for key thymic cytokines and chemokines detected in the Visium data and performed hierarchical clustering of the fetal distribution profiles to determine co-expressed cytokine groups (Fig. 3f). This revealed a large cluster of medullary and a smaller cluster of more cortical cytokines and chemokines. We further calculated the cosine similarity and performed an analysis of variance (ANOVA) to estimate the effect of age group and CMA level on cytokine

expression. Although most cytokines exhibited significant differences in expression levels between the fetal and paediatric thymus according to ANOVA, the vast majority did show a non-significant interaction in the expression pattern across CMA bins as well as a very high cosine similarity, indicating developmental conservation in cytokine expression patterns (Fig. 3f and Supplementary Table 5). Notably, the classical cortical (*CCL25* and *CXCL12*) and medullary cytokines (*CCL19* and *CCL21*) were among the genes with the highest cosine similarity (Fig. 3f and Extended Data Fig. 4b). Cytokines or chemokines with a low cosine similarity and/or a significant interaction effect between age and CMA layer, such as *IL34*, *IL33*, *IL1R1*, *CCL2* and *SPP1*, were mostly expressed in the fetal (sub)capsular region but shifted to the medulla in the paediatric thymus (Fig. 3f). scRNA-seq data suggested that this

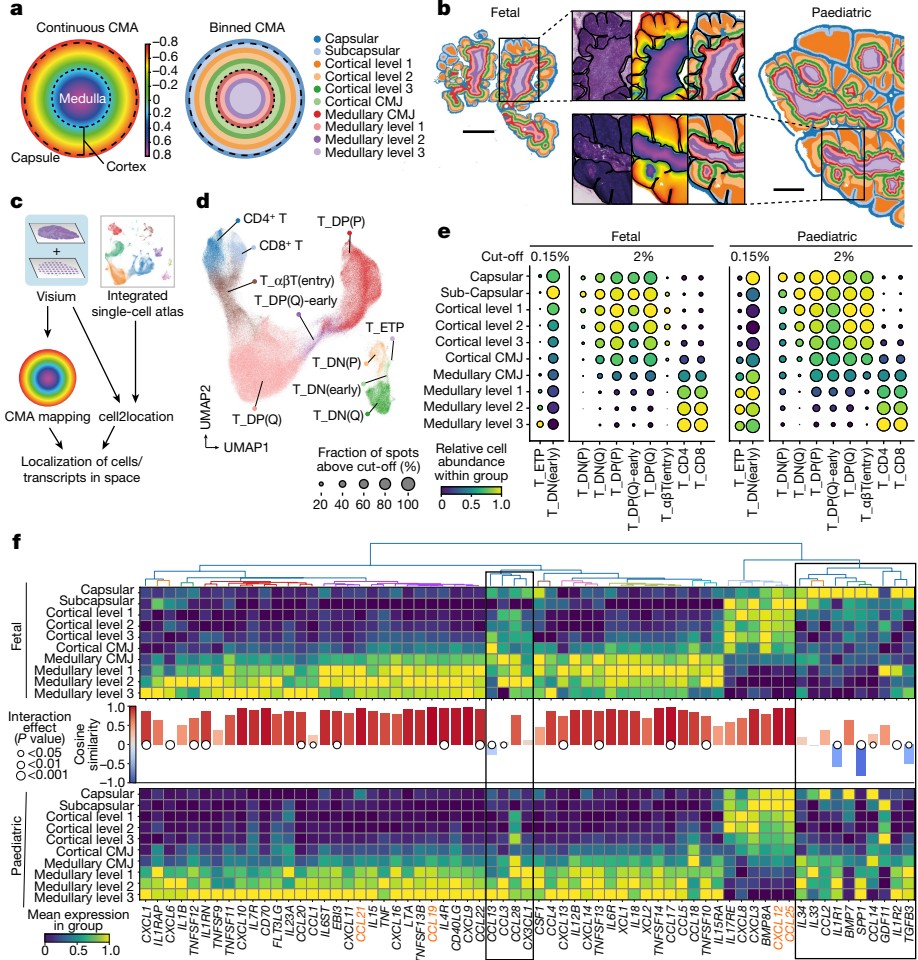

**Fig. 3 | CMA mapping of fetal and paediatric thymocytes reveals early establishment of T-lineage trajectories and cytokine landscape. a,** Schematic of the continuous CMA and the binned representation. **b,** Binned CMA space on fetal (p.c.w. 15) and paediatric (3 months) Visium sections with magnified regions showing H&E, continuous CMA and binned CMA (from left to right). Scale bars, 1 mm. **c,** Schematic of the analysis workflow: Visium spots are mapped to the CMA and deconvolved using the integrated fetal and paediatric scRNA-seq atlas. **d,** UMAP embedding of the main stages of developing αβ T lineage cells in fetal and paediatric scRNA-seq. Annotation of all T-lineage cells is provided in Supplementary Note 4. **e,** Binned CMA mapping of the main αβ T lineage differentiation stages for fetal and paediatric Visium. The cut-off indicates the minimum abundance threshold for inclusion of a Visium spot. The rare ETP and DN(early) stages were plotted with an adjusted cut-off to aid visualization. **f,** Spatial pattern of chemokine/cytokine transcripts across CMA bins derived from Visium data. Cytokines are clustered according to their spatial pattern in the fetal thymus. The bars indicate the cosine similarity between fetal and paediatric spatial gene expression patterns and the dots show an interaction effect between CMA bin and age-group. *P* values were calculated using two-way ANOVA with Bonferroni correction. Cytokines/chemokines that are critical for thymocyte migration are highlighted in orange. The boxes indicate clustered genes that diverge between ages.

was mainly due to differential expression between fetal and paediatric fibroblasts, endothelial cells, TECs and macrophages (Extended Data Fig. 4c).

Overall, we show that, by applying the CMA to spatial transcriptomics data and integrating this with a scRNA-seq reference, we are able to track the entire spatial trajectory of T lineage differentiation and demonstrate that it is already broadly established by the beginning of the second trimester of human development. We also show that, with few exceptions, the cytokine and chemokine expression patterns driving T cell migration are largely conserved between fetal and early postnatal thymus.

## Spatial mapping of TEC subtypes

The development of T cells is directly dependent on and supported by resident stromal and immune cells, which we were able to annotate and map using the single-cell atlas and Visium data as described above (Supplementary Note 3). TECs specifically have a key role in the selection of T cells with a diverse TCR repertoire (positive selection conveyed by cTECs) and tolerance to self-antigens (negative selection conducted by mTECs together with DCs, B cells and fibroblasts)[22].

To profile human fetal and paediatric TECs, we integrated several new and published scRNA-seq datasets enriched for CD205[+] cTECs and EPCAM[+] mTECs or depleted of CD45[+] lymphoid cells[7,15] (Extended Data Fig. 1a). This enabled us to identify three subsets of cTECs, multiple mTEC subtypes and putative bipotent TEC progenitor cells[23–25] termed mcTECs in earlier studies[3,26] (Fig. 4a,b). To infer the spatial localization of these groups of cells, we used Visium deconvolution (Fig. 4c) and further validated the findings using IBEX multiplex imaging with an antibody panel tailored towards TEC detection (Extended Data Fig. 5a). Segmented nuclei from IBEX data were integrated and annotated using the scRNA-seq data as a reference based on shared gene/protein space using a *k*-nearest neighbours (KNN) approach (ISS patcher; Methods and Extended Data Fig. 5b,c). We then inferred CMA values for the annotated TECs from the IBEX images (Fig. 4d,e).

Our scRNA-seq data revealed limited heterogeneity in the cTEC compartment, with three major subtypes, cTECI, II and III, mainly differing in the expression of MHCII genes, *DLL4*, *TP53AIP1* and *TBATA* (Fig. 4b).

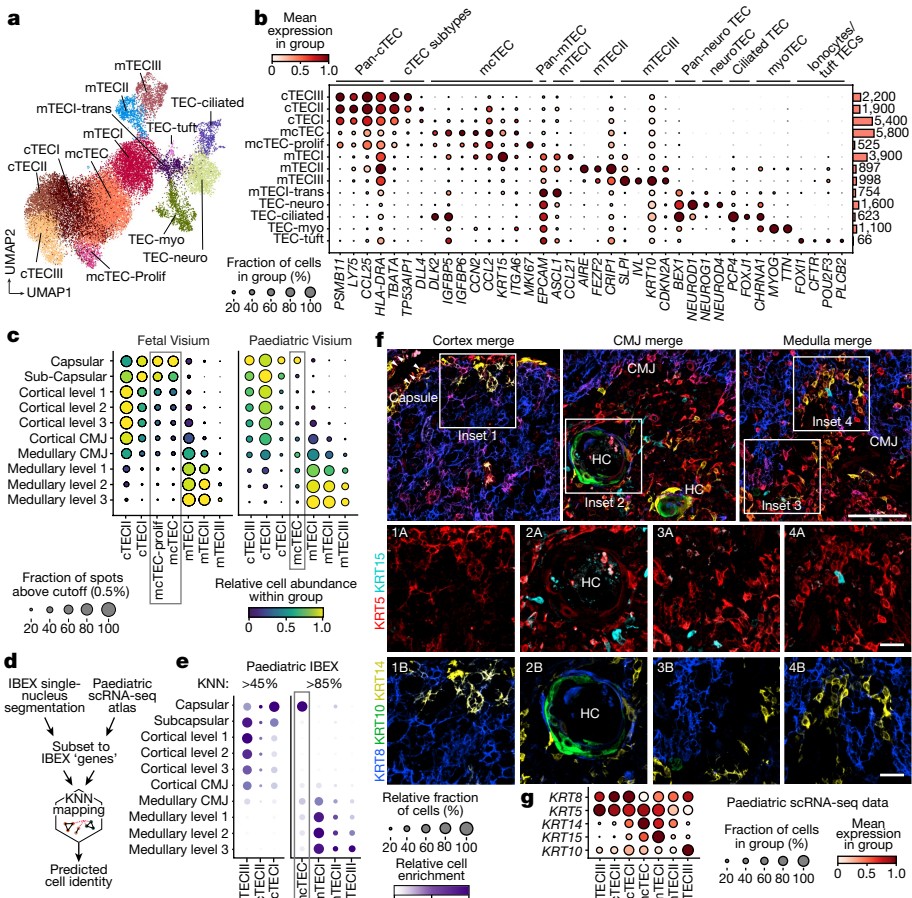

**Fig. 4 | CMA mapping pinpoints divergence in progenitor TEC localization across development. a**, UMAP embedding of integrated fetal and paediatric scRNA-seq data for TECs coloured by cell type. **b**, TEC marker gene expression according to scRNA-seq data. The bars indicate the total number of cells per cell type. **c**, The relative TEC distribution and enrichment in CMA bins based on Visium spot deconvolution. The boxes highlight mcTECs. Proliferating mcTECs were found only in fetal thymus and cTECIII was exclusively detected in paediatric data. The cut-off levels indicate the minimum cell abundance threshold for inclusion of a Visium spot. **d**, Schematic of the annotation workflow of segmented IBEX nuclei based on matching to the paediatric

scRNA-seq reference. **e**, The spatial distribution patterns of TEC subsets in IBEX data. The dot size represents the relative abundance and the colour depicts the local enrichments in the CMA bin. KNN indicates the cut-off for the percentage of KNNs that correspond to the eventually assigned majority cell type agreement (Methods). **f**, IBEX images from 7-day-old male thymus (IBEX_02) showing expression of five different keratins. Images are representative of eight donors/samples. Scale bars, 100 μm (top) and 25 μm (middle and bottom). **g**, The transcript levels of the corresponding keratins in the major paediatric TEC subtypes shown in **c**.

For both fetal and paediatric thymus, Visium mapping of cTECs predicted mostly capsular and subcapsular localization of cTECI, while cTECII showed a broader distribution across (sub)capsular regions and cortex (Fig. 4c). cTECIII was detected only in paediatric samples and was similarly distributed in (sub)capsular areas and across the cortex. Analysis of IBEX data confirmed this enrichment pattern for the paediatric thymus and cells expressing the cTEC marker KRT8 were clearly detected in the cortex (Fig. 4e,f).

In the mTEC compartment, we could distinguish immature mTECI (*CCL21*+MHCII^lo^*AIRE*^−), mature *AIRE*-expressing mTECII (*AIRE*+MHCII^hi) and a set of *AIRE*^− mature mTEC populations that resembled mimetic cells in mice[6]. These included mTECIII keratinocytes (*IVL*+), myoTEC (*MYOG*+), neuroTEC (*NEUROD1*+), ciliated TECs (*FOXJ1*+) and a TEC subset resembling ionocytes and tuft cells (*FOXI1*+) (Fig. 4a,b). mTECIII was marked by strong expression of *KRT10*, and IBEX imaging clearly placed KRT10+ cells at the HCs in accordance with the known association of mTECIII with these structures[27] (Fig. 4f,g). Both Visium and IBEX mapping of mTECI−III highlighted a gradient-like localization pattern with mTECI displaying a broad distribution across the medulla and mTECIII almost exclusively located in medullary levels 2 and 3 (Fig. 4c,e). This mTEC distribution was highly conserved between the

fetal and paediatric thymus (Fig. 4c). Similarly, a high degree of developmental conservation was observed for most mimetic TECs, which were predominantly detected in the deep medullary layers, with the exception of ciliated TECs, which were enriched at the fetal CMJ versus the paediatric medulla (Extended Data Fig. 6a,b).

mcTECs were defined based on intermediate levels of cTEC and mTEC markers and high expression of stem cell markers *KRT15* and *ITGA6*[15,24,28] (Fig. 4b). They were further characterized by the expression of *DLK2* (reported previously[3]), *IGFBP6*, *CCN2* and *CCL2*. Visium and IBEX mapping of mcTECs in paediatric thymus indicated their predominant location in the capsule. Moreover, a second population was detected around the medullary CMJ (Fig. 4c,e), in line with their junctional location in mouse[29,30]. Further refinement of the position of CMJ-associated mcTECs using histological annotations revealed that they were mostly localized in the PVS of the paediatric thymus (Extended Data Fig. 6c,d), which aligns with recent suggestions of the subcapsular regions and PVS as TEC progenitor niches[20]. Using the keratin expression profile of mcTECs in the scRNA-seq data (*KRT8*^−*KRT5*+*KRT14*+), we were also able to identify putative mcTEC niches in the paediatric capsule and CMJ in our IBEX data (Fig. 4f,g (insets 1 and 4)). Through integrated analysis of single-cell, Visium and IBEX data, we found that capsular

mcTECs displayed priming towards cTEC fate, while CMJ mcTECs were either more mTEC-primed or unprimed (Supplementary Note 5). Visium mapping predicted fetal mcTECs to be strongly enriched in (sub)capsular regions, but, in contrast to paediatric thymus, no enrichment was detected at the CMJ (Fig. 4c). We confirmed this using single-molecule fluorescence in situ hybridization (RNAscope), which demonstrated a clear capsular localization of cells expressing the mcTEC markers *DLK2* and *IGFBP6* in the fetal thymus (Extended Data Fig. 6e). Finally, multiplexed RareCyte protein imaging possibly indicated the presence of proliferating epithelial cells (CD45⁻PanCK⁺Ki-67⁺) in distinct subcapsular zones of the early fetal thymus (p.c.w. 12), indicating the location of the putative proliferating mcTEC niche. The paediatric capsule contained no such cells, but some PVS regions in CMJ and medulla displayed epithelial cells (CD45⁻PanCK⁺) representing putative mcTEC niches (Extended Data Fig. 6f,g).

Overall, our data reveal the tissue niche of a putative TEC progenitor population in capsular regions of the fetal thymus and suggest a partial shift in its localization towards junctional regions and PVS in the paediatric thymus.

## mTEC association with HCs

HCs are composed of concentric layers of cornified epithelium produced by mTECIII keratinocytes[27]. These medullary structures have been implicated in negative selection[31] and autoimmunity[32,33] and reports have claimed their association with cell types such as T$_{reg}$ cells, B cells, plasmacytoid DCs (pDCs) and neutrophils[34,35].

To investigate this further, we annotated HCs in Visium sections using TissueTag (Extended Data Fig. 2a,b). We then calculated the distance from each spot to the edge of the nearest HC and subset the data using a maximum distance cut-off of 350 μm (Fig. 5a and Extended Data Fig. 7a). In contrast to the CMA, which is nonlinearly anchored by three reference landmarks (capsule, cortex, medulla) and describes relative localization, the HC distance is linearly dependent on a single landmark and is therefore best suited for studying proximal associations (Fig. 5b).

Using this measure, we found that mTECIII, mTECII and specialized TECs were closely associated with HCs, followed by certain subsets of B cells, DCs and T$_{reg}$ cells (Fig. 5c). Importantly, these cells had also shown predominant enrichment in the deep medullary level 3 (Fig. 4c and Supplementary Note 3). To better resolve whether specific types of cells were more strongly associated with medullary depth or with HCs, we compared the mean-weighted CMA position against the mean-weighted HC distance for all medullary cell types (Methods). Similar to our earlier observations of the main mTEC subtypes, mTECI showed the most superficial localization (Figs. 4c and 5d), whereas mTECII and mTECIII were found deeper along the CMA with mTECIII notably closer to HCs than mTECII (Fig. 5d). Among the specialized TECs, TEC-ciliated and mTECI-trans were the closest to HCs, with a mean-wighted distance located within a 110 μm radius, while TEC-neuro and TEC-myo were positioned further away. Importantly, several haematopoietic cell types (activated DCs (aDCs), B cells and T$_{reg}$ cells) were found at a similar medullary depth as the TEC subtypes but showed less association with HCs (Fig. 5d).

To investigate whether the expression of specific genes by medullary cells types exhibits a distinct organization around HCs independently of deconvolved data, we determined genes with medullary expression ($n$ = 867) that were uniquely detected in a single cell type in our paediatric scRNA-seq dataset (specialization genes (SGs); $n$ = 184) (Fig. 5e and Methods). The highest number of SGs originated from mTECIII (93 SGs), followed by mTECII (58 SGs) and myo-TECs (18 SGs), with 20 SGs being associated with all remaining medullary cell types. Weighted spatial mapping of SGs to the CMA and HC distance revealed that mTECII- and mTECIII-associated SGs were the closest to HCs and deep in the medulla (Fig. 5e,f), in contrast to myo-TEC SGs, which were further away. Among mTECIII SGs closest to HCs, we noted two different groups of genes,

one group associated with mucosal epithelia (including *CXCL17*, *PSCA*, *MUC4*), and the other a set of kallikrein-related peptidases (*KLK6*, *KLK7*, *KLK8*, *KLK10*) frequently found in keratinocytes.

To unravel the possible association of these SGs with mTECIII subsets, we performed trajectory analysis on the combined mTECII and mTECIII populations, revealing three distinct branches ending in mature mTECII and two mTECIII subsets (Fig. 5g,h). One mature mTECIII population, which we termed mTECIII-muc, was characterized by the expression of the previously identified mucosal SGs (Fig. 5i,j and Extended Data Fig. 7b,c), which are known to be expressed by the mucosal epithelia of the gut, lungs, stomach and kidneys[36]. The second branch population, termed mTECIII-skin, expressed classical keratinocyte genes, such as *KRT10* and *KRTDAP*, and a number of kallikrein-related peptidases, which are normally involved in skin desquamation[37] and marked the most differentiated state of that branch (Fig. 5i).

In summary, we show that cells in the thymic medulla can be distinguished on the basis of their localization along the CMA (medullary depth) and their proximity to HCs. This demonstrated that the mTECIII population is strongly associated with HCs and contains additional specialized cell subtypes expressing mucosal- and skin-related genes. By contrast, other specialized TEC subtypes and haematopoietic cell types were not universally associated with HCs and showed a stronger association with medullary depth.

## Mapping thymocytes at high resolution

Traditionally, studies of thymic T cell development have relied on surface markers for flow cytometric or imaging-based analysis, but these findings are proving to be difficult to relate to scRNA-seq data due to different abundances and delays between transcription and cell surface expression of marker genes. To overcome this issue, we performed CITE-seq using a 143-plex customized antibody panel against a broad range of cell surface markers commonly used in thymocyte research (Supplementary Table 6) in combination with single-cell TCR sequencing. Using this approach, we were able to annotate T cell developmental stages at a much higher resolution than when using scRNA-seq alone (Extended Data Fig. 8a and Supplementary Note 6). This substantially increased the number of identified discrete differentiation stages compared to the small number of other studies that have recently applied CITE-seq to the human thymus[12,38].

We then mapped the annotated thymocyte subsets to paediatric Visium data to determine their distribution along the CMA (details on CITE-seq annotation and mapping are provided in Supplementary Note 6). The distinction between immature, semi-mature and mature stages for CD4 and CD8 lineage thymocytes based on surface marker profiles (Extended Data Fig. 8b and Supplementary Note 6) revealed a notable difference in the localization of immature CD8 versus immature CD4 lineage thymocytes (Fig. 6a), which suggests divergence in their corticomedullary transition after positive selection and lineage bifurcation. To investigate this in detail, we performed trajectory analysis on the CITE-seq data to predict the differentiation pseudotime of CD4 and CD8 lineage cells from initiation of positive selection to full maturity (Fig. 6b,c). Moreover, to obtain more continuous spatial mapping of the cells, we subjected the CITE-seq data to high-resolution Leiden clustering before Visium mapping. By transferring the mean CMA values of each cluster back onto the CITE-seq data (Methods), we were able to better locate cells in both space and developmental time, independent of discrete annotations, and could therefore explore the relationship between differentiation and migration.

We found that positively selected thymocytes were initially still located in the cortex and only in the subsequent CD4^hiCD8^lo stage a minority of cells had transitioned to the thymic medulla (Fig. 6d). CD4SP immature thymocytes were enriched in the medulla, as were CD4⁺ semi-mature and CD4⁺ mature cells, indicating that, for this lineage, the corticomedullary migration occurs soon after lineage

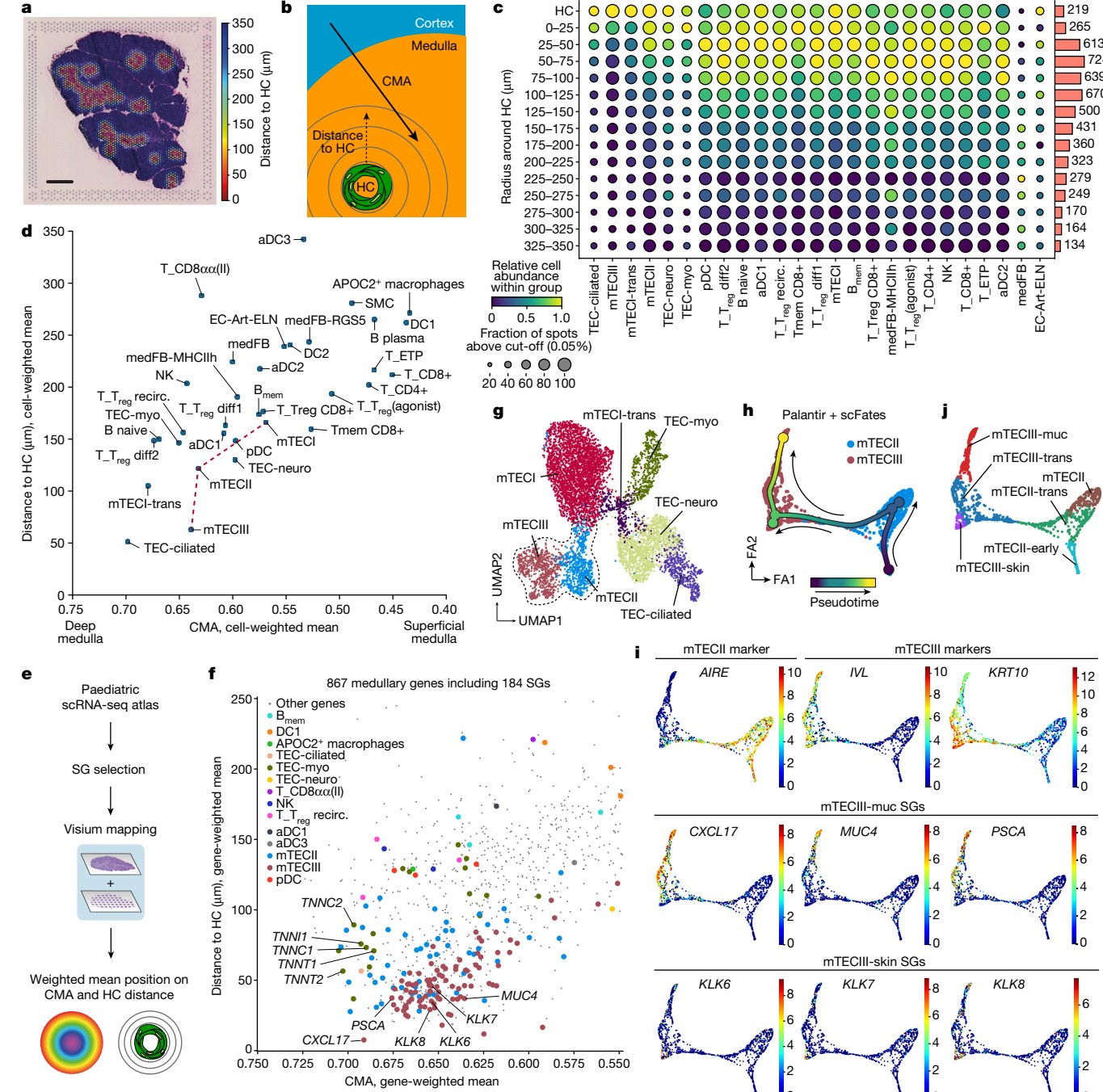

**Fig. 5 | Specialized mTECs are organized around HCs in the paediatric medulla. a**, Representative paediatric (3 months old) Visium H&E image overlaid with minimal distance to HC (maximum cut-off, 350 μm). Scale bar, 1 mm. **b**, Illustration of the distance to the HC as opposed to the CMA, which is parallel to medullary depth. **c**, The distribution of medullary cells around HCs based on deconvolved paediatric Visium data. HC distance was split into 25 μm bins. The bars show the absolute spot numbers. Cells are sorted by the mean distance to, 25 cell types closest to HC are shown. **d**, The weighted mean position of medullary cells along the two axes based on deconvolved paediatric Visium data. The red line connects major mTEC subtypes. **e**, The workflow for identifying SGs and their spatial mapping using paediatric Visium data. **f**, The

weighted mean position of all 867 medullary genes along the CMA and HC distance axes. SGs are represented by large dots and coloured according to the cell type in which they are uniquely expressed. **g**, UMAP embedding for mTECs; the mTECII/mTECIII lineage is highlighted. **h**, Trajectory analysis of the mTECII/III lineage. Colour indicates pseudotime, the arrows were manually added to illustrate direction. **i**, Trajectory embeddings showing the expression of canonical mTECII and mTECIII markers, mucosal and skin SGs. Colour scales represent expression levels. **j**, Annotation of mTECII/III states, highlighting the mTECIII-muc and mTECIII-skin subtypes, guided by mucosal and skin SGs. B_mem, memory B cells; medFB, medullary fibroblasts; NK, natural killer cells; EC-Art-ELN, elastin-expressing arterial endothelial cells.

bifurcation (Fig. 6d (left)). By contrast, the majority of immature CD8[+] cells was predicted to remain in the cortex, while they underwent co-receptor reversal from the CD4[hi]CD8[lo] to CD4[−]CD8[+] phenotype (Fig. 6d (right)) and Extended Data Fig. 8b). Similarly, semi-mature

CD8[+] cells were still found in the cortex in substantial numbers and only mature CD8[+] cells were finally mainly detected in the medulla (Fig. 6d (right)). To further validate this relative delay in the corticomedullary transition of CD8 lineage cells using IBEX, we applied our KNN

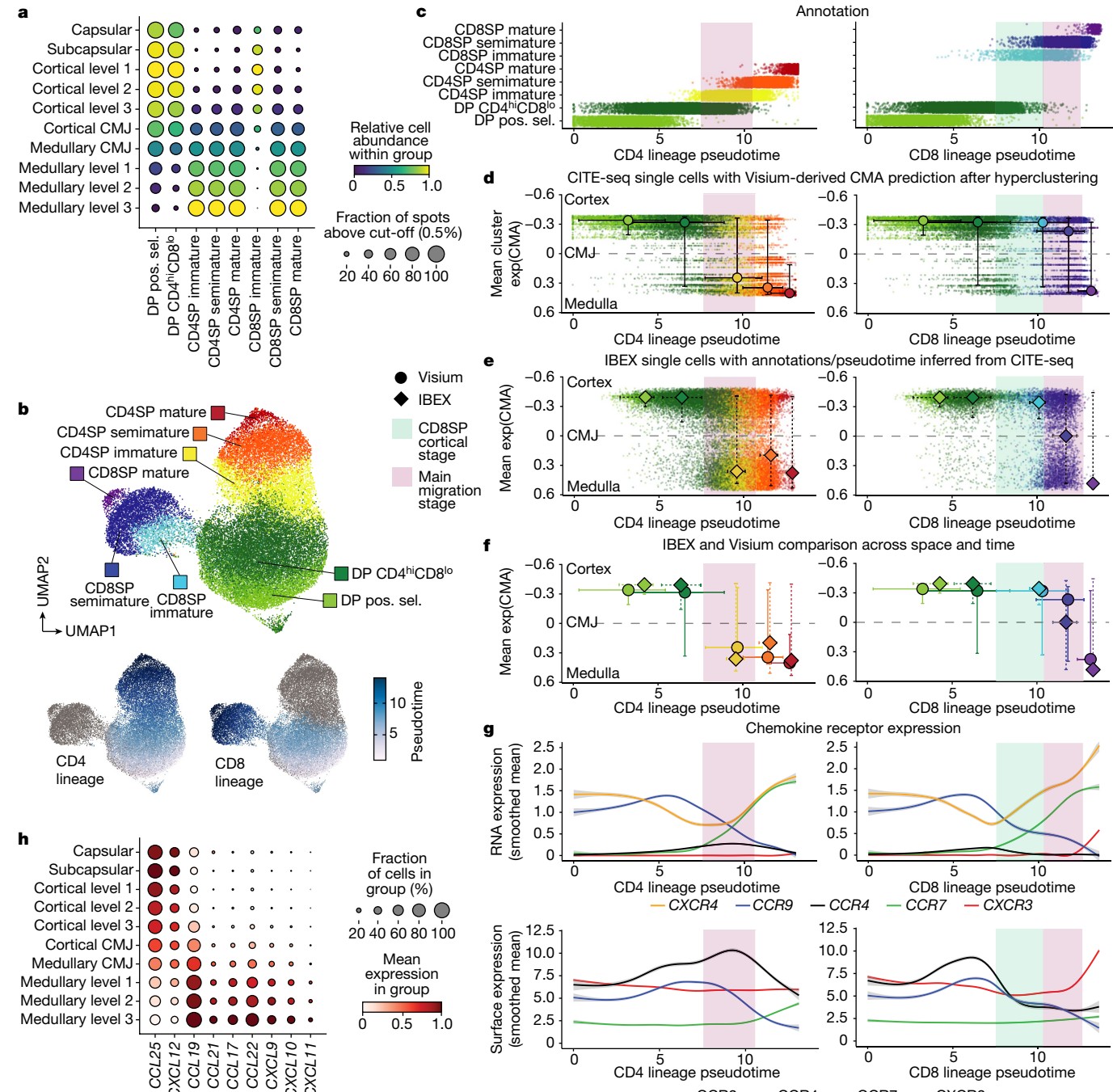

**Fig. 6 | High-resolution annotation and CMA mapping of single-positive thymocytes reveals spatiotemporal differences in their corticomedullary migration. a**, Cell abundances of αβ T lineage thymocytes after positive selection along the binned CMA based on deconvolved paediatric Visium data. The minimum-abundance cut-off for the inclusion of a Visium spot was 0.5%. **b**, Weighted nearest-neighbour (WNN) UMAP representation of paediatric CITE-seq data for conventional αβ T lineage cells from positive selection to full maturity (top). Developmental pseudotimes for the CD4 and CD8 lineage (bottom). **c**, CITE-seq cells ordered along CD4 (left) and CD8 (right) lineage pseudotime with the colour indicating discrete annotations. **d–f**, Spatial

mapping of pseudotime-ordered CD4 (left) and CD8 (right) lineage cells to the CMA. Median values and the 0.05–0.95 quantiles of both the CMA and the pseudotime value for each annotated substage are shown. **d**, Cell localization inferred from Visium deconvolution using hyperclustered CITE-seq data. **e**, The position of IBEX cells after segmentation and CITE-seq-derived annotation. **f**, Direct comparison of the Visium (circle) and IBEX CMA mapping (diamond) shown in **d** and **e**. **g**, RNA and surface protein levels of relevant chemokine receptors along the pseudotimes for CD4 (left) and CD8 lineage (right) cells. Lines represent the smoothed mean ± s.e.m. **h**, The spatial distribution of chemokine transcripts along the CMA in paediatric Visium data.

matching approach (ISS patcher) to annotate IBEX T cells and infer developmental pseudotimes using the CITE-seq data as a reference by leveraging the 19 protein targets covered by both antibody panels (Methods). This resulted in a predicted spatial distribution of CD4 and CD8 lineage cells that was highly similar to that previously obtained

from Visium data (Fig. 6e,f), confirming differences in the migratory kinetics between the two lineages.

To examine which chemokines could be involved in this lineage-specific migration pattern, we assessed chemokine receptor kinetics at the RNA and protein level. In both lineages, expression of *CXCR4*

and *CCR9* (also known as *CD199*) dropped in the positively selected and CD4$^{hi}$CD8$^{lo}$ thymocytes, consistent with previous reports on their role in cortical retention through the cortically expressed chemokines CXCL12 and CCL25[39,40] (Fig. 6g,h). Expression of CCR4 (also known as CD194), which is known to mediate corticomedullary migration in response to CCL17 and CCL22[39,41], increased throughout the CD4$^{hi}$CD8$^{lo}$ stage, but while cells of the CD4 lineage remained CCR4$^{high}$ until the end of their migration window, immature CD8 lineage thymocytes exhibited a swift reduction in receptor RNA and protein levels (Fig. 6g), suggesting diverging CCR4 expression patterns as a possible driving force behind the earlier medullary migration of CD4$^+$ thymocytes. *CCR7* (also known as CD197), which mediates responses to CCL19 and CCL21 and is essential for medullary homing and negative selection[39], was upregulated in immature cells of both lineages. These observations align with a recent report on mouse thymocytes, which describes the staggered upregulation of CCR4 and CCR7 and notes reduced CCR7-mediated chemotaxis of immature SP thymocytes[41], which may explain our observed lack of migration of *CCR7*$^+$ immature CD8SP thymocytes. Finally, CXCR3 (also known as CD183) was not detected in CD4 lineage cells but was upregulated in semi-mature CD8SP thymocytes, possibly indicating a role in the late stages of medullary entry for this lineage (Fig. 6g). Notably, we found that CXCR3 ligands (*CXCL9*, *CXCL10*, *CXCL11*) were expressed in a broad range of haematopoietic and stromal cells, whereas the CCR4 ligands (*CCL17*, *CCL22*) were predominantly detected in aDC1–3 cells, B cells and at a low level in mTECII (Extended Data Fig. 8c). This suggests that CD4 and CD8 lineage cells may be recruited to the medulla by different cell types.

In summary, by using two highly different spatial technologies in conjunction with a multimodal single-cell reference atlas, we reproducibly detect a substantial difference in the timing of corticomedullary migration for CD4 versus CD8 lineage thymocytes that is associated with differing expression profiles for the chemokine receptors CCR4 and CXCR3.

## Discussion

Next-generation cell atlases have the potential to transform our understanding of the human body at the cellular and tissue level. However, harmonization and integration of single-cell and spatial datasets remains a major hurdle for multigroup consortia. Here we present OrganAxis, a computational approach for converting discrete tissue annotations into a spatial topological model that can serve as a CCF for the diagonal integration of spatial data from unpaired samples. Owing to its derivation from well-defined histological tissue landmarks, it supports comparisons between datasets obtained from diverse spatial methods. By adapting the OrganAxis framework to the thymus, we established the CMA, which allowed us to obtain a holistic representation of the human thymus throughout early development. This enabled us to compare cytokine gradients and thymocyte trajectories in early fetal and postnatal life, identify age-specific TEC progenitor niches and explore the migration kinetics of conventional αβ lineage thymocytes (Supplementary Discussion).

Overall, the breadth and depth of our multimodal thymus cell atlas, together with the CMA framework, might provide a strategy for charting changes during thymic ageing and involution. The construction of a spatial human thymus atlas may aid tissue engineering efforts and offer insights into thymic pathologies, for example, myasthenia gravis, Down's syndrome, DiGeorge syndrome and others, and in vitro tissue engineering efforts. Moreover, our ability to integrate both de novo generated and published datasets through the CCF empowers further refinement of our understanding of the thymus through the integration of additional datasets from diverse human populations and acquired using newly developed single-cell and spatial approaches. Beyond the thymus, the TissueTag and OrganAxis framework can be adapted to

any tissue with consistent landmarks, enabling synthesis of knowledge about organs and tissues across different institutions and using a broad variety of spatial technologies.

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

## Methods

### Data generation by institute

Metadata about scRNA-seq and CITE-seq samples, including information on source study, cell enrichment and donor age, are provided in Supplementary Table 1. Information about spatial data, including Visium, IBEX, RareCyte and RNAscope, is provided in Supplementary Table 2. In brief, all CITE-seq data were generated at Ghent University and mapped to the human genome (GRCh38) at the Wellcome Sanger Institute (WSI). All other original scRNA-seq data and 10x Visium data were generated and mapped to the human genome (GRCh38) at WSI. All IBEX imaging was performed at the National Institute of Allergy and Infectious Diseases (NIAID), NIH. All non-IBEX imaging datasets (RareCyte, RNAscope, Visium H&E) were generated at WSI. No fetal work was performed at the NIH and at Ghent University.

Human fetal and paediatric data from several previous studies[3,7,15,16] were included and reanalysed from raw fastq files. Details on sample processing, ethics and funding are available in the respective publications; details on the origin of each sample are provided in Fig. 1c and Supplementary Tables 1 and 2.

For samples processed at WSI, paediatric samples were obtained from cardiac corrective surgeries and provided by Newcastle University collected under REC approved study 18/EM/0314 and Great Ormond Street Hospital under REC approved study 07/Q0508/43. Human embryonic and fetal material was provided by the joint MRC & Wellcome Trust (grant MR/006237/1) Human Developmental Biology Resource (http://www.hdbr.org) with written consent and approval from the Newcastle and North Tyneside NHS Health Authority Joint Ethics Committee (08/H0906/21+5). Samples processed at Ghent University were obtained according to and used with the approval of the Medical Ethical Commission of Ghent University Hospital, Belgium (EC/2019-0826) through the haematopoietic cell biobank (EC-Bio/1-2018). Samples processed at the NIH were obtained under NIAID MTA 2016-250 from the pathology department of the Children's National Medical Center in Washington, DC, following cardiothoracic surgery from children with congenital heart disease. Use of these thymus samples for this study was determined to be exempt from review by the NIH Institutional Review Board in accordance with the guidelines issued by the Office of Human Research Protections. Informed consent was obtained from all donors or their legal guardians.

### Sample processing and library preparation for 10x scRNA-seq at WSI

Surgically removed paediatric thymi were directly moved to HypoThermosol (Sigma-Aldrich, H4416-100ML), shipped by courier with ice packs, and processed within 24 h from time of surgery. For scRNA-seq experiments, we performed single-cell dissociation as described in the protocol available online (https://doi.org/10.17504/protocols.io.bx8sprwe). In brief, thymic tissue was finely minced and cell dissociation was performed using a mixture of liberase TH (Roche, 05401135001) and DNase I (Roche, 4716728001) for ~30–60 min in two rounds. Digested tissue was filtered through a 70 μm strainer and digestion was stopped with 2% FBS in RPMI medium. Red blood cell lysis was then performed on the cell pellet using RBC lysis buffer (eBioscience, 00-4333-57), after which cells were washed and counted. Magnetic and/or FACS sorting were performed to enrich stromal populations. Magnetic sorting to enrich for EPCAM⁺ or to deplete CD45⁺ or CD3⁺ cells was performed for U09, U48 and Z11 samples (Supplementary Table 1) using a magnetic sorting kit from Miltenyi Biotec including LS Columns (130-042-401) and the following bead-tagged antibodies: human CD45 MicroBeads (130-045-801), human CD326 (EPCAM) MicroBeads (130-061-101), human CD3 MicroBeads (130-050-101). Z11 and U40 samples were FACS-sorted to enrich for CD45⁻ stromal cells (Z11) or to obtain both CD45⁻ cells and total TECs (U40) (Supplementary Table 1). To perform FACS sorting, cells were resuspended in FACS buffer (0.5% FBS and 2 mM EDTA in PBS),

blocked with TruStainFcX (BioLegend, 422302) for 10 min and were stained with antibodies against EPCAM (anti-CD326 PE, 9C4, BioLegend, 324206), CD45 (anti-CD45 BV785, HI30, BioLegend, 304048), CD205 (anti-CD205 (DEC-205) APC, BioLegend, 342207) and CD3 (anti-CD3 FITC, OKT3, BioLegend, 317306) and DAPI for 30 min. All antibodies were diluted 1:50 for staining. After staining, cells were washed and sorted using Sony SH800 or Sony MA900 sorters with a 130 μm nozzle. Samples were first gated to remove debris and dead cells, but no singlet gate was applied to ensure large TECs would not be excluded. For samples stained with anti-CD45, total CD45⁻ cells were sorted to enrich for stroma. For EPCAM/CD205-stained cells, EPCAM⁺CD205⁻ and EPCAM⁻CD205⁺ cells were sorted to obtain the total TEC fraction including cortical and medullary epithelial cells but exclude autofluorescent cells (Supplementary Fig. 19a). FCS Express v.7.18.0025 was used for data analysis. After sorting/enrichment, cells were resuspended in collection buffer to the recommended concentration (10⁶ cells per ml) and loaded to capture around 8,000–10,000 cells on the 10x Genomics chromium controller to generate an emulsion of cells in droplets using Chromium Next GEM Single Cell 5′ Kit v2 (1000263). GEX and TCR-seq libraries were further prepared using the Library Construction Kit (1000190) and Chromium Single Cell Human TCR Amplification Kit (1000252) according to the manufacturer's instructions. Sequencing was performed on the NovaSeq 6000 sequencer (Illumina). Additional details are provided in Supplementary Table 1.

### Curation of published datasets

Data from several previous studies[3,7,15,16] are included in this Article. For public datasets deposited on ArrayExpress, paired-end fastq files were downloaded from ENA, and the .sdhf file was used to determine the type of experiment (3′/5′ and the version of 10x Genomics kit). For public datasets deposited on GEO, if the data were deposited as a Cell Ranger .bam file, URLs for the bam files were obtained using srapath v.2.11.0. The .bam files were downloaded and converted to .fastq files using 10x bamtofastq v.1.3.2. If GEO data were deposited as paired-end fastq files, sra files were located using the search utility from NCBI entrez-direct v.15.6, downloaded and converted to fastq files using fastq-dump v.2.11.0. Sample metadata were curated from the abstracts deposited on GEO. Finally, published datasets that had been generated at WSI were downloaded from iRODs v.4.2.7 in the form of cram files and converted to fastq files using samtools v.1.12. using the command 'samtools collate -O -u -@16 $CRAM $TAG.tmp | samtools fastq -N -F 0×900 -@16 −1 $TAG.R1.fastq.gz −2 $TAG.R2.fastq.gz -'. Sample metadata were obtained using the imeta command from iRODs.

### Processing of published and newly generated scRNA-seq and TCR-seq datasets at WSI

After fastq file generation, 10x Genomics scRNA-seq experiments were processed using the STARsolo pipeline detailed on GitHub (https://github.com/cellgeni/STARsolo). A STAR human genome reference matching Cell Ranger GRCh38-2020-A was prepared as per instructions from 10x Genomics. Using STAR v.2.7.9a and the previously collected data about sample type (3′/5′, 10x Genomics kit version), we applied the STARsolo command to specify UMI collapsing, barcode collapsing, and read clipping algorithms to generate results maximally similar to the default parameters of the "cellranger count" command in Cell Ranger v.6: "--soloUMIdedup 1MM_CR --soloCBmatchWLtype 1MM_multi_Nbase_pseudocounts --soloUMIfiltering MultiGeneUMI_CR --clipAdapterType CellRanger4 --outFilterScoreMin 30". For cell filtering, the EmptyDrops algorithm of Cell Ranger v.4 and above was invoked using "--soloCellFilter EmptyDrops_CR". The option "--soloFeatures Gene GeneFull Velocyto" was used to generate both exon-only and full length (pre-mRNA) gene counts, as well as RNA velocity output matrices. TCR-seq samples were processed using Cell Ranger v.6.1.1 with VDJ reference vdj-GRCh38-alts-5.0.0. The default settings of the reference-based "cellranger vdj" command were used. Fastq files were

converted to <Sample>_S1_L001_R1_001.fastq.gz format to be compatible with Cell Ranger.

## scRNA-seq quality control, data integration and annotation

Jupyter notebooks used for data quality control, preprocessing, integration and annotations are available in the GitHub repository for this manuscript (Code availability). Scanpy v.1.9.1 with anndata v.0.10.7 and the statistics and plotting libraries pandas v.2.2.2, numpy v.1.26.4, scipy v.1.13.0, seaborn v.0.13.2 and matplotlib v.3.8.4 were used for data analysis and visualization. Mapped libraries were subjected to computational removal of ambient RNA using CellBender[42] v.0.1.0. Next, all datasets underwent cell quality-control filtering and cells with <400 or > 6500 genes, >6% mitochondrial reads or <5% ribosomal counts were removed. Doublets were annotated using Scrublet[43] v.0.2.3. Next, datasets were integrated with scVI from scvi-tools[44] v.0.19.0, for which mitochondrial, TCR and cell cycle genes were removed, and cells were annotated into major lineages (cell_type_level_0: T_DN, T_DP, T_SP, Epithelial, Stroma, Myeloid, RBC, B and Schwann) by Leiden clustering. Individual cell lineages were then separated and integrated with scVI to perform fine-grained annotation and remove remaining doublets picked up by manual annotation. Cell annotations were assigned based on four sequential steps: (1) high-resolution Leiden clustering was performed to find all potential cell clusters. (2) Annotations of new cells generated in this study were predicted (i) based on KNN graph majority voting of neighbouring cells with annotations from previous studies by adaptation of the weighted KNN transfer solution from scArches[45], or (ii) automatic label transfer using CellTypist[46] v.1.6.2 with the Developing_Human_Thymus or Pan_Fetal_Human models. (3) Calling the cell type annotation for a given cluster was then informed by a combination of the predicted labels and by marker genes reported in the literature. (4) Additional QC was performed and newly detected doublet clusters were removed where applicable. Steps 1–4 were repeated until final, fine-grained annotations were reached for the highest resolution (cell_type_level_4). Importantly, cell clusters that passed quality control but that we could not confidently assign to a defined cell type either in the literature or by cell markers by the strategy described above, were kept in the integrated object as 'cell_type_level_4_explore', which we recommend for future exploration and validation of these cell states. Annotations in 'cell_type_level_4' were grouped into five hierarchical levels from the finest (cell_type_level_4) to the broadest (cell_type_level_0) (Supplementary Table 7).

Single-cell TCR-sequencing data were processed using Dandelion[47] v.0.3.1, and a detailed notebook can be found in the GitHub repository for this Article (see Code availability). In brief, a Dandelion class object with n_obs was constructed from all combined TCR libraries. Cells were then further subset to only include DP and SP subtypes that contained V and J rearrangements for both *TRB* and *TRA* loci. Next, milo neighbourhoods were constructed based on scVI neighbourhood graphs (n_neighbors = 100) and VDJ genes and frequency feature space was calculated.

## mTEC trajectory analysis

The scFates package[48] v.1.0.7 was used for trajectory analysis on the combined mTECII and mTECIII cells. mTECII and mTECIII cells from the paediatric scRNA-seq dataset were reintegrated and batch-corrected using scVI. Next, the object was preprocessed according to recommendations from Palantir[49] and the scVI latent embedding was used as an input to 'palantir.utils.run_diffusion_maps' (Palantir v.1.3.3). Tree learning was performed using scf.tl.tree by using multiscale diffusion space from Palantir as recommended by scFates. Next, the node that was characterized by expression of some mTECI markers (*ASCL1*, *CCL21*) together with mTECII markers (*AIRE*, *FEZF2*) was used as a root to compute pseudotime using scf.tl.pseudotime. Finally, milestones obtained after running the pseudotime were used to adjust the annotations. To derive differentially expressed genes across pseudotime branches, we applied the scf.tl.test_fork function, followed by scf.tl.branch-

specific as described in the scFates article[48]. In brief, it fits a generalized additive model for each gene using pseudotime, branch and interaction between pseudotime and branch as covariates; two-sided *P* values were extracted for the interaction term (pseudotime:branch) and corrected using FDR to obtain significant differentially expressed genes. Next, these genes were tested for upregulation in each branch and assigned to different branches based on a cut-off of 1.3-fold upregulation.

## mcTEC differentiation potential analysis

The STEMNET package[50] (v.0.1) was used to determine the differentiation potential of mcTEC progenitor cells. For this purpose, only mTEC, cTEC and mcTEC(-Prolif) cells were retained in the dataset. mTECI/II/III and cTECI/II/III were set as maturation endpoints and the probability of each cell to adopt any of these six possible fates was calculated. To identify priming of mcTEC(-Prolif) towards mTECI versus cTECI fate, we derived a priming score by calculating the difference between the posterior probabilities for these two fates for each cell. Cells with priming score <−0.5 or >0.5 were labelled as mTECI-primed and cTECI-primed, respectively, whereas all other cells with comparable mTECI and cTECI potential were deemed to be unprimed.

## Formulation of the CMA with OrganAxis

OrganAxis is a mathematical model aimed to derive the relative, signed position of a point in space in respect to two morphological landmarks. The OrganAxis base function $H$ is highly flexible and tuneable with respect to the research question, spatial resolution and sampling frequency. In Supplementary Notes 1 and 2, we provide a detailed guide on tissue annotations with TissueTag v.0.1.1 and the details of OrganAxis derivation. The CMA is an extrapolation of OrganAxis, which is defined by a weighted linear combination of two $H$ functions (CMA = $0.2 \times H$(edge-to-cortex) + $0.8 \times H$(cortex-to-medulla)). All IBEX and Visium images were annotated with TissueTag v.0.1.1 at a resolution of 2 um per pixel (ppm = 0.5). Then, annotations were transferred to a quasi-hexagonal grid that was generated by placing points with r-microns spacing in the x and y directions and staggering every other row by r/2. Throughout this study we used r = 15. L2 distances to broad level annotations (annotation_level_0) and the corresponding CMA values were calculated with $k = 10$ for all spots in the hexagonal grid. L2 distances to fine annotations (annotation_level_1) were calculated with $k = 1$. CMA and L2 distances were then transferred to spots in Visium datasets or nuclei segmentations for IBEX by nearest-neighbour mapping and were therefore spatially homogeneous for these two spatial technologies.

To provide a common reference, we also binned the axis to a sequential discrete space for the entire thymus by ten levels (capsular, subcapsular, cortical level 1, cortical level 2, cortical level 3, cortical CMJ, medullary CMJ, medullary level 1, medullary level 2, medullary level 3; details are provided in Supplementary Note 2 and Supplementary Table 8).

## 10x Visium spatial transcriptomics sample processing and sequencing

Resected fetal and paediatric thymi were directly moved to HypoThermosol (Sigma-Aldrich, H4416-100ML) (paediatric samples) or cold PBS (fetal samples), shipped by courier with ice packs and processed within 24 h from time of surgery. For embedding, fetal or paediatric thymus tissue was first transferred to PBS and then placed onto ice for a few minutes to clear away any excess medium and preservation liquid (such as HypoThermosol). Next, as much liquid as possible was removed from the sample and, if necessary, the tissue was trimmed to fit into a cryomold. The sample was placed into a cryomold (Tissue-Tek, AGG4581) filled with OCT (Leica biosystems, 14020108926) and positioned according to the desired orientation. The cryomold was then placed in isopentane that had been equilibrated to −60 °C and left to fully freeze for 2 min. The sample was then rested on dry ice to allow draining

of the isopentane. Finally, the cryomolds were wrapped in foil and stored at −80 °C. On the day of sectioning, the samples were removed 1 h before sectioning and placed into the cryostat (Leica biosystems) at −18 °C to equilibrate. Tissue was sectioned (section thicknesses are provided in Supplementary Table 2) and sections were placed onto Visium slides according to the manufacturer's protocol. The sections were stained with H&E and imaged at ×20 magnification (Hamamatsu Nanozoomer 2.0 HT). Libraries were further processed according to the manufacturer's protocol (Visium Spatial Gene Expression Slide & Reagent Kit, 10x Genomics, PN-1000184) (permeabilization times are shown in Supplementary Table 2). The samples were sequenced on the NovaSeq 6000 sequencer (Illumina) and the obtained fastq files were mapped with Space Ranger (10x Genomics; version numbers are provided in Supplementary Table 2).

### Visium preprocessing, image registration and annotation
To process the Visium histology image data in higher resolution than the SpaceRanger defaults, we built a custom pipeline to extract an additional layer of image resolution at up to 5,000 pixels (hires5K), which we found to be more suitable for morphological analysis. We also developed our own fiducial image registration pipeline for increased accuracy where the fiducials are detected with cellpose v.2.1.1 and RANSAC from scikit-image v.0.22.0 is used for affine registration of reference fiducial frame (information provided by 10x Genomics). Lastly, for flexible tissue detection, we used Otsu thresholding with an adjustable threshold.

We subsequently used TissueTag v.0.1.1 for semiautomated image annotation (Supplementary Note 1). Cortical and medullary pixels were predicted with a pixel random forest classifier by generating training annotations based on spots with the highest gene expression of *AIRE* (for medulla) and *ARPP21* (for cortex). Automatic cortex/medulla annotations were then adjusted manually where necessary. Moreover, we manually annotated individual thymic lobes and specific structures, such as capsule/edge, freezing/sectioning artefacts, HCs, PVS and fetal thymus-associated lymphoid aggregates (as defined previously[16]). The morphological annotation and evaluations were done in consultation with expert human thymic pathologists. A full example of the Visium processing pipeline and annotation is provided on the GitHub repository for this Article (see Code availability).

### Spatial Visium mapping with cell2location
To ensure the best possible matching between Visium and single-cell profiles, we performed spatial mapping using cell2location[51] v.0.1.3 separately for fetal and paediatric datasets. We therefore subset our single-cell reference datasets according to either fetal or paediatric stage and removed rare cell types that were predominantly found in one of these stages. We then further removed cell types that showed stress signatures (which we believed to originate from technical factors) and cell types with the total number of cells < 40. A list of cell types that were excluded and the exclusion criteria are provided in Supplementary Table 9. Before cell2location deconvolution, we removed cell cycle genes (Supplementary Table 10) and mitochondrial genes, as well as TCR genes using the regex expression '^TR[AB][VDJ]|^IG[HKL][VDJC]'. We then further calculated highly variable genes and used relevant metadata cofactors (sample, chemistry, study, donor, age) to correct for batch effects in the cell2location model.

### Quality control and batch correction of Visium data
After deconvolution, all Visium data were subjected to filtering based on read coverage and predicted cell abundance. Spots with fewer than 1,000 genes per spot or fewer than 25 predicted cells were omitted. Furthermore, annotated tissue artefacts and areas not assigned to a specific structure were removed. Next, to generate a common embedding we performed scVI integration after removing cell cycle, mitochondrial and TCR genes from the highly variable gene selection. scVI training

was performed with 'SampleID' as the batch key, 'SlideID, Spaceranger, section_thickness(um)' as categorical covariates, and 'Age(numeric), n_genes_by_counts' as continuous covariates.

Before performing any association analysis with the CMA, we further removed lobules (based on 'annotations_lobules_0') that had no or small medullar or cortical regions, as we expected our CMA model to be less accurate in these cases.

### PCA cumulative contribution of CMA
To estimate the dependency of the axis on spot gene variance across samples, we first normalized to a target sum of 2,500 counts and performed log transformation followed by combat regression (https://scanpy.readthedocs.io/en/stable/api/generated/scanpy.pp.combat.html) by sample to adjust for the batch effect of individual samples. We then computed the PCA for batch-corrected gene expression and calculated the Spearman correlation between the first ten PCs and CMA or the number of genes detected per spot (n_gene_by_counts). Note that number of genes per spot, in our hands, was mostly influenced by inconsistent permeabilization during Visium library preparation and constituted the largest technical source of within-sample variance that we found in both fetal and paediatric Visium samples. To estimate the cumulative contribution of either CMA or the number of genes per spot, we multiplied the Spearman's *R* with the percentage cumulative explained variance of the first 10 PCs.

### Cytokine clustering by expression along the binned axis in Visium data
To analyse cytokine gradients based on the spatial distribution across CMA bins, we first selected a group of 65 cytokines that were broadly expressed from the CellphoneDB database (v.4.1.0, genes annotated as 'Cytokine', 'growthfactor | cytokine', 'cytokine', 'cytokine | hormone'). We excluded cytokines that were expressed in less than 5% of the spots in all CMA bins. We then performed hierarchical clustering on gene expression batch-corrected (combat) fetal Visium samples of the standardized mean expression of genes across bins using the Ward linkage method with the linkage function from the scipy.cluster.hierarchy module. A heat map was generated using the matrixplot function from the scanpy package[52].

### Cytokine two-way ANOVA analysis and cosine similarity on binned axis Visium data
To compare the distribution of cytokines across developmental groups (fetal versus paediatric) and identify differentially distributed genes, we implemented a two-way ANOVA approach. We initially log-normalized Visium gene expression, then removed lobules for which not a single cortex or medulla Visium spot was detected to increase CMA confidence in both datasets. Data were grouped by mean expression per sample and CMA bin, such that each sample had a single datapoint per CMA bin; $n = 16$ (paediatric) and $n = 12$ (fetal) samples. Cosine similarity was calculated based on the median values of the pooled sample bins between fetal and paediatric gene profiles with sklearn.metrics.pairwise.cosine_similarity from scikit-learn v.0.22.0. Two-way ANOVA for age group (fetal versus paediatric) and CMA bin was calculated with statsmodels.api.stats.anova_lm(model, type=2). *P* values for main and interaction effects were Bonferroni corrected with statsmodels.stats.multitest.multipletests (pvals, alpha=0.05, method='bonferroni'). For the full report of the results refer to Supplementary Table 5.

### Cell- and gene-weighted mean CMA location on Visium data
To estimate the average position of a cell or gene distribution along CMA and HC axes (L2 distance to the nearest HC), spots with low gene expression were filtered out by using appropriate thresholds (0.2 for scVI corrected gene expression and 0.5 for predicted cell abundances). The position of a gene or cell was then calculated according to the following formula: for every gene/cell and axis positions, the weighted

mean was calculated as a dot product of spot cell abundance values and CMA position divided by the sum of the cell abundance values across spots.

### Identification of cell-type-specific SGs

To identify genes exclusively expressed in a specific cell type or subset thereof ('specialization genes', SGs), we developed custom Python functions. Starting from raw read count, gene expression was scaled with scipy.stats.zscore(). Cells that showed expression below a cut-off of 0.05 and genes that had expression below 1.5 mean counts were excluded from further steps. Next, a quantile threshold (>95%) was used to select cells with the highest expression level of a specific gene. A $\chi^2$ test (scipy chi2_contingency) was performed per gene to identify if the selected cells were over-represented in a specific cell type (cell_type_level_4_explore), indicating the gene to be a marker gene. Genes that were predicted to be expressed only in a single cell type ($\chi^2$ $\alpha = 1 \times 10^{-50}$) were considered to be SGs and used for further analyses.

### IBEX clinical cohort details and sample preparation

Human thymus samples were obtained from the pathology department of the Children's National Medical Center in Washington, DC, after cardiothoracic surgery from children with congenital heart disease, as thymic tissue is routinely removed and discarded to gain adequate exposure of the retrosternal operative field. Use of these thymus samples for this study was determined to be exempt from review by the NIH Institutional Review Board in accordance with the guidelines issued by the Office of Human Research Protections. There were no genetic concerns for the patients in this cohort. Details about the cohort can be found in Supplementary Table 2. Human thymi were placed in PBS on receipt and processed within 24 h after surgery. Excess fat and connective tissue were trimmed and sectioned into <5 mm cubes. For IBEX imaging, human thymi were fixed with BD CytoFix/CytoPerm (BD Biosciences) diluted in PBS (1:4) for 2 days. After fixation, all tissues were washed briefly (5 min per wash) in PBS and incubated in 30% sucrose for 2 days before embedding in OCT compound (Tissue-Tek) as described previously[17,53].

### IBEX sample imaging preparation

IBEX imaging was performed on fixed frozen sections as described previously[17,53]. In brief, 20 μm sections were cut on a CM1950 cryostat (Leica) and adhered to two-well chambered cover glasses (Lab-tek) coated with 15 μl of chrome alum gelatin (Newcomer Supply) per well. Frozen sections were permeabilized, blocked and stained in PBS containing 0.3% Triton X-100 (Sigma-Aldrich), 1% bovine serum albumin (Sigma-Aldrich) and 1% human Fc block (BD Biosciences). Immunolabelling was performed with the PELCO BioWave Pro 36500-230 microwave equipped with a PELCO SteadyTemp Pro 50062 Thermoelectric Recirculating Chiller (Ted Pella) using a 2-1-2-1-2-1-2-1-2 program. The IBEX thymus antibody panel can be found in Supplementary Table 3 and has been formatted as an Organ Mapping Antibody Panel[54] (OMAP-17) accessible online (https://humanatlas.io/omap). Custom antibodies were purchased from BioLegend or conjugated in house using labelling kits for Lumican (AAT Bioquest, 1230) and LYVE-1 (Thermo Fisher Scientific, A20182). A biotin avidin kit (Abcam, ab64212) was used to block endogenous avidin, biotin and biotin-binding proteins before streptavidin application. Cell nuclei were visualized with Hoechst 33342 (Biotium) and the sections were mounted using Fluoromount G (Southern Biotech). Mounting medium was thoroughly removed by washing with PBS after image acquisition and before chemical bleaching of fluorophores. After each staining and imaging cycle, the samples were treated with two 15 min treatments of 1 mg ml$^{-1}$ of LiBH$_4$ (STREM Chemicals) prepared in deionized H$_2$O to bleach all fluorophores except Hoechst.

### IBEX image acquisition and alignment

Representative sections from different tissues were acquired using the inverted Leica TCS SP8 X confocal microscope with a ×40 objective (NA 1.3), 4 HyD and 1 PMT detectors, a white-light laser that produces a continuous spectral output between 470 and 670 nm as well as 405, 685 and 730 nm lasers. Panels consisted of antibodies conjugated to the following fluorophores and dyes: Hoechst, Alexa Fluor (AF)488, FITC, AF532, phycoerythrin (PE), eF570, AF555, iFluor 594, AF647, eF660, AF680 and AF700. All images were captured at an 8-bit depth, with a line average of 3 and 1024 × 1024 format with the following pixel dimensions: $x$ (0.284 μm), $y$ (0.284 μm) and $z$ (1 μm). Images were tiled and merged using the LAS X Navigator software (LAS X v.3.5.7.23225). For IBEX tissue imaging, multiple tissue sections were examined before selecting a representative tissue section that contained several distinct lobules with multiple functional units, often resulting in an unusually shaped region of interest. Fluorophore emission was collected on separate detectors with sequential laser excitation of compatible fluorophores (3–4 per sequential) used to minimize spectral spillover. The Channel Dye Separation module within LAS X v.3.5.7.23225 (Leica Microsystems) was then used to correct for any residual spillover. For publication-quality images, Gaussian filters, brightness/contrast adjustments and channel masks were applied uniformly to all images. Image alignment of all IBEX panels was performed as described previously using SimpleITK[55,56]. Additional details on antibodies, protocols and software can be found on the IBEX Knowledge-Base (https://doi.org/10.5281/zenodo.7693279).

### IBEX 3D single nuclei segmentation with cellpose

IBEX images were converted from .ims format (Imaris, Oxford Instruments, v.9.5.0 and v.10.0.0) to 3D stacks (TIFF) by individual channels with FIJI v.1.54j. We then applied a custom pipeline for 3D single nuclei segmentation with cellpose v.2.1.1:

(1) Image preparation: individual TIFF images were separated by $z$ plane and channel and the sample channel metadata were extracted as a .csv file. The nuclear staining channel (Hoechst) was located, and a set of tiled 3D image arrays was generated.

(2) Segmentation: the image arrays were sequentially segmented with cellpose using specific parameters, for example, diameter, resampling, anisotropy, thresholds and batch size. We used tiles to overcome restrictions on GPU and RAM resources.

(3) Restitching and formatting: after segmentation, tiles were stitched back together, and all segmentation masks were stored in a uint32-bit image format to hold more than 65,535 mask labels, as a typical sample produces 200,000–700,000 cell masks.

(4) Handling high nuclear density: the high nuclear density, especially in the thymic cortex, produced scenarios in which segmentation masks were often bordered by neighbouring cells, resulting in substantial inter-cell signal bleed. To overcome this issue, we excluded pixels in the interface between two cells such that the mask boundary was preserved if no neighbouring cell was present.

(5) Mask filtering: we removed small cell masks to avoid noise and cell fragments using skimage.morphology.remove_small_objects.

(6) Signal intensity extraction: for each cell mask post-filtering, we extracted the mean and maximum signal intensity for each channel in the IBEX image. This produced a cellXprotein file for each sample.

(7) Data merging: once all sample single-cell segmentations were collected, we merged all samples ($n = 8$) and stored metadata, channel and spatial information in a unified AnnData object, totalling more than 1.1 million cells.

### Label transfer to IBEX cells from scRNA-seq and CITE-seq reference atlas with ISS patcher

We used a KNN algorithm to compare the annotated cell types in our scRNA-seq reference atlas with the IBEX single-cell segmentations. For this purpose, the protein expression in the IBEX query cells was matched with the RNA expression of the corresponding genes in the scRNA-seq reference. Protein and gene names were matched according to Supplementary Table 11. Batch effects were removed from IBEX with

scanpy.pp.combat(ibex_gene, key='sample', inplace=True). We next subsetted each IBEX sample and ran the KNN prediction algorithm per sample with $k = 30$, including the following steps:

(1) The shared feature space between the two objects was identified, log-normalized and z-scored on a per-object basis.

(2) The KNN of the low-dimensional observations (IBEX) in the high-dimensional space (GEX) were identified. The counts of absent features were imputed as the mean of the high-dimensional neighbours.

(3) On the basis of majority voting, the most frequent cell annotation in the GEX reference was assigned to the IBEX query cell. The proportion of KNNs that contributed to the majority voting was recorded as the evidence fraction (KNNf). For example, if out of the 30 nearest neighbours, 13 were labelled as A, 10 as B and 7 as C in GEX, then the IBEX cell received the label A. The fraction in this case refers to the proportion of the 30 nearest neighbours that contributed to the majority label, which would be $K$NNf = 13/30 = 0.43 for label A.

In some of our IBEX samples AIRE staining in particular produced a relatively high level of non-specific signal and low signal-to-noise ratio. As a consequence, we identify some predicted AIRE[+] mTECII cells that are not in the medulla and have capsular/cortical localization (Fig. 4g). We had cases of individual samples for which a specific antibody did not perform well or was missing (for example, for IBEX_01 the KRT10 antibody was out of stock). However, as ISS patcher was run on each sample separately, this did not affect proper scaling of that marker and its use of it for mapping in other samples. Moreover, by selecting cells with a higher proportion of matched KNN cells from the single-cell data (KKNf) these effects are reconciled through the removal of cells with low-confidence mapping. We would like to flag this point for future researchers who want to reuse our datasets.

The general code for the ISS patcher KNN mapping algorithm can be found in the dedicated GitHub repository (https://github.com/Teichlab/iss_patcher/tree/main) and the full example for KNN mapping using IBEX is reported in the GitHub repository for this Article (see Code availability).

To annotate T cell types in the IBEX data with high accuracy, we applied the ISS patcher with the CITE-seq T lineage data as a reference as follows: we first applied the KNN algorithm to IBEX data using the scRNA-seq reference atlas to identify and subset IBEX T lineage cells. We then used the CITE-seq data as a reference to repeat the KNN-based annotation on these selected cells. This KNN-based reannotation was performed on a hybrid RNA/protein reference, which included protein measurements for the 19 markers assayed in both CITE-seq and IBEX in addition to RNA measurements for the remaining 23 genes as described in Supplementary Table 11. We used the same KNN implementation as described above but with $k = 7$, while also imputing CD4 and CD8 pseudotime.

### Human thymus FFPE processing

Human paediatric samples were obtained from cardiac corrective surgeries. Removed thymi were directly moved to HypoThermosol (Sigma-Aldrich, H4416-100ML), shipped with a courier with ice packs and processed in under 24 h after surgery. After arrival, for FFPE processing, the samples were cut to approximately 1 cm² pieces with sharp scissors in 1× DPBS. Tissue pieces were then rinsed in clean DPBS to remove any excess HypoThermosol, patted dry with wipes (Kimtech) and placed into 10% formalin (cellpath, BAF-6000-08A) for 16–24 h at room temperature. The next day, tissues were dehydrated and embedded in wax, then kept at 4 °C.

### RareCyte immunostaining and 14-plex imaging

Multiplex immunofluorescence and single round imaging was performed as described previously[57]. All steps were performed at room temperature unless stated otherwise. In brief, FFPE blocks were sectioned using a microtome (Leica, RM2235) at 3.5–5 μm thickness and placed onto a superfrost slide (Thermo Fisher Scientific, 12312148). Slides were dried at 60 °C for 60 min to ensure that tissue sections had adhered to the slides. Tissue sections were deparaffinized and subjected to antigen retrieval using the BioGenex EZ-Retriever system (95 °C for 5 min followed by 107 °C 5 min). For OCT sections, 7 μm sections were taken using a cryostat (Leica CM3050S), placed onto SuperFrost Plus slides (VWR) and immediately submerged in 10% buffered saline formalin (Cellpath, BAF-6000-08A) for 1 h at room temperature. The samples were then subjected to the following steps similarly to the FFPE samples. To remove autofluorescence, slides were bleached with AF quench buffer (4.5% $H_2O_2$, 24 mM NaOH in PBS). The slides were quenched for 60 min using the 'high' setting with a strong white-light exposure followed by further quenching for 30 min using the 365 nm 'high' setting using a UV transilluminator. The slides were rinsed with 1× PBS and incubated in 300 μl of Image-iT FX Signal Enhancer (Thermo Fisher Scientific, I36933) for 15 min. The slides were rinsed again and 300 μl of labelled primary antibody staining cocktail was added to the tissue, which was subsequently incubated for 120 min in the dark within a humidity tray. All antibodies were prediluted according to company recommendations and were not adjusted further (Supplementary Table 4). The slides were washed with a surfactant wash buffer and 300 μl of nuclear staining in goat diluent was added to the slide. The slides were then incubated in the dark for 30 min in a humidity tray. The slides were then washed and placed in 1× PBS. Finally the slides were coverslipped using ArgoFluor mount medium and left in the dark at room temperature overnight to dry. The slides were imaged the next day using the RareCyte Orion microscope with a ×20 objective and relevant acquisition settings were applied using the software Artemis v.4.

### RNAscope processing and imaging

For RNAscope analysis, thymus tissue was processed as described above for Visium sectioning. Sections were cut from the fresh frozen OCT-embedded (OCT, Leica) samples at a thickness of 10 μm using a cryostat (Leica, CM3050S) and placed onto SuperFrost Plus slides (VWR). Sections were stored at −80 °C until staining. The sections were removed from the −80 °C storage and submerged in chilled (4 °C) 4% PFA for 15 min, then acclimatized to room temperature 4% PFA over 120 min. The sections were then briefly washed in 1× PBS to remove any remaining OCT. Then, the sections were dehydrated in a series of 50%, 70%, 100% and 100% ethanol (5 min each) and air-dried before performing automated 4-plex RNAscope.

Using the automated Leica BOND RX, RNAscope staining was performed on the fresh frozen sections using the RNAscope LS multiplex fluorescent Reagent Kit v2 Assay and RNAscope LS 4-Plex Ancillary Kit for LS Multiplex Fluorescent (Advanced Cell Diagnostics (ACD), Bio-Techne) according to the manufacturer's instructions. All of the sections were subjected to 15 min of protease III treatment before staining protocols were performed. Before running RNAscope probe panels, the RNA quality of fresh frozen samples was assessed using multiplex positive (RNAscope LS 2.5 4-plex Positive Control Probe, ACD Bio-Techne, 321808) and negative (RNAscope 4-plex LS Multiplex Negative Control Probe, ACD Bio-Techne, 321838) controls.

The probes were labelled using Opal 520, 570 and 650 fluorophores (Akoya Biosciences, 1:1,000) and one probe channel was labelled using Atto 425-streptavidin fluorophore (Sigma-Aldrich, 1:500), which was first incubated with TSA−biotin (Akoya Biosciences, 1:400). The following RNAscope 2.5 LS probes were used for this study: Hs-AIRE (ACD Bio-Techne, 551248), Hs-LY75-C2 (ACD Bio-Techne, 481438-C2), Hs-CAMP-C3 (ACD Bio-Techne, 446248-C3), Hs-EPCAM-C4 (ACD Bio-Techne, 310288-C4), Hs-IGFBP6-C1 (ACD Bio-Techne, 496068) and Hs-DLK2-C3 (ACD Bio-Techne, 425088-C3). All nuclei were DAPI stained (Life Technologies, D1306). Details are provided in Supplementary Table 12.

Confocal imaging was performed on the Perkin Elmer Operetta CLS High Content Analysis System using a ×20 (NA 0.16, 0.299 μm px$^{-1}$) water-immersion objective with 9-11 z-stacks with 2 μm step. Channels: DAPI (excitation, 355–385 nm; emission, 430–500 nm), Atto 425 (excitation, 435–460 nm; emission, 470–515 nm), Opal 520 (excitation, 460–490 nm; emission, 500–550 nm), Opal 570 (excitation, 530–560 nm; emission, 570–620 nm), Opal 650 (excitation, 615–645 nm; emission, 655–760 nm). Confocal image stacks were stitched as individual z stacks using proprietary Acapella scripts provided by Perkin Elmer, and visualized using OMERO Plus (Glencoe Software).

The contrast used for Extended Data Fig. 6e was as follows: *DLK2* (magenta 150–500), *IGFBP6* (yellow 200–1500), *LY75* (green 200–4000), *EPCAM* (red 300–2500).

## CITE-seq tissue processing
Paediatric thymus samples from children undergoing cardiac surgery were obtained according to and used with the approval of the Medical Ethical Commission of Ghent University Hospital, Belgium (EC/2019-0826) through the haematopoietic cell biobank (EC-Bio/1-2018). Thymus tissue was cut into small pieces using scalpels and digested with 1.6 mg ml$^{-1}$ collagenase (Gibco, 17104-019) in IMDM medium for 30 min at 37 °C with regular agitation to generate a single-cell suspension. The reaction was quenched with 10% FBS and the thymocyte suspension was passed through a 70 μm strainer to remove undigested tissue. Cells were frozen in FBS containing 10% DMSO and stored in liquid nitrogen until needed.

## CITE-seq antibody preparation
The TotalSeq-C Human Universal Cocktail 1.0 (BioLegend, 399905) was resuspended according to the manufacturer's instructions. In brief, the lyophilized cocktail was equilibrated to room temperature for 5 min and then centrifuged at 10,000g for 30 s. Then, 27.5 μl cell staining buffer (BioLegend, 420201) was added and the tube was vortexed for 10 s, incubated for 5 min at room temperature and vortexed again. The resuspended cocktail was centrifuged for 30 s at room temperature at 10,000g and the entire volume was transferred to a low-bind tube. Finally, the tube was centrifuged again at 14,000g for 10 min at 4 °C and 25 μl of the supernatant was used per sample ($2 \times 10^6$ cells in 200 μl).

In total, 13 TotalSeq-C antibodies (BioLegend) were titrated individually by flow cytometry using PE-conjugated versions of the same antibody clone as recommended by BioLegend. After choosing a suitable concentration for each antibody, a master mix was prepared for cell staining. For this, all antibodies were initially diluted in the cell staining buffer to obtain a concentration 100-fold higher than the desired final staining concentration. Then, 2 μl of each diluted antibody substock was combined in a master mix, which was added to the cells for labelling in a total volume of 200 μl. Details on the TotalSeq-C antibodies are provided in Supplementary Table 6.

## CITE-seq sample preparation
Cells were thawed slowly by gradually adding 15 volumes of pre-warmed IMDM medium and pelleted at 1,700 rpm for 6 min at 4 °C. After resuspending in PBS, cells were passed through a 70 μm strainer to remove clumps. Enrichment for viable cells was achieved using a magnetic bead-based dead cell removal kit (Miltenyi, 130-090-101). For this, cells were pelleted as before, washed with 1× binding buffer (part of the kit, prepared with sterile distilled water) and resuspended in dead cell removal microbeads (part of the kit) at a concentration of $10^7$ cells per 100 μl beads. After incubation at room temperature for 15 min, cells were applied to an LS column (Miltenyi, 130-122-729), which was prerinsed with 3 ml 1× binding buffer. The column was washed four times with 3 ml binding buffer and the flow-through containing viable cells was collected. Cells in the flow-through were pelleted and viability was confirmed using trypan blue. A total of $2 \times 10^6$ viable cells was used for TotalSeq-C and anti-CD3-PE antibody staining. For this purpose, cells were washed with cell staining buffer (BioLegend, 420201), pelleted at 600g for 10 min at 4 °C and resuspended in 90 μl cell staining buffer. Then, 10 μl Human TruStain FcX blocking solution (BioLegend, 422301) was added and cells were incubated for 10 min at 4 °C. The TotalSeq-C Human Universal Cocktail 1.0 (BioLegend, 399905) was resuspended as described above, centrifuged at 14,000g for 10 min at 4 °C and 25 μl of the supernatant was added to the blocked cells. Individual TotalSeq-C antibodies were prepared as described above and 26 μl of the master mix was added to each sample. To facilitate enrichment of immature and mature thymocytes by FACS, 10 μl anti-CD3-PE (SK7, BioLegend, 344805) was added and the samples were topped up with 40 μl cell staining buffer resulting in a total staining volume of 200 μl. The samples were incubated for 30 min at 4 °C in the dark. To wash off unbound antibodies, cell staining buffer was added to the samples, and cells were pelleted for 10 min at 600g at 4 °C. All supernatant was removed, cells were resuspended in cell staining buffer, transferred to a new tube and pelleted as before. Cells were again resuspended in cell staining buffer and pelleted and this wash step was repeated once more before cells were resuspended in 200 μl MACS buffer (2% FCS, 2 mM EDTA in PBS) in preparation for sorting. Then, 1 μl propidium iodide (Invitrogen, 230111) was added for detection of dead cells and samples were sorted on the BD FACSAria III or BD FACSAria Fusion cell sorter using a 100 μm nozzle and a maximum flow rate of 4 (FACSDiva v.8.0.2, reanalysis with FlowJo v.10.8.2). Cells were gated using forward/side scatter to remove doublets and debris, then dead cells were excluded based on PI staining. CD3$^-$ and CD3$^+$ cells were collected separately in cooled IMDM + 50% FCS (Supplementary Fig. 19b). After completion of the sort, collection tubes were topped up with DPBS and cells were pelleted at 400g for 10 min at 4 °C. The supernatant was removed and cells were resuspended at an estimated concentration of 1,500 cells per μl in PBS + 0.04% BSA (Miltenyi, 130-091-376), of which 16.5 μl was used in the GEM generation step. The Next GEM Single Cell 5′ Kit v2 (10x Genomics, 1000265) was used to prepare the reaction master mix, and load cells, gel beads and partitioning oil on a Chip K (10x Genomics, 1000286) according to the manufacturer's protocol CG000330 Rev A. GEMs were generated using a Chromium Controller (10x Genomics), transferred to a tube strip and reverse transcription was performed in a BioRad C1000 Touch Thermal Cycler according to the protocol. The samples were stored at 4 °C overnight and the library preparation was carried out the next day.

## CITE-seq library preparation and sequencing
Feature barcode (FB), gene expression (GEX) and TCR libraries were prepared according to protocol CG000330 Rev A (10x Genomics) using the Chromium Next GEM Single Cell 5′GEM Kit v2 (10x Genomics, 1000244), Library Construction Kit (10x Genomics, 1000190), 5′ Feature Barcode Kit (10x Genomics, 1000256), Human TCR Amplification Kit (10x Genomics, 1000252), Dual Index Kit TT set A (10x Genomics, 1000215) and Dual Index Kit TN set A (10x Genomics, 1000250). The protocol version for >6,000 cells was followed and libraries were amplified for 13 cycles (cDNA), 14 cycles (GEX), 8 cycles (FB) or 12 + 10 cycles (TCR libraries). Library quality and quantity were checked on the Bioanalyzer instrument (Agilent) using a high-sensitivity DNA assay. Libraries were pooled and sequenced on the NovaSeq 6000 instrument (Illumina) to a minimum of 25,000 reads per cell for GEX, 10,000 reads per cell for FB and 5,000 reads per cell for TCR libraries.

## CITE-seq quality control and denoising
CITE-seq data were processed using the R packages Seurat[58] (v.4.3.0), SeuratObject (v.4.1.4), SeuratDisk (v.0.0.0.9021), SingleCellExperiment (v.1.24.0), Matrix (v.1.6-4), matrixStats (v.1.2.0), dplyr (v.1.1.4), tidyr (v.1.3.1), reshape2 (v.1.4.4), BiocNeighbors (v.1.20.2), BiocParallel (v.1.36.0), stringr (V.1.5.1), reticulate (v.1.35.0) and sceasy (v.0.0.7).

Data were visualized using ggplot2 (v.3.5.0), ggrastr (v.1.0.2), ggridges (v.0.5.6) and RColorBrewer (v.1.1-3).

Fastq files from FB libraries were mapped with Cell Ranger v.7.0.0. GEX libraries were mapped with STARsolo as described above for scRNA-seq data. CITE-seq data were first subjected to quality-control processing based on RNA properties as described above. For the retained high-quality cells, ADT data were then denoised using dsb (v.1.0.3)[59]. For this purpose, empty droplets were identified in the unfiltered mapping output and used as a reference for estimating noise and antibody background levels. Approximately 1.4 million droplets were selected on the basis of RNA count < 240 reads, ADT count between 120 and 350 reads, and <5% mitochondrial reads to ensure that damaged cells were not included in the subset. During denoising and normalization with DSBNormalizeProtein, ADT data from droplets were used for background correction and the seven isotype control antibodies included in the TotalSeq-C Human Universal Cocktail were used to determine technical variation. Data were scaled based on the background by subtracting the mean and then dividing by the standard deviation of the empty droplets. Negative values after denoising, which correspond to very low expression, were set to zero to improve interpretation and visualization. In an additional quality-control step, cells in which less than 100 of the antibodies were detected were removed from the dataset as technical artefact. Moreover, a subset of cells affected by aggregates of the antibodies against TCRγδ, CD199 (CCR9), CD370, CD357 and XCR1 was removed from the dataset due to unreliable surface-staining properties.

## CITE-seq annotation

Annotation of the CITE-seq data was carried out on integrated RNA and denoised ADT modalities. For this purpose, both data modalities were log-normalized and the top 2,000 highly variable genes were identified, followed by PCA on the scaled highly variable genes using the standard functions in the Seurat package. MNN correction was applied to the PCA loadings matrix using the reducedMNN function from the Batchelor package[60] (v.1.18.1) to reduce batch and donor effects between samples. To integrate both modalities, multimodal neighbours and modality weights were identified and a weighted nearest neighbours (WNN) graph[58] was constructed using FindMultimodalNeighbors based on the PCA. The number of PCs to be used was determined based on the difference in variation of consecutive PCs being higher than 0.1%. A UMAP visualization was generated based on the WNN graph to represent the weighted combination of both modalities. Furthermore, a supervised PCA (sPCA) was performed on transcriptome data using the RunSPCA function of the Seurat package to obtain dimensionality reduction for the RNA modality that represents the structure of the WNN graph[58].

To identify cell types and developmental stages, we performed Leiden clustering at a low resolution based on the sPCA using the Seurat functions FindNeighbors and FindClusters. The obtained clusters were then subsetted and analysed individually starting with the most immature cluster, which was identified based on high expression of the surface marker CD34. For each subset, normalization and scaling for RNA and ADT data were repeated as described above and a new WNN UMAP and sPCA were constructed. A combination of Leiden clustering on the sPCA and thresholding of protein levels was used to identify cell types. Moreover, to identify proliferating cells, cell cycle scoring was performed using the G2M and S phase markers supplied in the Seurat package. The FindMarkers function (Seurat) and the package singleCellHaystack[61] (v.1.0.0) were used to identify differentially expressed genes and surface markers in a cluster-based and cluster-independent manner, respectively. Distinction of CD4 and CD8 lineage maturation stages was based on CD45RA, CD45RO and CD1a and identical expression cut-offs were used for both lineages to ensure that subsets would be directly comparable.

Paired TCR-seq data were processed with Dandelion[47] v.0.3.1 and information about productive or non-productive *TRA* and *TRB* rearrangements was extracted for each cell to validate cell type annotations after clustering.

## CITE-seq pseudotime analysis

To carry out trajectory inference for αβT lineage cells, CITE-seq data were subsetted to contain DP_pos_sel, DP_4hi8lo, SP_CD4_immature, SP_CD4_semi-mature, SP_CD4_mature, SP_CD8_immature, SP_CD8_semi-mature and SP_CD8_mature cells. A new WNN UMAP was constructed based on surface protein and RNA. Slingshot[62] (v.2.6.0) was used to establish a minimum spanning tree on the WNN UMAP using the getLineages function based on mutual nearest neighbour-based distance with DP_pos_sel set as start point and SP_CD4_mature and SP_CD8_mature specified as end points. Smooth lineages were obtained using the getCurves function and the derived pseudotime orderings were used to assess transcript and surface marker expression throughout differentiation.

## CITE-seq cell2location mapping, integration and processing

For the focused multimodal analysis, we subsetted the full scRNA-seq dataset to include only paediatric data and further removed T lineage cells without CITE-seq protein information. We then performed cell2location as described above to obtain deconvolved cell type mapping based on CITE-seq annotations. In addition, as cell2location deconvolves spots by unique pseudobulk expression profiles from a single-cell reference based on discrete cell annotations, the mapping resolution is limited to that of the annotated cell subsets. To investigate the continuous spatiotemporal nature of CD4/CD8 cell lineages at increased resolution, we performed high-resolution Leiden clustering on the CITE-seq data using scanpy (resolution=35), resulting in 567 cell clusters. These cell clusters were then mapped to our paediatric Visium data with cell2location as described above. To measure the position of each cell cluster across the CMA, we selected Visium spots with the highest cluster label abundance (above percentile 95%) and calculated the weighted mean CMA values for these spots. This value was then assigned this to the cells comprising the associated cluster in the single-cell object. Further details are provided in the GitHub repository for this Article (see Code availability).

## Reporting summary

Further information on research design is available in the Nature Portfolio Reporting Summary linked to this article.

## Data availability

The annotated fetal and paediatric integrated scRNA-seq atlas and Visium objects for this study can be explored online https://cellxgene.cziscience.com/collections/fc19ae6c-d7c1-4dce-b703-62c5d52061b4. Paediatric CITE-seq data can be visualised and explored using a custom ShinyApp (https://ccgg.ugent.be/shiny/htsa_thymocyte_citeseq/). Sequencing data for the newly generated libraries for scRNA-seq and Visium data were uploaded to ENA under accession code PRJEB77091. Several samples were obtained under consent agreements that require data release with managed access, which is why these were deposited at EGA under accession code EGAD00001015384. Imaging data for Visium samples were deposited at BioImage Archive under accession number S-BIAD1257. CITE-seq data were uploaded to the GEO under accession number GSE271304. Imaging data generated for this study, including IBEX, RNAscope and RareCyte data, were deposited at the BioImage Archive under the accession number S-BIAD1257. Publicly available datasets were downloaded from the following sources: ref. 3 (ArrayExpress: E-MTAB-8581; GEO: GSE206710); ref. 15 (GEO: GSE159745); ref. 7 (GEO: GSE147520); and ref. 16 (ArrayExpress: E-MTAB-11341). All accession codes are also listed in Supplementary Tables 1 and 2. Source data are provided with this paper.

## Code availability

All code scripts and notebooks used in the study are open and available to the public. All code relating to analysis and figures in this manuscript has been deposited at GitHub (https://github.com/Teichlab/thymus_spatial_atlas). Code for the TissueTag package is available at https://github.com/Teichlab/TissueTag (https://github.com/nadavyayon/TissueTag). Moreover, we have curated a dedicated online tutorial for OrganAxis (https://organ-axis-tutorial.readthedocs.io).

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

**Acknowledgements** We thank the donors and their families for their consent and contribution to this study; the supporting teams of the Teichmann wet laboratory, CellGenIT, CellGen wet laboratory, Spatial Genomics Platform teams (SGP) and Cellular Operations, who have been central for this study; Z. Yaniv for computational support for IBEX image alignment; the staff at the Sanger Flow Cytometry Facility and the Flow Cytometry Core Facility at Ghent University for support with sample sorting; E. De Meester (Ghent University) for help with single-cell sample preparation and sequencing; B. Vandekerckhove (Ghent University) for providing thymus samples through the haematopoietic cell biobank; A. Marx for providing guidance on thymus tissue annotations; and E. Howley for coordinating collection and screening of thymus tissue at GOSH (UK). Finally, we would like to thank our beloved friend and colleague, Daniele Muraro, who tragically passed away in 2022, for his support in the early stages of this project. This study was primarily funded by the Chan Zuckerberg Foundation by grant number 2019-002445 and 2022-249170 (https://doi.org/10.37921/644286wvygde). The Wellcome Sanger Institute is supported by core funding from the Wellcome Trust (220540/Z/20/A). Work in the Taghon laboratory was supported by the Fund for Scientific Research Flanders (FWO, grant G075421N) and the Concerted Research Action from the Ghent University Research Fund (GOA, BOF18-GOA-024 and BOF24-GOA-035). Work at the NIH was supported by the Division of Intramural Research of the National Institute of Allergy and Infectious Diseases, NIH (Grant AI001222 to L.D.N.) and by the National Cancer Institute. Work at CSCI is funded in whole, or in part, by the Wellcome Trust (203151/Z/16/Z, 203151/A/16/Z) and the UKRI Medical Research Council (MC_PC_17230). All research at GOSH is supported by the UK National Institute of Health Research and Great Ormond Street Biomedical Research Centre, AYK Wellcome Trust (222096/Z/20/Z) EGD GOSH Children's Charity. S.A.T. is supported by co-director funding from the Canadian Institute for Advanced Research (CIFAR); N.Y. by a joint ESPOD fellowship between EMBL-EBI and the Wellcome Sanger institute; L.B. by a junior postdoctoral fellowship (12D9523N) from the Fund for Scientific Research Flanders (FWO). Parts of Figs. 1a,e and 5b,e and Supplementary Fig. 2a were created using BioRender under institutional licence agreement ZX270PGOI5. For the purpose of open access, the authors have applied a CC-BY public copyright licence to any Author Accepted Manuscript version arising from this submission.

**Author contributions** S.A.T., R.N.G. and T.T. conceived the initial concept and acquired funding for the study. N.Y. with input from J.C.M., V.U. and S.A.T. conceived the spatial modelling framework. V.R.K., N.Y., C.S., L.R. and E.P. generated single-cell data. L.B. with help from J.V.H. and F.V.N. generated CITE-seq data. A.J.R., B.T.W. and R.T.B. developed the IBEX panel and performed imaging. E.T., M.D., C.S., E.D. and M. Patel with input from N.Y. and V.R.K. performed Visium data generation. L.B., J.V.H., A.Y.K., M.B., J.E., E.S., F.P., F.D.R., D.C., X.H., M.A.H., R.A.B. and E.G.D. collected and preserved thymus samples. C.T. generated RNAscope data. S.P., N.-J.C. and O.A.B. generated RareCyte data for validations. N.Y., O.A., K.P., S.K., L.F. and K.T. developed software. A.V.P., T.P., K.P., V.K. and M. Prete preprocessed the data and provided computational support. N.Y., V.R.K., L.B., C.S., E.D. and S.K. analysed the data and interpreted the results. N.Y., L.B., V.R.K., A.J.R., S.A.T., R.N.G., T.T., V.U. and J.C.M. wrote the paper. L.D.N., S.K., M.B., J.V.H. and F.P. provided feedback and biological interpretation. S.A.T., T.T., J.C.M., A.J.R., R.N.G. and L.D.N. supervised the study.

**Competing interests** J.C.M. has been an employee of Genentech since September 2022. S.A.T. is a scientific advisory board member of ForeSite Labs, OMass Therapeutics, Qiagen, a co-founder and equity holder of TransitionBio and EnsoCell Therapeutics, a non-executive director of 10x Genomics and a part-time employee of GlaxoSmithKline. The other authors declare no competing interests.

**Additional information**
**Correspondence and requests for materials** should be addressed to Ronald N. Germain, Andrea J. Radtke, John C. Marioni, Tom Taghon or Sarah A. Teichmann.

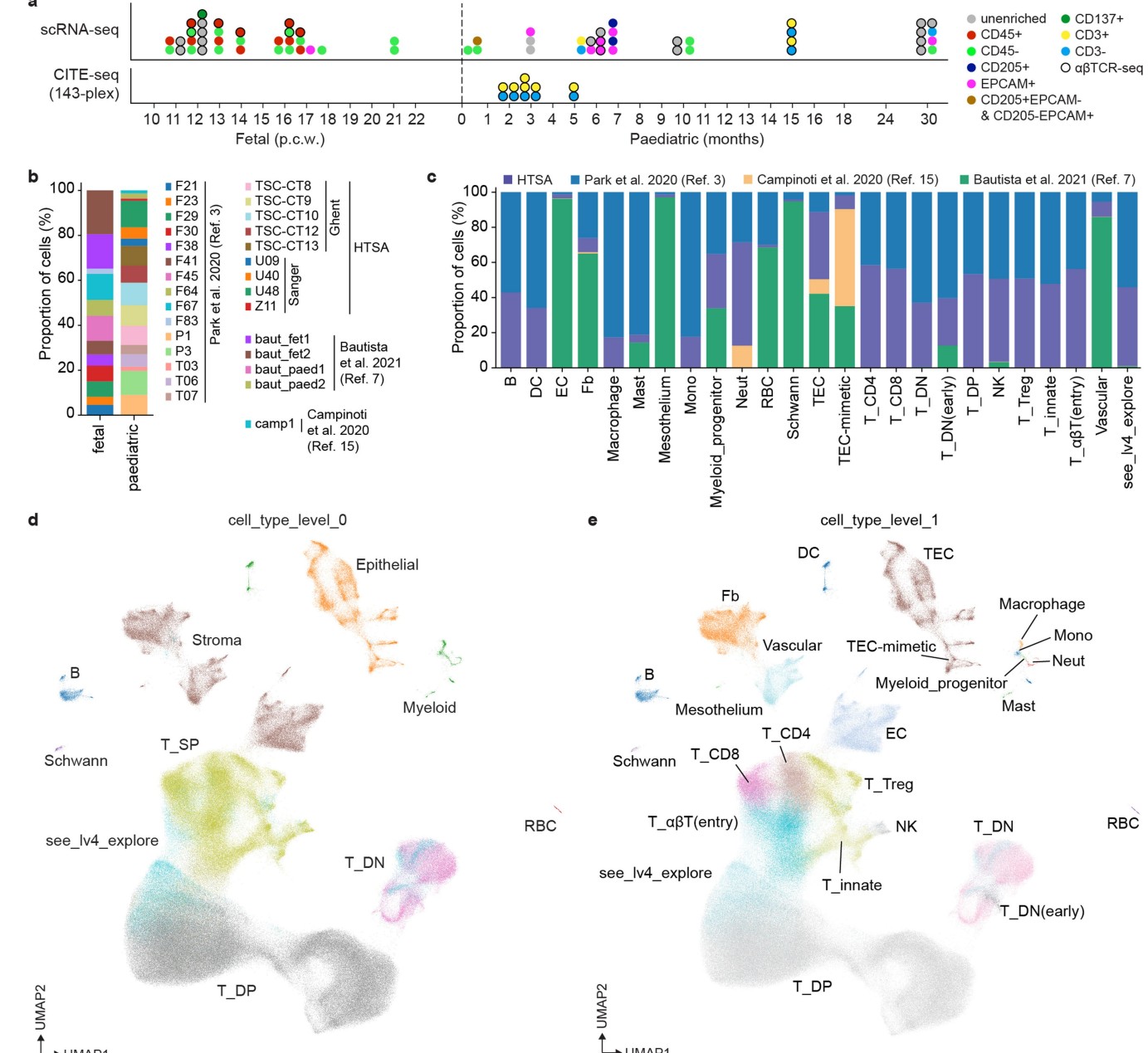

**Extended Data Fig. 1 | Composition of fetal and paediatric scRNA-seq data.**
**a.** Sample enrichment strategy for dissociated datasets as indicated by colour. Each dot represents a sample and stacked dots within a technology panel represent samples from the same donor. For samples marked by a black circle, αβTCR-seq was carried out. See Fig. 1c for sample source. **b.** Relative cell contribution to the dissociated dataset per donor, split by age group. Sample origin is indicated for all donors. $n = 12$ fetal donors, $n = 17$ paediatric donors. HTSA: Human Thymus Spatial Atlas. **c.** Relative contribution of published and newly generated scRNA-seq datasets by broad cell type (cell type level 1). $n = 29$ donors. **d.** UMAP embedding of the full, integrated scRNA-seq dataset with

annotations of the major cell lineages (cell type level 0). **e.** UMAP embedding of the full, integrated scRNA-seq dataset with more detailed lineage annotations (cell type level 1). See Supplementary Notes 3 and 4 for complete annotation of T lineage, hematopoietic and stromal cells. "See_lv4_explore" refers to cells which could not confidently be assigned to a unique cell type based on literature or cell markers but can be explored in the provided AnnData object (Data availability). DC, dendritic cell; EC, endothelial cell; Fb, fibroblast; Mast, mast cell; Mono, monocyte; Neut, neutrophil; RBC, red blood cell; DN, double negative thymocyte; DP, double positive thymocyte; NK, natural killer cell.

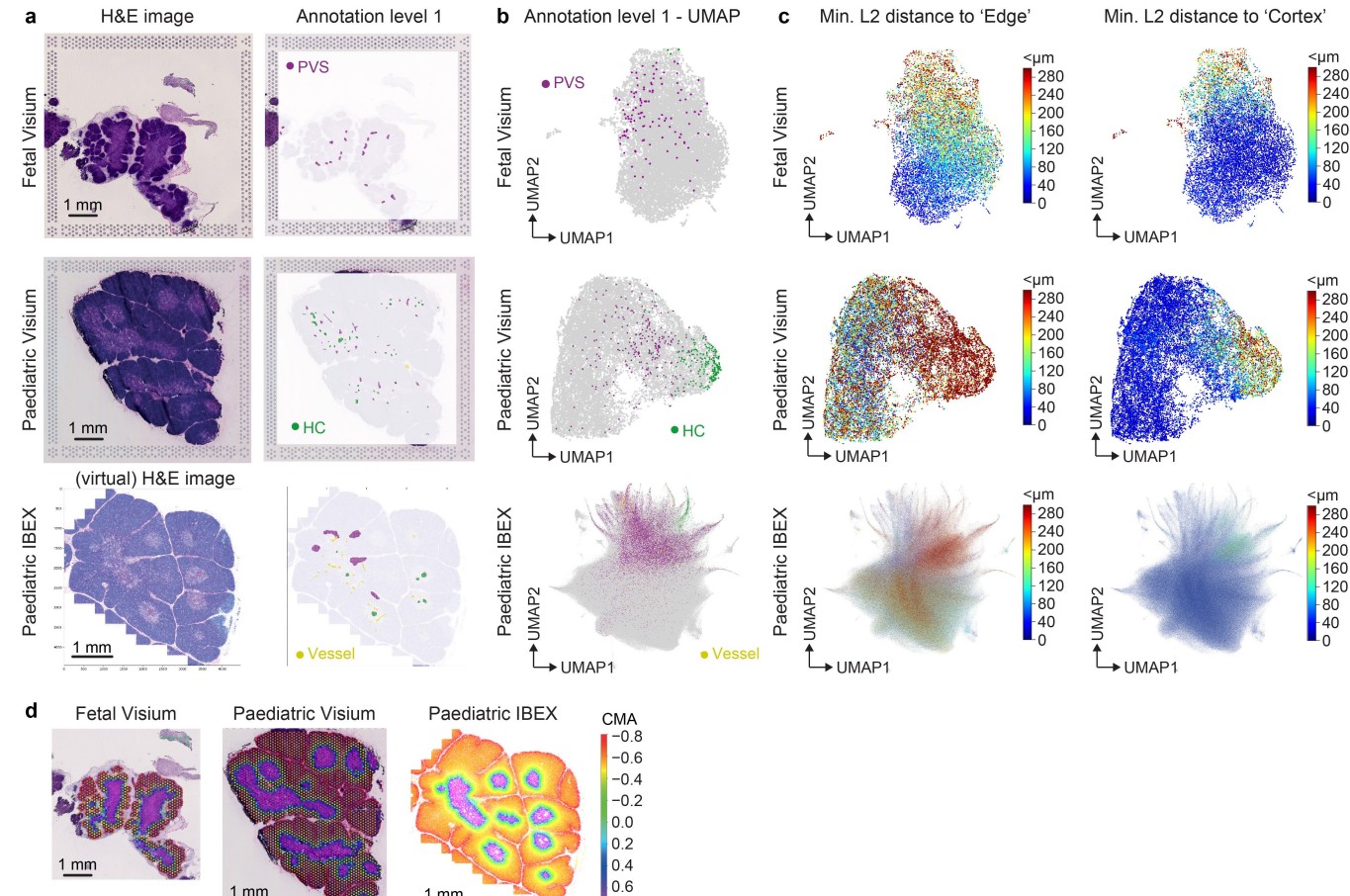

**Extended Data Fig. 2 | Annotation of secondary structures in the thymus and distance measurements to region boundaries. a**. Representative H&E sections (left) of fetal (p.c.w. 15) and paediatric (3 months) Visium data, and IBEX virtual H&E (7 days old). Corresponding discrete annotation (annotation level 1, right) curated with TissueTag for Hassall's corpuscles (HC), perivascular space (PVS), and additional small vessels. **b**. UMAP embedding of integrated samples for the three spatial datasets coloured by annotation level 1. **c**. UMAP embedding of integrated samples for the three spatial datasets coloured by min. L2 distances of each spot/cell to the cortex (right) and to the capsule/septum ("Edge", left) demonstrating distinct spatial variance. **d**. CMA projected to Visium spot data and IBEX single cells on the same sections as shown in a.

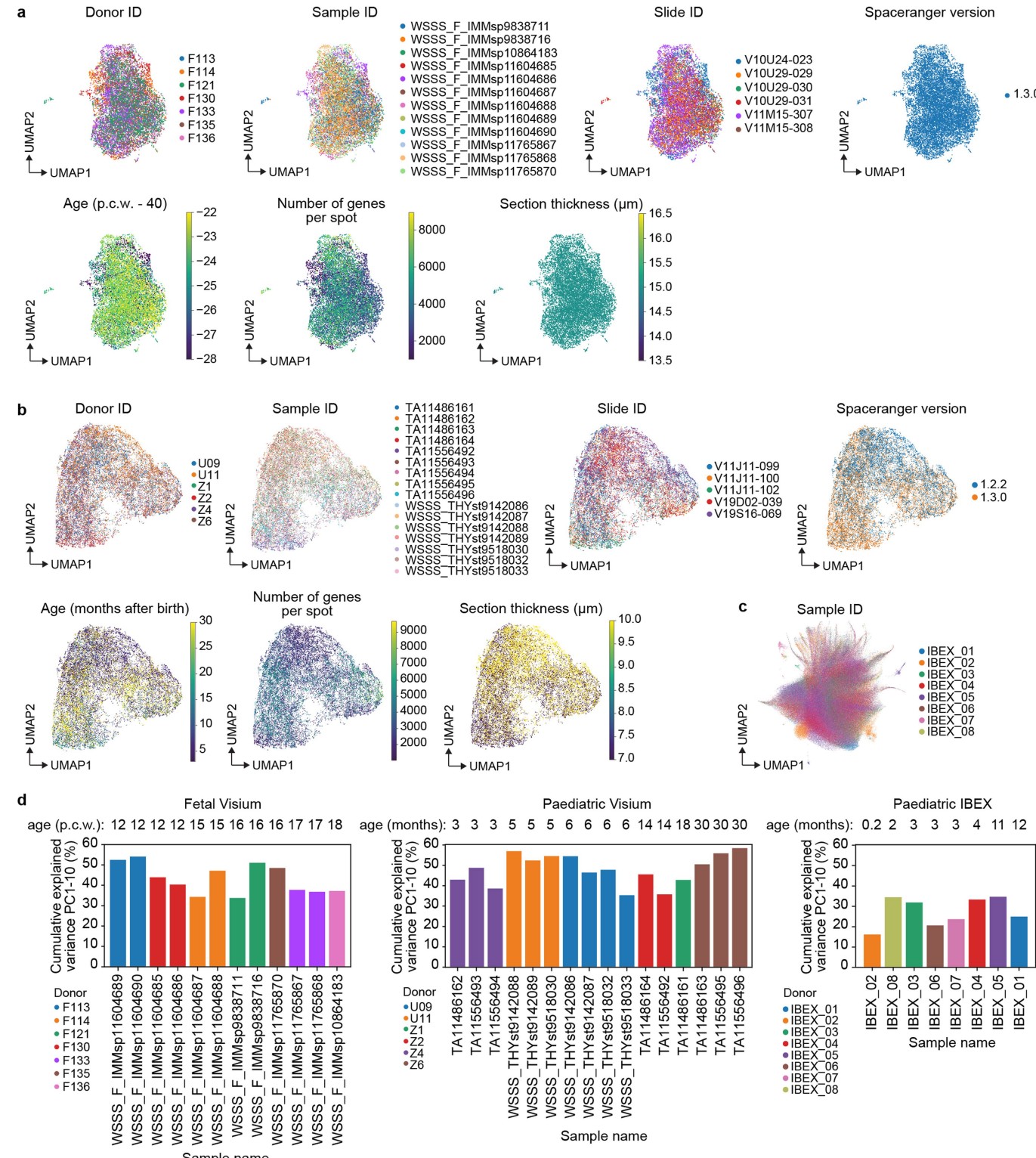

**Extended Data Fig. 3 | Spatial sample composition and variance. a-b**. UMAP embeddings of integrated Visium spots coloured by donor, sample (Visium capture region), Visium slide, SpaceRanger version, age, number of genes captured, and section thickness. **a**. Integrated UMAP embeddings for fetal Visium samples. Age is indicated as post-conception weeks (p.c.w.) – 40.

**b**. Integrated UMAP embeddings for paediatric Visium samples. **c**. UMAP embedding of integrated IBEX single nuclei data coloured by sample (one sample per donor). **d**. Cumulative explained variance of the ten first PCA components correlated to the spot CMA level for fetal and paediatric Visium, and paediatric IBEX samples. Samples are sorted by age from left to right and coloured by donor.

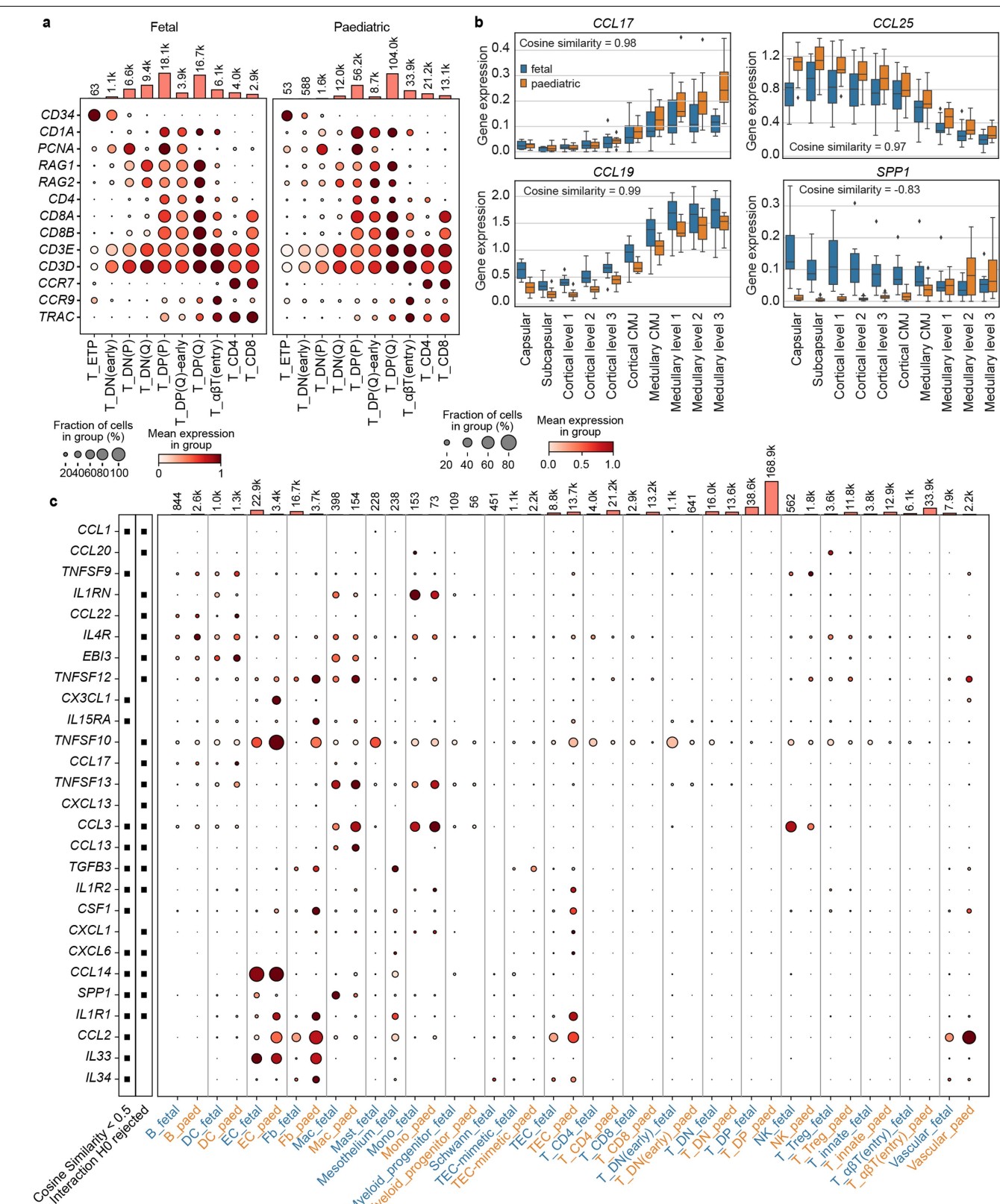

**Extended Data Fig. 4 | T cell markers and cytokine/chemokine expression profiles in fetal and paediatric single-cell and spatial datasets. a**. Dotplot showing expression of key cell type markers in the αβ T lineage differentiation stages depicted in Fig. 3d. Cells are arranged from most immature (left) to mature (right). Bar graphs indicate the total number of cells per subset. **b**. Boxplots showing the mean expression of selected chemokine genes in each CMA bin across different fetal vs. paediatric samples. Box boundary extends from the first to the third quartile of distribution with median in between, whiskers indicate min and max ranges with the exception of outliers (outside an inter-quartile range (IQR) of 1.5, indicated by diamonds). Cosine similarity between the mean fetal and paediatric expression values is indicated. *n* = 16 paediatric Visium samples, *n* = 12 fetal Visium samples **c**. Dotplot for chemokines/cytokines with differential distribution across the CMA bins for fetal vs. paediatric tissue as indicated either by low cosine similarity and/or a significant interaction effect. Bars show total number of cells per subset.

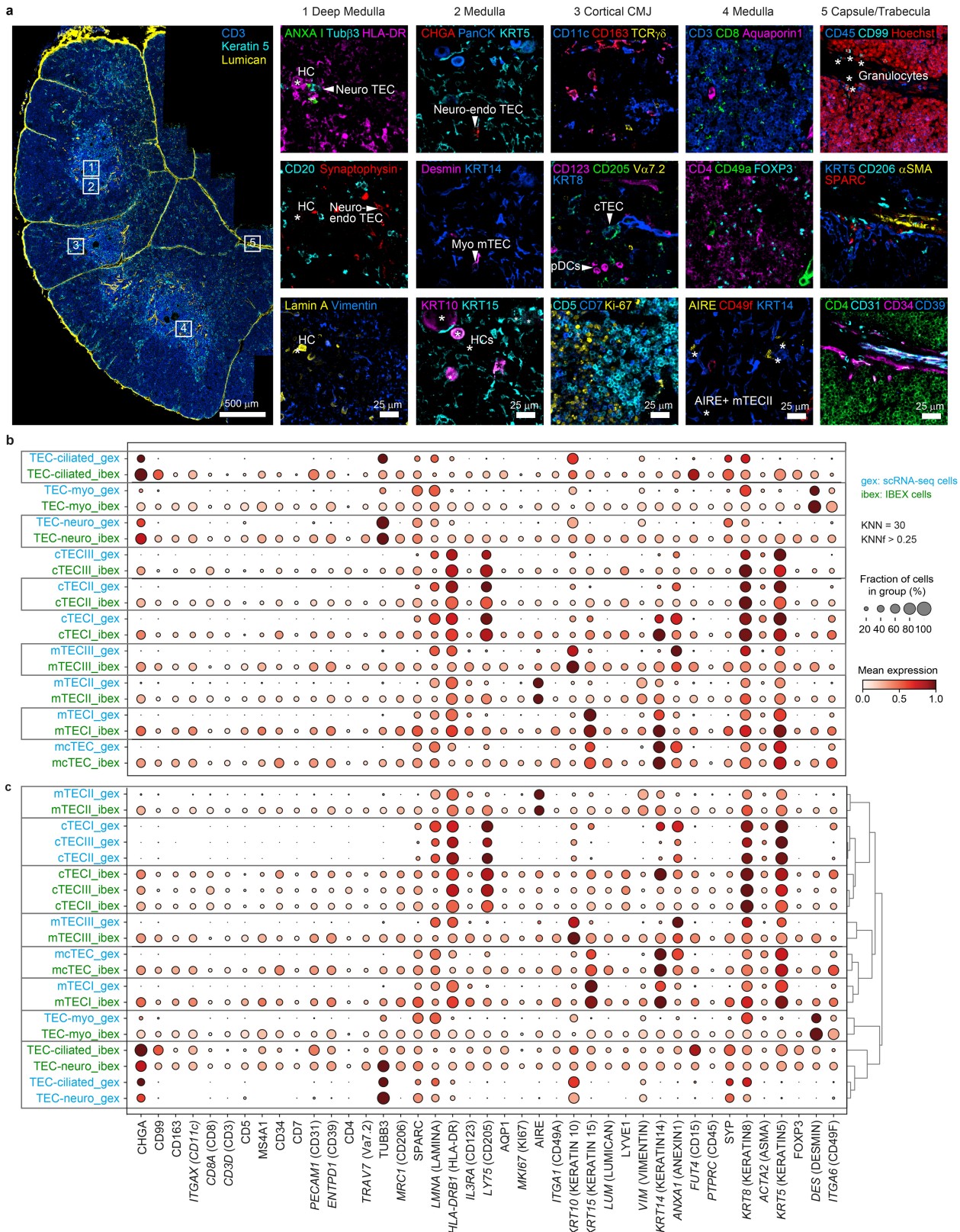

**Extended Data Fig. 5 | Identification of stromal and TEC subtypes using IBEX imaging. a**. IBEX confocal images from 2 month-old female thymus (IBEX_08) showing anatomical structures and cell types defined by 44-plex antibody panel. Images are representative of 8 samples/donors. Not shown: CD15 and LYVE-1. Large overview image shows a typical region of interest captured in each IBEX experiment (2-3 lobules). ANXA1, Annexin I; CHGA, Chromogranin A; Pan-CK, Pan-Cytokeratin; KRT, Keratin. **b**. Dot plot showing expression of proteins in IBEX data and of corresponding genes in scRNA-seq data for annotated TEC subsets. Depicted cell types represent cells in paediatric scRNA-seq data ("_gex") and the corresponding cell types predicted in IBEX data based on KNN matching ("_ibex"). Expression was normalized per row. Boxes highlight corresponding cell types in the two datasets. KKNf is the fraction of mapped target KNN cells that come from the same cell type. **c**. Same as in b. but with rows clustered by mutual similarity dendrogram linkage to show similarity level between cell types within and across datasets. Boxes highlight cell types with highest similarity according to dendrogram.

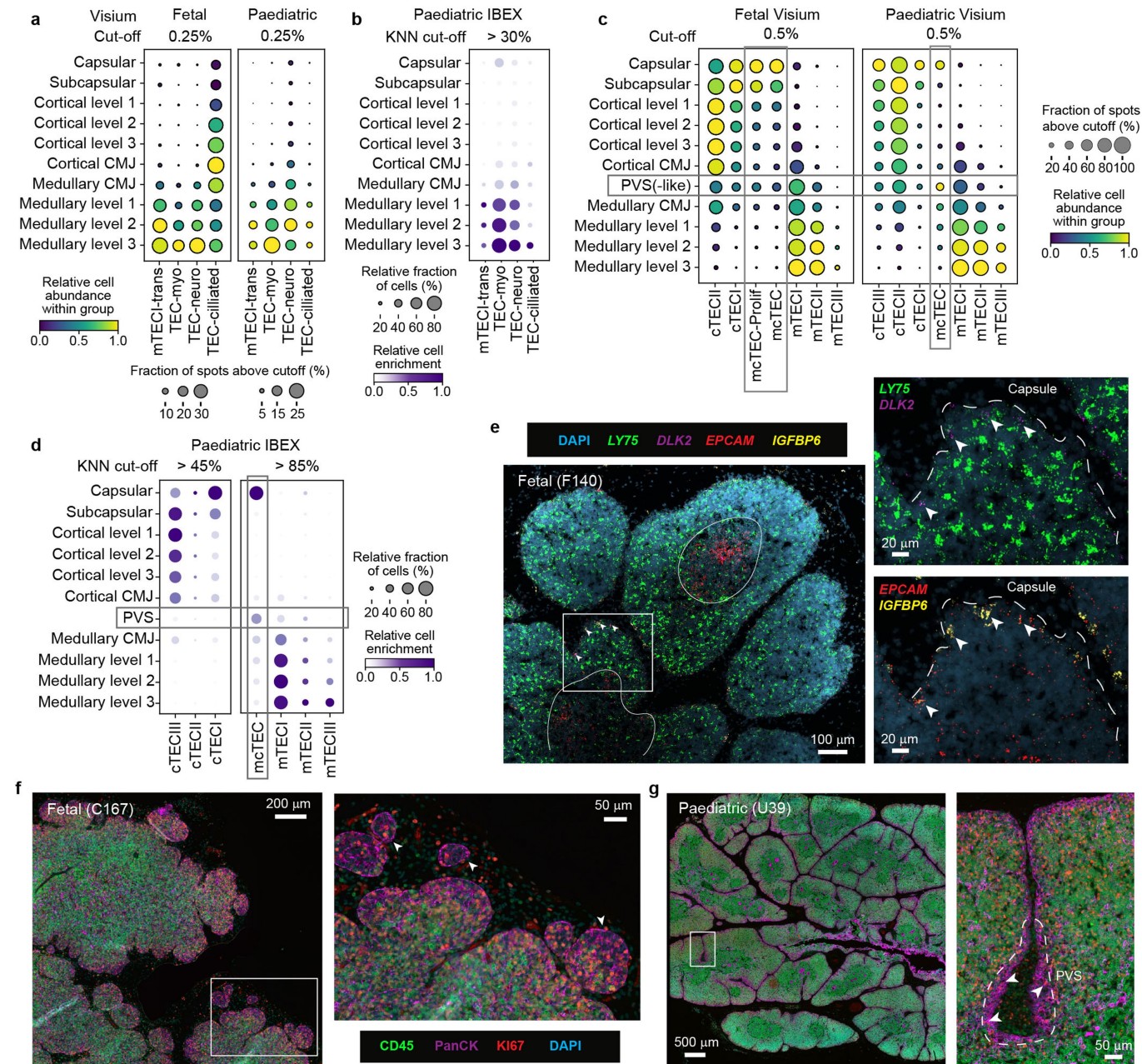

**Extended Data Fig. 6 | Distribution of TEC subtypes in the fetal and paediatric thymus. a**. Dot plots showing the relative cell abundance of specialized TECs across the CMA bins in fetal and paediatric deconvolved Visium datasets. In all Visium dot plots, cutoff indicates the minimum proportion of the respective cell type in a Visium spot for the spot to be included. **b**. Dotplots showing relative cell abundance of specialized TECs across the CMA bins in paediatric IBEX KNN-mapped single nuclei datasets. For IBEX datasets KNN cutoff indicates the minimum percentage of KNNs that corresponded to the eventually assigned majority cell type agreement (see Methods). **c**. Relative cell distribution of TECs in CMA bins and spots associated with perivascular space (PVS) based on deconvolved Visium data. Boxes highlight mcTECs and PVS annotations. Note that proliferating mcTECs were only found in fetal thymus and cTECIII was exclusively detected in paediatric data. **d**. Relative cell distribution of TECs in CMA bins as well as PVS region based on paediatric IBEX KNN-mapped single nuclei datasets. Boxes highlight mcTECs and PVS annotations. **e**. 4-plex

RNAscope staining of a fetal thymus tissue section (p.c.w. 13) for mcTECs (*DLK2*, *IGFBP6*), cTECs (*LY75*) and mTECs (*EPCAM*). DAPI was used to identify nuclei. White frame in the left image indicates magnified regions shown on the right. Lines in the left image indicate the CMJ, dashed lines in the right images highlight the capsule. Arrows indicate capsular mcTECs. Images are representative of four independent replicates. **f-g**. RareCyte protein staining of fetal and paediatric thymus sections with Ki-67, Pan-Cytokeratin (PanCK), and CD45 antibodies and DAPI. White frame in the left image indicates magnified regions on the right. **f**. Fetal thymus sample (p.c.w. 12). Arrows highlight a subcapsular niche with Ki-67+ non-lymphoid (CD45−) epithelial (PanCK+) cells. Images are representative of a total of 10 replicates from 6 donors. **g**. Paediatric thymus sample (1 month old). Arrows highlight epithelial (PanCK+) cells in a lymphocyte-free (CD45−) region in the PVS, which show little proliferation (Ki-67+/−). Images are representative of a total of 3 replicates from 3 donors.

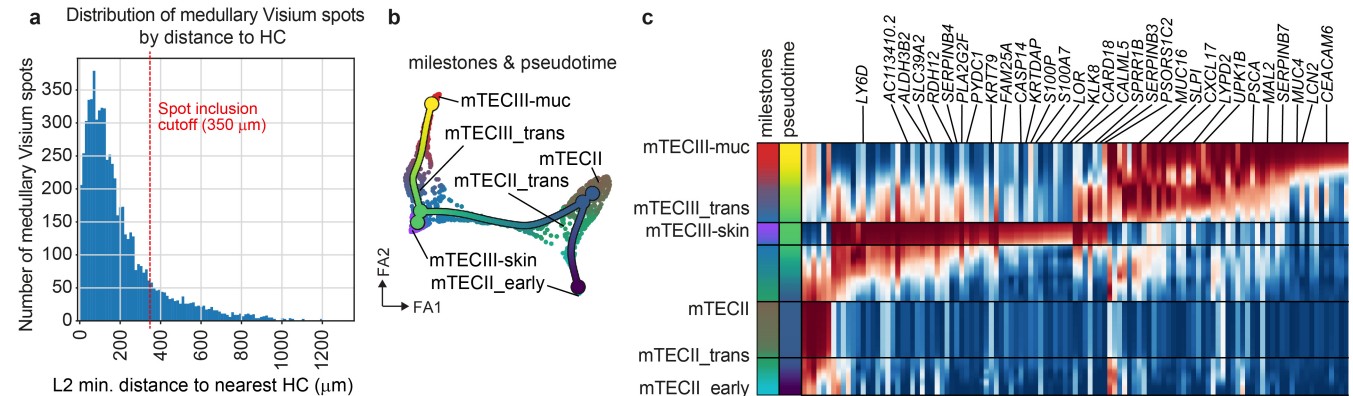

**Extended Data Fig. 7 | Expression of mucosa- and skin-related genes in differentiated mTECIII subtypes. a**. Histogram showing the number of Visium spots with medullary annotation by their distance to the nearest HCs. The red dashed line indicates the maximum distance cutoff for Visium spots to be included. Note that around 90% of medullary Visium spots fall within this window. **b**. Multiscale diffusion space embedding of the mTECII/mTECIII populations generated using Palantir (coloured according to cell populations) with trajectory generated using scFates overlaid on top (colour indicates pseudotime stretching from dark violet to yellow). **c**. Heatmap showcasing differentially expressed genes between three different branches of the mTECII/TECIII trajectory, namely mTECII, mTECIII-skin and mTECIII-muc. See Methods for details on derivation of DE genes and source data for p-values. Specialization genes (SGs) are labelled.

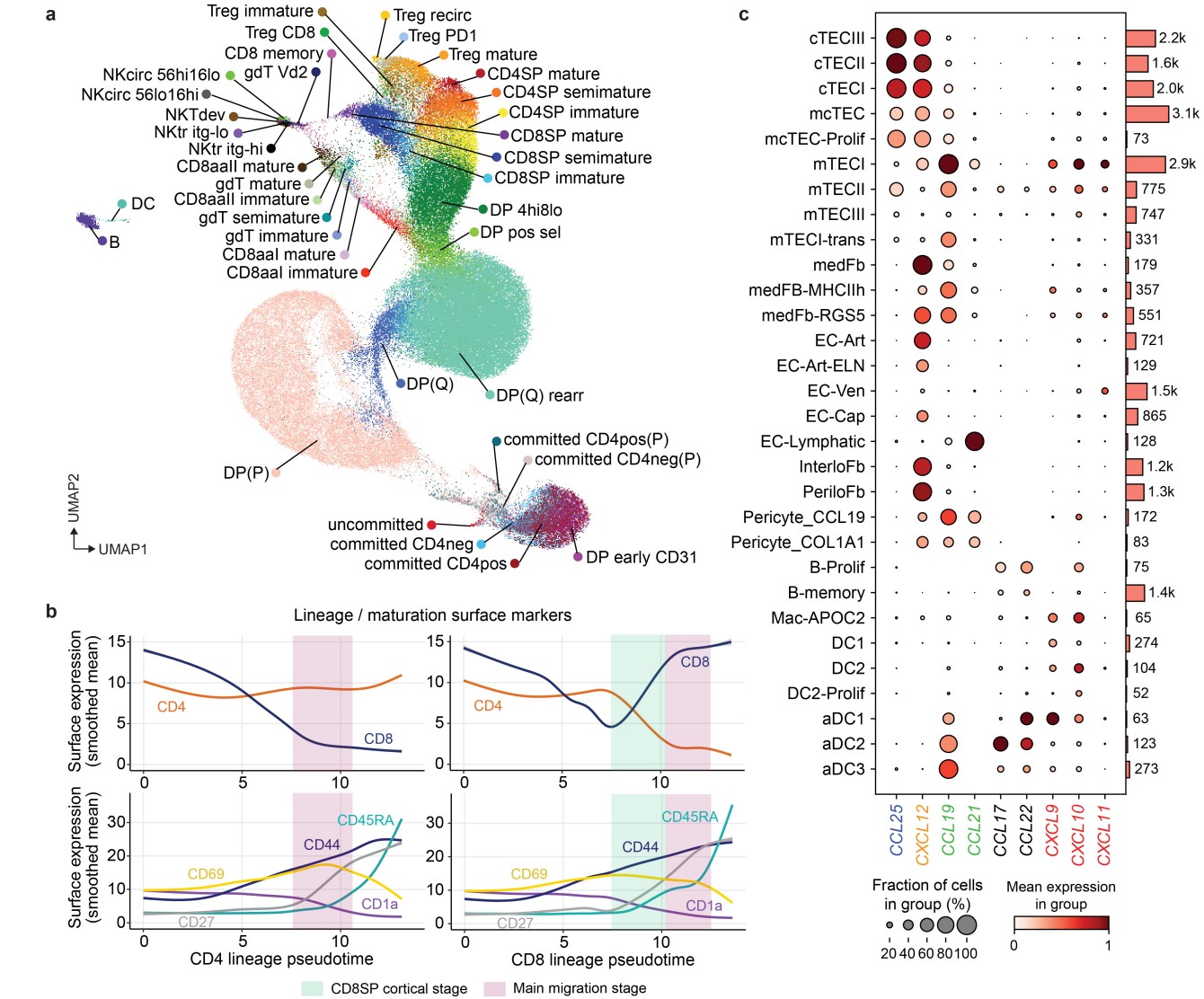

**Extended Data Fig. 8 | Annotation and expression profiling of developing human thymocytes using CITE-seq. a**. UMAP embedding of the full paediatric CITE-seq dataset with high-resolution annotation of T cell maturation stages as well as several non-T cell subtypes. Supplementary Note 6 provides additional details on CITE-seq-derived annotation of T lineage cells. DP, double positive; (P), proliferating; (Q), quiescent; rearr, TCR-rearranging; pos sel, positive selected; SP, single positive; recirc, recirculating; tr, tissue resident; circ, circulating; itg, integrin. **b**. Surface expression levels of lineage and maturation markers along predicted pseudotimes for CD4 (left) and CD8 lineage (right).

Line plots represent smoothed means ± s.e.m. of the expression levels in cells shown in Fig. 6c. Shaded boxes indicate the migration window and CD8SP cortical stage identified in Fig. 6d–f. **c**. Dot plot depicting expression of relevant chemokines in hematopoietic and stromal cells according to the paediatric scRNA-seq dataset. Colour coding indicates corresponding ligands to the receptors shown in Fig. 6g. Bar graphs indicate total number of cells per cell type. medFB, medullary fibroblast; EC, endothelial cell; Art, arterial; Ven, venous; Cap, capillary; Mac, macrophage; DC, dendritic cell.

# Reporting Summary

## Statistics

For all statistical analyses, confirm that the following items are present in the figure legend, table legend, main text, or Methods section.

| n/a | Confirmed | |
|---|---|---|
| ☐ | ☒ | The exact sample size (*n*) for each experimental group/condition, given as a discrete number and unit of measurement |
| ☐ | ☒ | A statement on whether measurements were taken from distinct samples or whether the same sample was measured repeatedly |
| ☐ | ☒ | The statistical test(s) used AND whether they are one- or two-sided *Only common tests should be described solely by name; describe more complex techniques in the Methods section.* |
| ☐ | ☒ | A description of all covariates tested |
| ☐ | ☒ | A description of any assumptions or corrections, such as tests of normality and adjustment for multiple comparisons |
| ☐ | ☒ | A full description of the statistical parameters including central tendency (e.g. means) or other basic estimates (e.g. regression coefficient) AND variation (e.g. standard deviation) or associated estimates of uncertainty (e.g. confidence intervals) |
| ☐ | ☒ | For null hypothesis testing, the test statistic (e.g. *F*, *t*, *r*) with confidence intervals, effect sizes, degrees of freedom and *P* value noted *Give P values as exact values whenever suitable.* |
| ☒ | ☐ | For Bayesian analysis, information on the choice of priors and Markov chain Monte Carlo settings |
| ☒ | ☐ | For hierarchical and complex designs, identification of the appropriate level for tests and full reporting of outcomes |
| ☒ | ☐ | Estimates of effect sizes (e.g. Cohen's *d*, Pearson's *r*), indicating how they were calculated |

*Our web collection on statistics for biologists contains articles on many of the points above.*

## Software and code

Policy information about availability of computer code

| Data collection | The following packages were used to collect the publicly available data: srapath v2.11.0, bamtofastq v1.3.2, entrez-direct v15.6, fastq-dump v2.11.0, samtools v1.12 and iRODs v4.2.7. |
|---|---|
| Data analysis | All code scripts and notebooks used in the study are publicly available here: https://github.com/Teichlab/thymus_spatial_atlas. Code for the TissueTag package is available in the following GitHub repository: https://github.com/Teichlab/TissueTag. |

We also list the major libraries used below:
- Data mapping and pre-processing: STARsolo with STAR v2.7.9a, Cell Ranger v6.1.1 and v7.0.0 ; soup removal: CellBender v0.1.0;
- Python libraries:
general: pandas v2.2.2, numpy v1.26.4, scipy v1.13.0, scikit-learn v1.4.2, seaborn v0.13.2, matplotlib v3.8.4; single-cell processing: scanpy v1.9.1 with anndata v0.10.7, doublet removal: scrublet v0.2.3, automatic cell annotation: celltypist v1.6.2, data integration and embedding: scvi-tools 0.19.0; trajectory inference: scFates v1.0.7 and Palantir v1.3.3; knn mapping: https://github.com/Teichlab/iss_patcher/tree/main.
- R libraries:
general: Matrix (v1.6-4), matrixStats (v1.2.0), dplyr (v1.1.4), tidyr (v1.3.1), reshape2 (v1.4.4), BiocNeighbours (v1.20.2), BiocParallel (v1.36.0), stringr (1.5.1), reticulate (v1.35.0), and sceasy (v0.0.7); visualisation: ggplot2 (v3.5.0), ggrastr (v1.0.2), ggridges (v0.5.6) and RColorBrewer (v1.1-3); single-cell processing: Seurat (v4.3.0), SeuratObject (v4.1.4), SeuratDisk (v0.0.0.9021), SingleCellExperiment (v1.24.0), sceasy (v0.0.7); CITEseq denoising: dsb v1.0.3; batch correction: Batchelor v1.18.1; marker identification: singleCellHaystack v1.0.0, trajectory inference: Slingshot v2.6.0; fate mapping: STEMNET v0.1.
- TCR data processing: https://github.com/zktuong/dandelion v0.3.1.
- Spatial data processing: SpaceRanger v1.2.2 and v1.3.0, cell2location v0.1.3.
- Image processing: https://github.com/Teichlab/TissueTag or tissue-tag 0.1.1 uses bokeh v3.4.1, scikit-image v0.22.0, cellpose v2.1.1.IBEX

```
data processing: LAS X Navigator v3.5.7.23225, Fiji 1.54j, IMARIS. RareCyte: Artemis v4.
- FACS data acquisition and analysis: FACSDiva v8.0.2, FlowJo v10.8.2 and FCS Express v7.18.0025.
```

For manuscripts utilizing custom algorithms or software that are central to the research but not yet described in published literature, software must be made available to editors and reviewers. We strongly encourage code deposition in a community repository (e.g. GitHub). See the Nature Portfolio guidelines for submitting code & software for further information.

# Data

Policy information about availability of data

All manuscripts must include a data availability statement. This statement should provide the following information, where applicable:
- Accession codes, unique identifiers, or web links for publicly available datasets
- A description of any restrictions on data availability
- For clinical datasets or third party data, please ensure that the statement adheres to our policy

The annotated fetal and paediatric integrated scRNA-seq atlas and Visium objects for this study can be explored online https://cellxgene.cziscience.com/collections/fc19ae6c-d7c1-4dce-b703-62c5d52061b4.
Paediatric CITE-seq data can be visualised and explored using acustom ShinyApp (https://ccgg.ugent.be/shiny/htsa_thymocyte_citeseq/). Sequencing data for the newly generated libraries for scRNA-seq and Visium data were uploaded to ENA under accession code PRJEB77091. Several samples were obtained under consent agreements that require data release with managed access, which is why these were deposited at EGA under accession code EGAD00001015384.
Imaging data for Visium samples were deposited at BioImage Archive under accession number S-BIAD1257. CITE-seq data were uploaded to the GEO under accession number GSE271304. Imaging data generated for this study, including IBEX, RNAscope and RareCyte data, were deposited at the BioImage Archive under the accession number S-BIAD1257. Publicly available datasets were downloaded from the following sources: ref. 3 (ArrayExpress: E-MTAB-8581; GEO: GSE206710); ref. 15 (GEO: GSE159745); ref. 7 (GEO: GSE147520); and ref. 16 (ArrayExpress:E-MTAB-11341). All accession codes are also listed in Supplementary Tables 1 and 2. Source data are provided with this paper.

# Research involving human participants, their data, or biological material

Policy information about studies with human participants or human data. See also policy information about sex, gender (identity/presentation), and sexual orientation and race, ethnicity and racism.

| Reporting on sex and gender | 1) scRNAseq: 6 male and 7 female fetal samples and 10 male and 8 female paediatric samples were used to generate the reference thymus scRNA-seq atlas.<br>2) Visium spatial transcriptomics: samples from 7 fetal donors were used (sex wasn't recorded) as well as 2 male and 4 female donors.<br>3) IBEX: samples from 4 male and 4 female patients were used.<br>Sex comparison wasn't performed. |
|---|---|
| Reporting on race, ethnicity, or other socially relevant groupings | Data on race, ethnicity and socioeconomics status wasn't collected. |
| Population characteristics | For the samples used for to put together reference single-cell and Visium spatial transcriptomics atlas age was recorded and used as covariate during the sample integrations.<br>1) scRNAseq: fetal donors ranged in age from 11 pcw to 21 pcw, while paediatric patients ranged from 6 days until 2.5 yrs.<br>2) Visium: fetal donors ranged in age from 12 until 18 pcw, while paediatric patients ranged in age from 3 months to 2.5 yrs.<br>3) IBEX: patients ranged in age from 7 days to 1 year. |
| Recruitment | Paediatric thymi samples were obtained from the patients undergoing corrective cardiac surgeries. Fetal samples were donated voluntarily by women who have had a termination of pregnancy from clinics collaborating with the Human Developmental Biology Resource, UK (https://www.hdbr.org/). |
| Ethics oversight | Wellcome Sanger Institute, UK: Paediatric thymi samples were provided by Newcastle University collected under REC approved study 18/EM/0314 and Great Ormond Street Hospital under REC approved study 07/Q0508/43. The human embryonic and fetal material was provided by the joint MRC / Wellcome Trust (Grant # MR/006237/1) Human Developmental Biology Resource (http://www.hdbr.org).<br><br>Ghent University, Belgium: Paediatric thymi samples were obtained according to and used with the approval of the Medical Ethical Commission of Ghent University Hospital, Belgium (EC/2019-0826) through the hematopoietic cell biobank (EC-Bio/1-2018).<br><br>NIH, US: Paediatric thymi samples processed at the NIH were obtained (under NIAID MTA 2016-250) from the pathology department of the Children's National Medical Center in Washington, DC, following cardiothoracic surgery from children with congenital heart disease. Use of these thymus samples for this study was determined to be exempt from review by the NIH Institutional Review Board in accordance with the guidelines issued by the Office of Human Research Protections.<br><br>Written informed consent from the donors or their families was obtained for the samples obtained at all the sites. |

Note that full information on the approval of the study protocol must also be provided in the manuscript.

# Field-specific reporting

Please select the one below that is the best fit for your research. If you are not sure, read the appropriate sections before making your selection.

☒ Life sciences ☐ Behavioural & social sciences ☐ Ecological, evolutionary & environmental sciences

For a reference copy of the document with all sections, see nature.com/documents/nr-reporting-summary-flat.pdf

# Life sciences study design

All studies must disclose on these points even when the disclosure is negative.

| | |
|---|---|
| Sample size | No sample size calculation was performed, sample size was determined by the number of integrated datasets and time availability to collect new ones.<br>scRNA-seq: 13 fetal donors and 18 paediatric donors;<br>Visium: 7 fetal donors and 6 paediatric donors;<br>IBEX: 8 paediatric patients. |
| Data exclusions | No samples were excluded from the study. |
| Replication | Validation using alternative approaches was performed to confirm the reproducibility of the findings, including using RNAscope (n=4 fetal donors) and RareCyte experiments (6 fetal donors and 3 paediatric). |
| Randomization | Randomization wasn't performed as it is not applicable to this project. Samples were allocated into fetal and pediatric age groups based on their origin before or after birth. |
| Blinding | Blinding for analysis of spatial data wasn't possible due to large size and morphological differences between fetal and paediatric thymus. |

# Reporting for specific materials, systems and methods

We require information from authors about some types of materials, experimental systems and methods used in many studies. Here, indicate whether each material, system or method listed is relevant to your study. If you are not sure if a list item applies to your research, read the appropriate section before selecting a response.

## Materials & experimental systems

| n/a | Involved in the study |
|---|---|
| ☐ | ☒ Antibodies |
| ☒ | ☐ Eukaryotic cell lines |
| ☒ | ☐ Palaeontology and archaeology |
| ☒ | ☐ Animals and other organisms |
| ☒ | ☐ Clinical data |
| ☒ | ☐ Dual use research of concern |
| ☒ | ☐ Plants |

## Methods

| n/a | Involved in the study |
|---|---|
| ☒ | ☐ ChIP-seq |
| ☐ | ☒ Flow cytometry |
| ☒ | ☐ MRI-based neuroimaging |

## Antibodies

| | |
|---|---|
| Antibodies used | Flow-cytometry antibodies:<br>1. EPCAM (anti-CD326 PE clone 9C4, Biolegend #324206, dilution 1:50), https://www.biolegend.com/en-gb/products/pe-anti-human-cd326-epcam-antibody-3757, stroma sorting.<br>2. CD45 (CD45 BV785, HI30 Clone (mouse), Biolegend #304048, dilution 1:50), https://www.biolegend.com/en-gb/products/brilliant-violet-785-anti-human-cd45-antibody-9325, stroma sorting.<br>3. DEC205 (APC anti-human CD205 (DEC-205), Biolegend #342207, dilution 1:50),https://www.biolegend.com/de-at/products/apc-anti-human-cd205-dec-205-antibody-5973, stroma sorting.<br>4. CD3 (FITC anti-human CD3, OKT3 clone (mouse), Biolegend #317306, dilution 1:50), https://www.biolegend.com/en-gb/products/fitc-anti-human-cd3-antibody-3644, stroma sorting.<br>5. CD3 (anti-CD3-PE clone SK7, Biolegend, 344805, 2ug/ml concentration), https://www.biolegend.com/en-gb/products/pe-anti-human-cd3-antibody-13257, T cell sorting.<br>CITEseq antibodies: The full list of TotalSeq-C antibodies used for CITE-seq is available in Supplementary Table 6 together with staining concentrations if available.<br>IBEX antibodies: All information relating to antibodies used for IBEX imaging, including dilutions, can be found in Supplementary Table 3.<br>RareCyte antibodies: Antibodies used for multiplex RareCyte staining were purchased from RareCyte: https://rarecyte.com/ and detailed in Supplementary Table 4 together with standard dilution (1:200). |
| Validation | Flow-cytometry antibodies: We have listed detailed antibody information as well as dilutions used in the field above. Please see references contained in the provided links to find available validations. |

The IBEX antibody panel was put together after thorough testing of a large list of antibody candidates. All antibodies use in the study have been used in our prior publications or underwent validation on human tissue individually to ensure expected distributions and co-localisation where appropriate. Antibodies used for IBEX experiments have been deposited in the IBEX Imaging Community with reference imaging data: https://ibeximagingcommunity.github.io/ibex_imaging_knowledge_base/ [ibeximagingcommunity.github.io]. A thymus Organ Mapping Antibody Panel is available through the HuBMAP Organ Mapping Antibody Panel (OMAP) initiative. See OMAP-17 here: https://humanatlas.io/omap [humanatlas.io].

CITE-seq antibodies: TotalSeq-C Human Universal Cocktail has been validated by manufacturer (BioLegend, see https://www.biolegend.com/en-gb/products/totalseq-c-human-universal-cocktail-v1-0-19736). 13 additional TotalSeq-C antibodies were titrated according to the manufacturer's instructions and introduced into CITE-seq panel.

RareCyte antibodies: Antibodies used for multiplex RareCyte staining were purchased from RareCyte (immuno-oncology panel), see https://rarecyte.com/orionpanels/ and validated by the company.

# Flow Cytometry

## Plots

Confirm that:

☒ The axis labels state the marker and fluorochrome used (e.g. CD4-FITC).

☒ The axis scales are clearly visible. Include numbers along axes only for bottom left plot of group (a 'group' is an analysis of identical markers).

☒ All plots are contour plots with outliers or pseudocolor plots.

☒ A numerical value for number of cells or percentage (with statistics) is provided.

## Methodology

| | |
|---|---|
| Sample preparation | 1. Stroma cell sorting: Briefly, tissue was finely minced and cell dissociation was performed using a mixture of liberaseTH (Roche, 05401135001) and DNAse-I (Roche, 4716728001) for ~30 mins in two rounds. Digested tissue was filtered through a 70 µm strainer and digestion was stopped with 2% FBS in RPMI media. Next, red blood cell lysis was performed on a cell pellet using the RBC lysis buffer from eBioscience™ (00-4333-57), after which samples were washed and counted. To perform FACS sorting, cells were resuspended in the FACS buffer (0.5% FBS and 2mM EDTA in PBS), underwent blocking in TruStainFcX (422302, Biolegend) for 10 mins and were stained with a mixture of EPCAM (anti-CD326 PE clone 9C4, Biolegend #324206), CD45 (CD45 BV785, HI30 Clone (mouse), Biolegend #304048), DEC205 (APC a 2nti-human CD205 (DEC-205), Biolegend #342207) and CD3 (FITC anti-human CD3, OKT3 clone (mouse), Biolegend #317306 ) antibodies and DAPI for 30 mins. Upon staining, cells were washed and analysed using the Sony SH800 or Sony MA900 sorters with 130 µm nozzle. 2. T cell enrichment for CITEseq: Cells were thawed slowly by gradually adding 15 volumes of pre-warmed IMDM media and pelleted at 1700 rpm for 6 min at 4°C. After resuspending in PBS, cells were passed through a 70 µm strainer to remove clumps. Enrichment for viable cells was achieved using a magnetic bead-based dead cell removal kit (Miltenyi, 130-090-101). For this, cells were pelleted as before, washed with 1X Binding Buffer (part of kit, prepared with sterile distilled water) and resuspended in Dead Cell Removal MicroBeads (part of kit) at a concentration of 10^7 total cells/100ul beads. After incubation at RT for 15 min, cells were applied to an LS column (Miltenyi, 130-122-729), which was pre-rinsed with 3ml 1X Binding Buffer. The column was washed 4x with 3ml Binding Buffer and the flow through containing viable cells was collected. Cells in the flow through were pelleted and viability was confirmed using trypan blue. 2x10^6 viable cells were used for TotalSeq-C and anti-CD3-PE antibody staining. For this purpose, cells were washed with Cell Staining Buffer (Biolegend, 420201), pelleted at 600g for 10 min at 4°C, and resuspended in 90 ul Cell Staining Buffer. 10ul Human TruStain FcX Blocking solution (Biolegend, 422301) was added and cells were incubated for 10 min at 4°C. The TotalSeq-C Human Universal Cocktail 1.0 (Biolegend, 399905) was resuspended as described above, centrifuged at 14.000g for 10 min at 4°C and 25ul of the supernatant was added to the blocked cells. Individual TotalSeq-C antibodies were prepared as described above and 26ul of the master mix was added to each sample. To facilitate enrichment of immature and mature thymocytes via FACS, 10ul anti-CD3-PE (clone SK7, Biolegend, 344805) was added and samples were topped up with 40ul Cell Staining Buffer resulting in a total staining volume of 200ul. Samples were incubated for 30 min at 4°C in the dark. To wash off unbound antibody Cell Staining Buffer was added to the samples, and cells were pelleted for 10 min at 600g at 4°C. All supernatant was removed, cells were resuspended in Cell Staining Buffer, transferred to a new tube and pelleted as before. Cells were again resuspended in Cell Staining Buffer and pelleted and this wash step was repeated once more before cells were resuspended in 200ul MACS buffer (PBS + 2% FCS + 2mM EDTA) in preparation for sorting. 1ul PI was added for detection of dead cells and samples were sorted on a BD FACSAria III or BD FACSAria Fusion cell sorter using a 100 µm nozzle and a maximum flow rate of 4. |
| Instrument | Stroma sorting: Sony SH800 or Sony MA900. T cell enrichment for CITEseq: BD FACSAria III or BD FACSAria Fusion cell sorter |
| Software | Stroma sorting: Data was analysed using FCSExpress7 version. T cell enrichment for CITEseq: For the analysis software: CITE-seq was analysed using BD FACSDiva v8.0.1 and FlowJo v10. |
| Cell population abundance | Stroma sorting: CD45- cells constituted ~0.68-0.83% of the live liberase TH digested cells, while Thymic Epithelial cells composed ~0.19-0.25%. T cell enrichment for CITEseq: The proportion of CD3- cells was 24.2-29.6% of live cells and the proportion of CD3+ cells was 70.4-75.9% of live cells depending on the donor. |
| Gating strategy | Stroma sorting: Small debris was removed using forward/side scatter, no multiplet filtering was performed to ensure large |

Gating strategy

thymic epithelial cells (thymic nurse cells) can be included, dead cells were removed based on DAPI staining. CD45- cells were sorted to obtain stroma, while EPCAM+CD205- and EPCAM-CD205+ gates were pulled together to obtain total TEC fraction including cortical and medullary epithelial cells and sorted into solution of 2% FBS in PBS (Supplementary Fig.19A).
T cell enrichment for CITEseq: Cells were gated using forward/side scatter to remove doublets and debris, then dead cells were excluded based on PI staining. CD3- and CD3+ cells were collected separately in cooled IMDM + 50% FCS (Supplementary Fig.19B).

☒ Tick this box to confirm that a figure exemplifying the gating strategy is provided in the Supplementary Information.

