## [Peer Review file · Nature]

Manuscript Title: A spatial human thymus cell atlas mapped to a continuous tissue axis

Reviewer Comments & Author Rebuttals

Reviewer Reports on the Initial Version:

Referees' comments:

Referee #1 (Remarks to the Author):

Summary: This manuscript represents the work of collaborators from Europe, UK and North America who have provided precious tissues and data from fetal and pediatric human thymus samples. Some new biology is uncovered but the main importance of the work is the sophisticated integration of multiple types of data, and the development of new computational concepts and tools that aim to allow comparisons of samples from widely different stages of thymus development.

The data presentation is very dense and complex for someone not closely familiar with the thymus field so clarification of the jargon in the text and clearer presentation of some figures and legends would be helpful. Nonetheless the introduction, abstract and discussion are very well written.

As I am not a computational expert I will not address the technical aspects of the bioinformatics analyses. Assuming the computational tools are appropriately applied and the methodology adequately presented for others to reproduce, the paper would provide an important resource for the field of human thymus biology.

Major

1. The authors make an excellent argument for the need for a common coordinate framework (CCF) to compare cell types across thymus samples (particularly given the variability of human tissues) and propose the use of the "cortico-medullary axis (CMA)" for this purpose. As the CMA appears to be the foundation for all subsequent computational steps the criteria/ components that define the CMA should be made much clearer.

- It appears that CMA is defined morphologically (in the discussion (line 851) it states "CMA is solely based on morphological landmarks")- is this defined by a human operator based only on the 3 structures of cortex, medulla and capsule? How was reproducibility defined for this analysis?

-and does morphology alone define annotation at level 0?

-If it is purely morphologic, is the medulla simply defined as visually less cellular than cortex?

-please clarify how these morphologic structures are validated across samples especially in early fetal thymus in which very little medulla has developed?

Minor

1. In figure 1, the colors of the EpCam and CD3/45 markers (pink and red) are difficult to distinguish. There is a yellow dot in the sample but not in the key. The green ring for TCRab is very difficult to see.
2. In reference to how the CMA is defined for the CCF, the morphology in Figure 2 should be more clearly shown. In figure 2A, the thymus shown as an example of the structural landmarks is paediatric. Although the medulla and cortex can be easily distinguished in mature tissues (when development is largely complete), in early fetal stages the medulla is rare and the tissue is quite morphologically different. In Figure 2B, the fetal H&E histology is very difficult to see in terms of even cortex and medulla-the fetal samples should be shown at the same size as the paediatric. How is artefact defined? How is lymph defined?
3. Fig 2D legend: what is 'broad level' annotations-is the level 0 or 0 and 1?
4. Fig 4A: the text is difficult to see when laid over some of the clusters
5. Fig 4B: why is DLL4 expression so low in all TECs? FOXP1 also seems low
6. Fig 4D etc: the use of the terms medulla level I,II,III with mTEC I,II,III seems confusing
7. I may have missed it but did not see how mcTECs were defined
8. DN and DP (P) and (Q) are mentioned in fig 3 but not explained in legend. I could not find them explained until the methods section
9. On page 11, line 322 it is stated that ETPs were found in the medulla in BOTH fetal and paediatric tissues (Fig 3E). Fig 3E appears to show almost no ETPs in any area of the fetal thymus, and even the DN early have very low/absent relative abundance in the medulla. The text later on points out that the DNs are in the capsular/subcapsular regions of the fetal thymus. In contrast the ETPs and DN early are in the medulla of paediatric thymus.
10. Page 21, line 623 please explain the term "root"
11. Page 24, line 705, do the authors really think that these processes in DN and DPs take place "in the capsule". I thought the capsule is a fibrous tissue in which little if any haematopoietic activity occurs.
12. Explain the terms CD4/8 inversion when first used (line 726).
13. Page 26, in the long section describing chemokine expression, please clarify in each case whether the data is from fetus or paediatric or both
14. As the vast majority of data on thymus development is derived from mouse, the profound differences at least in the timing of development during pre and perinatal life between mouse and human should be emphasized in this paper as it is a major strength of the work.
15. Although the methods described how paediatric thymi were acquired and processed I could not find the same info for fetal tissue
16. Discussion page 29: line 861 needs some clarification about relative proportion....
17. Discussion page 29, line 870 "maintained over the first year of post-natal life": isn't the data out to 3 years?
18. Discussion page 30, line 881: it cannot be assumed that the CD194-CD197- phase ...retards the migration, only that this phenotype is associated with later migration

Referee #2 (Remarks to the Author):

The authors combine multimodal spatial and single-cell omics data with highly innovative computational methods to integrate all datasets to generate a thymus cell atlas. Besides, this paper proposes a new quantitative framework, the Cortico-Medullary Axis, to delineate T cell developmental trajectories from pre- to post-natal states in human thymus. This framework might be applicable to many other human organs or tissues that are comprised of repeating anatomic units. One of the interesting findings in this work is that T cell developmental trajectory is widely established by post-conception week 12 to 18. On the other hand, the resident stromal and immune cells, which provide help to T cell maturation, vary between fetal and pediatric thymus. The scRNA-seq and Visium data revealed several major subtypes in TECs and the localization of these subtypes in human thymus. Spatial omics data also demonstrate specific mTECs subtypes are associated with Hassall's Corpuscles (HCs). The data integration tools were able to discover T lineage maturation of CD4 trajectory in earlier stage compared to CD8 in post conception weeks 12-18. Moreover, annotation of different immune cell and resident stromal subsets revealed their variation of spatial distribution in capsule cortex and medulla of fetal and pediatric thymus. Integration of scRNA-seq data with spatial omics data demonstrated the variation of TEC progenitors, fibroblasts, and thymocytes in tissue niche and increase of mTECII and mTECIII subsets in Hassall's Corpuscles. All these are important findings in addition to reporting a human thymus cell atlas as a highly valuable resource. In particular, this paper introduced conceptual novel computational methods to bridge different spatial modality to help understanding T cell lymphocyte development and migration in thymus across different ages. This is a very well-written paper. There are only several major comments for the authors to consider to further improve the manuscript. Major comments with regard to the Cortico-Medullary Axis and its binned version.

1. Many interesting findings are based on the cortico-medullary axis, which provides us a novel way to integrate and analyze different spatial data about the same organ. Two parameters in the definition, the spot grid diameter r and number of neighbors K , might impact the representation of different compartments. Could the authors provide more details about how different r and K would influence compartments in both fetal and pediatric tissue samples and how the authors decide to choose the final optimal values for r and K ?

2. The Binned Axis to represent discrete annotation layers has been utilized a lot in this paper to compare distributions of different cell types between fetal and pediatric samples. If I understand correctly, the binned axis is derived solely from pediatric tissue samples by combining information from TissueTag and IBEX. I wonder if the authors can show more spatial results to validate this binned axis on fetal samples, maybe using data from Visium?

3. The integration data tools employed by Yayon et al prove to be a valuable approach in gaining insight into subset development within tissue niches. The dataset primarily draws from prior research, encompassing 7 donors from the authors' study and 6 pediatric donors. Additionally, spatial Visium fetal sample data is derived from three publicly available Visium datasets. Expanding the reference data sources to encompass other studies, especially for Visium data in fetal tissue, could enhance

comparative analyses. However, the datasets were generated from quite different methods and sample preparation processes. How much impact by differences in sample source, batch, and data pre-processing processes should be evaluated.

4. In their data integration of IBEX and Visium, Yayon et al utilized a grid diameter approach to spatially analyze and define cytokines in the portico medullary axis. While the method is well-documented, providing more specifics regarding the threshold for diameter in segment-free grid analysis and addressing associated challenges would furnish users with valuable insights and the ability to reproduce some of these results.

5. Furthermore, the incorporation of Knn neighboring in data integration for FISH and IBEX 3D to define TRC subset mapping lacks a comprehensive explanation of how it mitigates the heightened variability in spatial data compared to single-cell data. Elaborating on the components and thresholds of the Knn method would further enhance the manuscript's utility.

Referee #3 (Remarks to the Author):

General comments:

Yayon et al. report their studies aimed at (1) establishing a reference database for early human thymopoiesis, and (2) identifying novel biological features of pre- and postnatal stromal and lymphocyte development. Overall, this is an impressive, resource-like effort that will have a major impact on our understanding of normal physiology, and for clinical purposes, given that the increase in perinatal genetic studies uncover more and more subtle immunodeficiency states in both stromal and haematopoietic compartments. It also sets a standard for a similar stud(ies) in mice, still the most popular model for experimental immunology. The authors are to be gratulated for their efforts and their exceptionally well executed study.

This said, one cannot escape to note that the manuscript is a hard read, especially with the copious use of acronyms, the exquisite detail in the text, and the less than adequate figure legends. I have no easy solution for this, except to say that it may be worth considering focusing on the most interesting aspects in the main text, and to provide a more in-depth discussion in the supplementary material. Otherwise, I fear that the impact of the paper would be less than it deserves, which would be a pity.

Specific comments:

Introduction:

The authors begin by introducing some of the well-known features of thymopoiesis (a section that could well be shortened), and then clearly spell out the major gap in our knowledge about thymus function

that could not be addressed in the past: the dynamic interaction between a stationary component and the transient haematopoietic cells, nicely put in lines 118-121, and re-emphasized in lines 243-246.

Results_part 1: Method development

The authors provide, in exquisite detail the rationale of their computational tool development, and how these strategies might allow future data integration into this foundational reference set (Fig. 1,2; Extended Data Figures 1-4). I am not an expert in bioinformational tool development and hence cannot critique their strategy in detail, although it would seem to me that it does address the main biological aspects of a tissue as complex as the thymus.

Two aspects are notable here. First, annotations into different levels of resolutions are an excellent way of allowing other researchers to input their own data, even if they don't cover all the levels in a single experiment. Second, the idea of the "boundary-centred" approach is a clever way of dealing with the irregular 3D shape of the main tissue compartments.

Minor comments:

Fig. 2A/B: What are the ages of the thymi shown? Label: Septum (singular) rather than septa; the term "artefact" is not explained (fat tissue?)

Fig. 2F: typo CMA paediat – r -ic Visium

Fig. 2H: It is unclear what the green columns mean: # total genes/spot vs. # total exp./spot; please clarify
Scale bars undefined in B, C, D, missing in F

Results_part 2: General aspects of the biology of thymopoiesis

What are the novel aspects arising from the use of the newly developed data integration tools with respect to building and interpreting a continuous coordinate framework?

(1) Thymus morphology and spatial distribution of thymocyte subsets (Related to Figures 2/3; EDF 5/6)

Major comments:

The fine-grained binning analysis is a welcome and major improvement over previous analyses. I note that their first time point is too late with respect to capturing the initial formation of medullary islets and the formation of the fledgling cortico-medullary junction, as observed in the mouse [PMID1742403]). The bud-like appearance of cortical regions may be related phenomenon; it appears that the coalescence of the cortical structure happens after that of fusion the medullary islets into a continuum.

In this context: What does the equivalent of Fig. 3A look like in a fetal thymus? This may be relevant to the dissimilar distribution of ETP and DN thymocytes, apart from the developmentally related expansion of DP cells. In fact, it may be a reflection of bi-directional lympho-stromal interactions, as a consequence of the migratory landscape, partly illustrated in Fig. 3F. I think that Fig. 3F would be even more informative, if the distributions of the highlighted chemokines were correlated to the distributions of the corresponding chemokine receptors, such that – if there is no 1:1 correspondence across bins – one may even be able to infer something about the physiological range of chemokine gradients.

Minor comments:

Line 305: typo Medullary levels I-III (one - three!); check all figures!!

Fig. 3: age of paediatric thymus tissue?; typo: medullar -y.

Cell type annotation unclear (missing reference in legend) for Fig. 3 and EDF 5 (ie., T_DP (Q); T_DP(P); etc.)

(2) Non-epithelial stromal cells (Related to EDF 6-8)

Major comments:

I may have missed it, but do the authors have an indication about IL7 and FGF7/FGF10 expression (and if so where)?

The authors indicate that they have indications of fibroblast migration or maturation in the fetal thymus. Could one not distinguish between these two (non-mutually exclusive) alternatives by looking at chemokine/chemokine receptor expression patterns, and thereby also get an idea of potential fibroblast<>other interactions?

A rather remarkable finding is the presence of developing B cells in the cortex of the fetal thymus. Is this related to transiently (?) higher expression of B cell-promoting cytokines in the fetal cortex?

Minor comment:

EDF 7B: typo: General f-r-ibroblast

(3) Epithelial stromal cells (Related to Figures 4,5; EDF 9,10)

Major comments:

The major question addressed in this section is about the site of residence of bipotent TEC progenitors; this is indeed an important and (especially for the human situation) largely unresolved issue. This said, the data on the segmentation of cTEC and mTEC populations is a welcome refinement and consolidation of scattered earlier data, and provides important new data on the “mimetic”-type TECs, whose developmental history and position is not yet well known.

The most exciting observation of this section, at least in my mind, is the finding that putative precursor TECs are detectable at two sites, the capsular region, and at the CMJ. I could not discern whether the authors also obtain evidence of compartment-specific progenitors, as it is well possible that the “cortical” mcTECs are cTEC biased, and the “CMJ” mcTECs are mTEC biased. This should actually be possible to test by looking at their individual expression profiles. The overall conclusion (lines 540-542) may be somewhat premature unless this potential bias among the “cortical” mcTECs and the “CMJ” mcTECs is better resolved; note that the early stages of the thymus (at least by inference from mouse studies) might be heavily biased towards cTEC differentiation, in line with positive selection coming first, and that mTEC development occurs later (as it is also known that it is heavily dependent on thymocyte maturation). By contrast, I would tend to agree with the “supply role” of the capsular regions (lines 551-553).

Another interesting aspect of the present study concerns the characterization of Hassall’s corpuscles, a

tell-tale sign of human thymic stroma, but of unclear functional relevance. To do this, the authors change tactics, and instead of using a non-linear anchoring scheme (CMA), they measure features in linear distance from HCs. However, I am somewhat confused by the description of the cellular microenvironment of HCs. First of all, the TEC type of HCs does not appear in the fine-grained TEC subset analysis (or did I miss something?). I think this is important to state, as it would give (developmental) “meaning” to the spatial and fate-type analyses that follow in this section. I am also unsure about a proper control analysis: Would it be possible to perform the same analysis by starting from a point in the medulla equidistant to HCs? It could well be that there are other, so far hidden, organizing centres in the medulla, the HCs simply a most obvious one to test; the myo-TEC (surprisingly large numbers!) may be the obvious candidate to look at.

Minor comment:

Again, please check nomenclature, and the presence of scale bars, information on age of thymi etc.

(4) T cell development in thymic space (Related to Fig. 6; EDF 11,12)

Major comments:

The analysis of canonical T lineage differentiation is a highlight of the manuscript; I think that panel D of EDF 11 belongs to main Fig. 6, to put the presentation into context without having to refer back to the additional presentations. This topographical analysis provides a very strong data set supporting previous sketchy data/assumptions and is an important resource for further studies on lympho-stromal interactions, which underlie lineage commitment progression.

One such aspect is explored here in the analysis of CD4/CD8 bifurcation (starting from positive selection), focusing on the relationship between location (as a function of migration) and differentiation state. I do think, however, that the data presentation in Fig. 6 is somewhat confusing. First, panel A may well be relocated to EDF, then, once one has gone through the entire figure, one realizes that the colour code in B is not matching that in the remaining panels. Moreover, it seems that panels E and C are the most illuminating, and should perhaps be presented in reverse order?

The distinct differences of deduced migration patterns of CD4 and CD8 is a major finding, hence it would help to move panel C from EDF 12 to the main Fig. 6 to more easily illustrate the ligand/receptor relationships, and to accompany this presentation with a schematic highlighting the distinct responsiveness to CCL17/22.

I was also impressed by the seemingly perfect concordance between IBEX/Visium data sets, providing a showcase for the computational pipeline developed in this paper.

Discussion:

This section more or less recapitulates the results section. I suggest the authors focus on the differences between fetal and postnatal tissue; perhaps, they can also speculate what happens during thymic involution, and chart a road map for a more focused analysis of “older” thymi and those that are found in patients with inborn errors of stromal and haematopoietic compartments.

Referee #4 (Remarks to the Author):

Yayon et al present their work in mapping reference cell types atlases (using an all-integrated dataset) on spatial data. Multiple types of single cell and spatial datasets were used: scRNA, TCR-seq and CITE-seq for single cell data, and Visium, IBEX (for the most part) for spatial data. First, the authors present their approach to establishing a coordinate system to able to compare their results across different spatial platforms/modalities, while they use previously published approached for cell type calling or deconvolution. Next, they present the results, first on larger macrostructures in the thymus using their binned CMA regions (first on T cells and then on other cell populations), and then they move on to a higher resolution structure (HC) before finally back to mapping finer grained CITE-seq-defined immune T cells subsets across the inferred spatial regions in pediatric samples only.

There are two major highlights in the presented work: attempting to use a CCF for multi-tissue and multi-patient spatial analysis and integrating publicly available and newly generated single cell and spatial datasets from difficult to obtain fetal samples.

However, there are two major concerns the reviewer would like to point out:

1. Briefly, one could describe the CCF approach as follows:

1. manually annotate some components in an image (either HE or IF) based on choosing spots with highest marker expression
2. train and refine pixel classifier (basic random forest) to annotate the rest of the tissue section
3. calculate distances between the centroid of a region and the edge of another region or the edge of the tissue
4. bin these distances using user-based cut offs and add user-based labels to these bins marking regions of interest
5. repeat for every image in the dataset

While the authors posit the benefit of having a CCF approach (lines 127-143), this is not really what they did - there is not a real common coordinate system here. They used annotated tissue areas to bin their data into categories of observations so that they could compare those between conditions ie. pediatric and fetal; or within a condition ie. pediatric alone across the binned annotations. As such, the authors cannot argue that they built a completely novel spatial processing and analysis pipeline with a thymus CCF.

1. There is no analysis of the power vs. spatial variance in their study. Given that their analysis was not performed using any spatial statistics of Bayesian inference methods, it is hard to put value into observations from a handful of patients. While the reviewer fully acknowledges these samples are hard to obtain, there needs to be a clear presentation of the statistical power of this study to the readers, while now the results reads as an absolute "spatial thymus atlas". A suggestion could be to:

1. reanalyse a part of their data (eg. Figure 3 where they had both IBEX and Visium data) with a spatially

aware approach to see if they can with confidence detect the differentially expressed genes between regions or conditions:

2. report the spatial variance explained by this model alongside their results in Figure 2H as it is clear from results in 2H, that their own results show the importance of having “a spatial component” (albeit this is only in binned regions so there is much more space for improvement).
3. in case a) and b) yield good results, then a simple statement in the paper saying “these genes that we identify were confirmed with alternative spatially-aware approaches giving us confidence in our own approach” should suffice.

The reviewer is aware that there is not much difference between the binned IBEX vs. Visium data (for specific targets) and that the authors can argue they have cross-validated their results already. However, using a small targeted panel of antibodies also proves the point that these targets have already been validated and are not especially novel, while the authors went through lengths to create spatial RNA-seq data and integrated it with cell2location with scRNAseq (and CITE-seq) for novel marker discovery in order to make the thymus atlas.

Author Rebuttals to Initial Comments:

Reviewer #1:	
Comments	Response
This manuscript represents the work of collaborators from Europe, UK and North America who have provided precious tissues and data from fetal and pediatric human thymus samples. Some new biology is uncovered but the main importance of the work is the sophisticated integration of multiple types of data, and the development of new computational concepts and tools that aim to allow comparisons of samples from widely different stages of thymus development. The data presentation is very dense and complex for someone not closely familiar with the thymus field so clarification of the jargon in the text and clearer presentation of some figures and legends would be helpful. Nonetheless the introduction, abstract and discussion are very well written. As I am not a computational expert I will not address the technical aspects of the bioinformatics analyses. Assuming the computational tools are appropriately applied and the methodology adequately presented for others to reproduce, the paper would provide an important resource for the field of human thymus biology.	We thank the reviewer for pointing out that this was, and still is, a collaborative effort of multiple labs each bringing their own experience and expertise. We agree that the manuscript was very complex. To retain focus on the key findings and to make it more accessible to a broader audience, we have dramatically restructured and shortened the manuscript as follows: 1. We moved technical details on TissueTag and the CMA axis into two dedicated Supplementary Notes 1 and 2. This way we can utilise more space to explain in detail the motivation behind the tissue annotation and axis framework, and the mathematical principles. We have also added simulation and visualisation of data for further clarification and show how to implement and adapt the approach. The Supplementary Notes are also accompanied by 5 Supplementary Figures (3 added to this revision). We have also generated a dedicated online tutorial to explain the principle of the axis we show in this manuscript https://organ-axis-tutorial.readthedocs.io/en/latest/index.html#. Through this change we hope to make the axis framework and the toolkit we developed more accessible.2. We merged Imagespot into TissueTag for the sake of simplicity so that all annotation and axis functions and utilities are in one place.3. We moved the single-cell atlasing parts, in which we show cell annotations and Visium spatial mapping of detailed T lineage stages, myeloid, B, vascular and stromal cells in the scRNA-seq and CITE-seq data, to Supplementary Notes. This way, we hope to focus the text on the specific biological findings and technical advancements that are

	relevant to the CMA, while retaining the valuable information on the generated data sets, which is needed in order for them to function as universal reference for other researchers.
1.1 The authors make an excellent argument for the need for a common coordinate framework (CCF) to compare cell types across thymus samples (particularly given the variability of human tissues) and propose the use of the “cortico-medullary axis (CMA)” for this purpose. As the CMA appears to be the foundation for all subsequent computational steps the criteria/ components that define the CMA should be made much clearer.  1. It appears that CMA is defined morphologically (in the discussion (line 851) it states “CMA is solely based on morphological landmarks”)- is this defined by a human operator based only on the 3 structures of cortex, medulla and capsule? How was reproducibility defined for this analysis? 2. and does morphology alone define annotation at level 0? 3. If it is purely morphologic, is the medulla simply defined as visually less cellular than cortex? 4. please clarify how these morphologic structures are validated across samples especially in early fetal thymus in which very little medulla has developed? 	This is a very important question and we put a great deal of effort into getting accurate morphological annotations, work that was not reflected enough in the initial submission. The morphological annotation and evaluations were done in consultation with expert human thymic pathologists: Prof Dr Alexander Marx who leads the tertiary referral centre for thymic abnormalities in Germany, now rightfully credited in the Acknowledgements, as well as co-authors Dr Alexandra Kreins and Prof Dr Graham Davies, who lead the European/UK centre for thymic transplantation at GOSH. We also benefit from the long experience of working on human thymic tissues of our co-authors Luigi Notarangelo, Ronald Germain and Andrea Radtke at the NIH. Specifically for the boundary between the cortex and medulla, which is very difficult for the human eye to call objectively, we introduce a machine learning approach and train a random forest classifier based either on two marker genes (ARPP21 for cortex and AIRE for medulla) for Visium datasets or using manual “scribbling” on easily discernible regions of cortex and medulla for IBEX sections (Supplementary Note1, Supplementary Figure 1). This classifier predicts the location of cortex and medulla accurately in an unbiased and observer-agnostic way. In addition, we manually added the annotation for the capsule, PVS, Hassall’s corpuscles, large vessels, and lymphoid aggregates, guided by the abovementioned experts and our own expertise. We now emphasise this additional information in the methods and supplementary sections of the

		manuscript and clarify our approach in the main text. With respect to the clarification of the CMA criteria and components we have generated a comprehensive Supplementary Note 2, which elaborates on the key prerequisites for axis establishment and further offers several visual examples on how parameter choice affects the axis. We hope that this will help clarify the OrganAxis approach and facilitate its uptake by scientists without computational/statistical background.
1.5	1. In figure 1, the colors of the EpCam and CD3/45 markers (pink and red) are difficult to distinguish. There is a yellow dot in the sample but not in the key. The green ring for TCRab is very difficult to see.	This figure has now been simplified to place emphasis on spatial data curation in this study, since this had apparently caused some confusion. We have moved the schematic on enrichment strategies to Extended Data Figure 1, simplified it and made it accessible to colorblind individuals.
1.6	2. In reference to how the CMA is defined for the CCF, the morphology in Figure 2 should be more clearly shown. In figure 2A, the thymus shown as an example of the structural landmarks is paediatric. Although the medulla and cortex can be easily distinguished in mature tissues (when development is largely complete), in early fetal stages the medulla is rare and the tissue is quite morphologically different. In Figure 2B, the fetal H&E histology is very difficult to see in terms of even cortex and medulla-the fetal samples should be shown at the same size as the paediatric.	In this version of the manuscript we now display a fetal tissue example alongside all the locations where fetal data is presented. We have moved the basic thymus morphology to Figure 1 for general reference. In Figure 2 we have enlarged the H&E images as much as possible within the space limitations and offer a cropped zoomed in image to further illustrate the differences in morphology. Figure 3 now also shows the binned axis on fetal samples and in Supplementary Figures 4-5 we highlight how the axis is established on fetal samples as well. Finally, we will upload all imaging datasets to repositories that allow browsing and interactivity at full resolution.
1.7	Figure 2: How is artefact defined? How is lymph defined?	Similar to other morphological regions, we have consulted with leading histologists and identified regions of holes, tears, freezing, sectioning artefacts

		by manual assessment and annotation. Thymus-associated lymphoid aggregates had been previously described in the study of Suo et al. (2022), from which the samples in question originate.
1.8	3. Fig 2D legend: what is 'broad level' annotations-is the level 0 or 0 and 1?	With "broad level annotation" we were referring to macroscopic regions like the cortex and medulla. We have now removed this phrasing from the text to avoid confusion. Instead, we refer to it as "annotation level 0", which holds the basic components for calculating the CMA.
1.9	4. Fig 4A: the text is difficult to see when laid over some of the clusters	We apologise for this oversight and have corrected this in the revised manuscript.
1.10	5. Fig 4B: why is DLL4 expression so low in all TECs? FOYN1 also seems low	We think that this issue of DLL4/FOYN1 detection is related to a lack of sensitivity. Moreover, it is possible that transcript levels for these genes are generally low compared to the corresponding protein levels, for instance due to repeated translation of the same transcripts in combination with high protein stability. In a previous study, in which we quantified DLL4 expression using qRT-PCR, we found this to be difficult to detect and observed low transcript levels in cTECs but no expression in mTECs (Van de Walle et al., Blood 2011). Moreover, a recent single cell study also demonstrates low DLL4 levels in cTECs and absence of DLL4 transcripts in mTECs (Stankiewicz et al., bioRxiv 2023).
1.11	6. Fig 4D etc: the use of the terms medulla level I,II,III with mTEC I,II,III seems confusing	We have changed the naming of the axis levels to Arabic numerals ('Medullary level 1, 2, 3' etc.) throughout to make it clear that m/cTEC/I/II/III and CMA levels are not connected.
1.12	7. I may have missed it but did not see how mcTECs were defined	We have now clarified the definition of mcTEC in the text: "mcTECs were defined based on intermediate levels of cTEC and mTEC markers and high expression of stem cell markers KRT15 and ITGA6

		(Ulyanchenko et al. 2016; Campinoti et al. 2020; Giroux et al. 2017) (Figure 4b). They were further characterised by the expression of DLK2 (reported in (Park et al. 2020)), insulin-like growth factor binding protein IGFBP6 , connective tissue growth factor CCN2 and the cytokine CCL2 .”
1.13	8. DN and DP (P) and (Q) are mentioned in fig 3 but not explained in legend. I could not find them explained until the methods section	We have clarified the meaning of (P) and (Q) in the figure legend for Figure 3 and have additionally introduced the terms in the corresponding main text.
1.14	9. On page 11, line 322 it is stated that ETPs were found in the medulla in BOTH fetal and paediatric tissues (Fig 3E). Fig 3E appears to show almost no ETPs in any area of the fetal thymus, and even the DN early have very low/absent relative abundance in the medulla. The text later on points out that the DNs are in the capsular/subcapsular regions of the fetal thymus. In contrast the ETPs and DN early are in the medulla of paediatric thymus.	We would like to clarify that the plots represent relative abundance. Due to increased cellularity of the capsular region in the paediatric compared to the fetal thymus, the relative cell density for ETP and DN(early) thymocytes in this region might be lower in the paediatric tissue even if the total cell numbers may be comparable between the two developmental stages. The absolute cell numbers are very difficult to infer from Visium but the overall numbers of these early thymocytes are very low, as indicated in the barplots in Extended Data Figure 4a, which further complicates their robust mapping in the Visium data. In fact, after the aforementioned reprocessing of the data, the updated plots provide improved comparability of fetal and paediatric mapping of the ETP/DN thymocytes (see Figure 3e). We attribute this to the fact that several of the affected cells that have now been removed were part of the ETP and DN(early) population and may have had a contaminating effect.
1.15	10. Page 21, line 623 please explain the term “root”	The term root refers to the trajectory analysis and defines the population that was used as a point of origin for this analysis. We have now shortened the main text so this explanation is only mentioned in the methods section.

1.16	11. Page 24, line 705, do the authors really think that these processes in DN and DPs take place “in the capsule”. I thought the capsule is a fibrous tissue in which little if any haematopoietic activity occurs.	The reviewer is correct that DN/DP thymocytes are not actually located in the capsule as the capsule is indeed a fibrous structure a few micrometres thick. We use the term “capsular” (instead of “capsule”) to indicate a manually annotated 25-50 um region spanning this structure to capture these areas consistently in fetal and paediatric Visium analyses as well as using the same definition of that region for IBEX. This region therefore naturally includes some cell-containing areas directly surrounding the actual capsule. By integrating multiple samples with the same CCF we can overcome the extreme sparsity in this region but not overcome the 55um resolution innate to Visium spots and cannot distinguish structures smaller than 50 um in width.
1.17	12. Explain the terms CD4/8 inversion when first used (line 726).	We have extended the sentence on co-receptor reversal to clarify that this refers to the switch from the CD4 ^{hi} CD8 ^{lo} to CD4 ⁻ CD8 ⁺ phenotype.
1.18	13. Page 26, in the long section describing chemokine expression, please clarify in each case whether the data is from fetus or paediatric or both	Given that the CITE-seq data was exclusively generated from paediatric thymus samples, all analyses relating to Figure 6 were carried out using paediatric single cell and spatial data. We have clarified that in the corresponding figure legends for Figure 6h and Extended Data Figure 8c.
1.19	14. As the vast majority of data on thymus development is derived from mouse, the profound differences at least in the timing of development during pre and perinatal life between mouse and human should be emphasized in this paper as it is a major strength of the work.	We fully agree with this statement and have elaborated on this in the discussion (lines 480-489). Specifically, we have added a comparison of the timing of T cell generation and organ development between human and mouse.
1.20	15. Although the methods described how paediatric thymi were acquired	We thank the reviewer for pointing out this missing piece of information. We have now added this to the methods and indicate the origin of all tissues both in the methods and in Figure 1c. Of note, we

	and processed I could not find the same info for fetal tissue	did generate fetal Visium, RareCyte and RNAscope data, whereas all fetal single cell data used in this study had been previously published.
1.21	16. Discussion page 29: line 861 needs some clarification about relative proportion....	We have rewritten the discussion, which no longer references the relative proportion of cells. Our intention was to emphasise that local cell density within thymic regions is expected to change throughout development due to migration and proliferation of thymocytes in particular. Thus, when visualising relative proportions of cells, rare cell types, such as ETPs, may be masked in samples with high cell density due to the overwhelming abundance of other cells (DPs) in this region, despite comparable absolute ETP numbers.
1.22	17. Discussion page 29, line 870 “maintained over the first year of post-natal life”: isn’t the data out to 3 years?	The reviewer is correct that our data extends up to 3 years of age. We have corrected this mistake.
1.23	18. Discussion page 30, line 881: it cannot be assumed that the CD194-CD197- phase ...retards the migration, only that this phenotype is associated with later migration	We agree that we cannot prove a causal connection. We have rephrased our statements to reflect this.
Reviewer #2:		
Comments		Response
The authors combine multimodal spatial and single-cell omics data with highly innovative computational methods to integrate all datasets to generate a thymus cell atlas. Besides, this paper proposes a new quantitative framework, the Cortico-Medullary Axis, to delineate T cell developmental trajectories from pre- to post-		We thank the reviewer for this positive response and for recognising the relevance of our findings for the thymus field.

natal states in human thymus. This framework might be applicable to many other human organs or tissues that are comprised of repeating anatomic units. One of the interesting findings in this work is that T cell developmental trajectory is widely established by post-conception week 12 to 18. On the other hand, the resident stromal and immune cells, which provide help to T cell maturation, vary between fetal and pediatric thymus. The scRNA-seq and Visium data revealed several major subtypes in TECs and the localization of these subtypes in human thymus. Spatial omics data also demonstrate specific mTECs subtypes are associated with Hassall's Corpuscles (HCs). The data integration tools were able to discover T lineage maturation of CD4 trajectory in earlier stage compared to CD8 in post conception weeks 12-18. Moreover, annotation of different immune cell and resident stromal subsets revealed their variation of spatial distribution in capsule cortex and medulla of fetal and pediatric thymus. Integration of scRNA-seq data with spatial omics data demonstrated the variation of TEC progenitors, fibroblasts, and thymocytes in tissue niche and increase of mTECII and mTECIII subsets in Hassall's Corpuscles. All these are important findings in addition to reporting a human thymus cell atlas as a highly valuable resource. In particular, this paper introduced conceptual novel computational methods to bridge different spatial modality to help understanding T cell lymphocyte development and migration in thymus across different ages. This is a very well-written paper. There are only several major comments for the authors to consider to further improve the manuscript.

2.1	Major comments with regard to the Cortico-Medullary Axis and its binned version. 1. Many interesting findings are based on the cortico-medullary axis, which provides us a novel way to integrate and analyze different spatial data about the same organ. Two parameters in the definition, the spot grid diameter r and number of neighbors K, might impact the representation of different compartments. Could the authors provide more details about how different r and K would influence compartments in both fetal and pediatric tissue samples and how the authors decide to choose the final optimal values for r and K?	The reviewer makes an excellent point that we had admittedly not clarified enough in the text mostly due to lack of space. In this revision, we moved technical details on TissueTag and the CMA into two dedicated Supplementary Notes 1 and 2 in order to have sufficient space to explain the motivation and mathematical principles, add data simulations, and show how to implement the approach, further supporting this by 5 Supplementary Figures. In addition, we have generated a dedicated online tutorial to explain the principle of the axis we show in this manuscript https://organ-axis-tutorial.readthedocs.io/en/latest/index.html#. To illustrate the impact of the choice of r/K to the reader, we show, using a simplified simulated dataset with varying parameters, how different r/K combinations affect robustness, shape and sensitivity of the axis to different distances from the border between regions (Supplementary Note 2 (lines 164-186), Supplementary Figure 3). We further calculate the axis on both fetal and paediatric datasets using varying values for K to show how this reflects the innate tissue curvature, the influence on robustness and why we selected specific parameters ($r=15\mu\text{m}$ and $K=10$) for our study (Supplementary Note 2 (lines 220-225), Supplementary Figure 4). Through these simulations we also demonstrate that within a large window of parameters the axis is very robust and consistent. Nevertheless, once selected, the parameters should be consistent across the entire study to compare spatial observations. We want to emphasise that the choice of r/K will depend on the tissue, technology and research question. In this study, our aim was to facilitate comparability between samples obtained from different donors at varying ages and profiled with different technologies, which is why we chose the
------------	---	--

		same parameters for fetal vs. paediatric and Visium vs. IBEX data.
2.2	2. The Binned Axis to represent discrete annotation layers has been utilized a lot in this paper to compare distributions of different cell types between fetal and pediatric samples. If I understand correctly, the binned axis is derived solely from pediatric tissue samples by combining information from TissueTag and IBEX. I wonder if the authors can show more spatial results to validate this binned axis on fetal samples, maybe using data from Visium?	We apologise for the confusion. We took both age ranges (fetal and paediatric) into account when selecting the bin boundaries. We now provide a detailed explanation in Supplementary Note 2 (lines 226-246) on how the binned axis was obtained and have added examples of the fetal dataset highlighting the implementation of both continuous and binned axis (Supplementary Figure 4-5).
2.3	3. The integration data tools employed by Yayon et al prove to be a valuable approach in gaining insight into subset development within tissue niches. The dataset primarily draws from prior research, encompassing 7 donors from the authors' study and 6 pediatric donors. Additionally, spatial Visium fetal sample data is derived from three publicly available Visium datasets. Expanding the reference data sources to encompass other studies, especially for Visium data in fetal tissue, could enhance comparative analyses. However, the datasets were generated from quite different methods and sample preparation processes. How much impact by differences in sample source, batch, and data pre-processing processes should be evaluated.	We acknowledge that our previous Figure 1 did not clarify enough which datasets were original to our study and which had been integrated from past studies. Figure 1c now specifically focuses on this and highlights the major addition of spatial data in this manuscript. Specifically, all Visium data except for 3 fetal samples published in Suo et al. (2022) was newly generated. We agree that more fetal data would have been extremely useful as the organ undergoes rapid changes during the course of a few weeks in fetal development. However, no other publicly available fetal thymus Visium data exists that we are aware of, and thus such data could not be incorporated in our analysis. Moreover, fetal samples are very difficult to obtain, especially for later gestational timepoints and we were, therefore, unable to generate more fetal Visium data or expand the age range. To address concerns about batch effects stemming from differences in sample source and data

		processing, we have added multiple analyses to estimate sample and batch variance in Visium data: 1. We calculated the CMA-PCA explained variance per sample (Extended Data Figure 3d) and observed no donor or age dependent effect. In fact, the variance within donor samples is within the range of variance between age groups. We estimate this variance to come mostly from technical factors.2. To evaluate differential gene expression patterns (for cytokines) across the thymus we performed a 2-way ANOVA for the expression across bins while grouping data per sample to avoid artificially inflating statistical power. This analysis is presented in Extended Data Figure 4b and highlights the sample variance per gene across the axis in a more informative way.3. Finally we wanted to use the full extent of the data across the axis so we fit a spline model to the axis. This allows us to estimate, within each bin, if a gene is significantly associated to the axis across a specific range. Here we could account also for donor and age variance. We used this model to investigate the association of cytokines and their receptors within an age group. This analysis is presented in the Revision Figure 1 (at the end of this document). However, it is important to emphasise that the ability to infer cytokine functionality by measuring the RNA level might lead to very different results if compared to protein data. 4. In addition, we show potentially confounding metadata parameters, such as sample batch and processing pipeline version on UMAPs of combined samples for the fetal and paediatric Visium and the paediatric IBEX data, which confirm the absence of substantial batch effects (Extended Data Figure 3a-c). The functionalities mentioned in points 2 and 3 will be included in the TissueTag software package as
--	--	---

		additional “axis utilities” in order to allow future users to carry out similar analyses.
2.4	4. In their data integration of IBEX and Visium, Yapon et al utilized a grid diameter approach to spatially analyze and define cytokines in the portico medullary axis. While the method is well-documented, providing more specifics regarding the threshold for diameter in segment-free grid analysis and addressing associated challenges would furnish users with valuable insights and the ability to reproduce some of these results.	We had utilised the segmentation-free grid approach to investigate the effect of the 50 um spot environment on the ability of the CMA to explain spatial variance and found that indeed this type of segmentation-free analysis preserved more spatial information than single nuclei segmentations. However, this analysis was only used for that purpose and led to confusion with the high-resolution grid we use as an intermediate step in order to calculate the axis in the same manner across all datasets. For these reasons we decided to remove the segmentation-free grid analysis of IBEX data in the revised version of the manuscript.
2.5	5. Furthermore, the incorporation of Knn neighboring in data integration for FISH and IBEX 3D to define TRC subset mapping lacks a comprehensive explanation of how it mitigates the heightened variability in spatial data compared to single-cell data. Elaborating on the components and thresholds of the Knn method would further enhance the manuscript's utility.	We thank the reviewer for this comment as the KNN method is indeed non-trivial but very powerful to infer the cell types from single cell spatial data. This method was originally developed in the context of another study from our lab and is now a part of a different manuscript which is under review but the package is functional and open with additional details provided here https://github.com/Teichlab/iss_patcher . To clarify, we did not use ISS patcher on RNAscope data as it is built for highly multiplexed data. In fact, the number of channels for IBEX is already on the lower side for this method.
Reviewer #3:		
Comments		Response

Yayon et al. report their studies aimed at (1) establishing a reference database for early human thymopoiesis, and (2) identifying novel biological features of pre- and postnatal stromal and lymphocyte development.

Overall, this is an impressive, resource-like effort that will have a major impact on our understanding of normal physiology, and for clinical purposes, given that the increase in perinatal genetic studies uncover more and more subtle immunodeficiency states in both stromal and haematopoietic compartments. It also sets a standard for a similar stud(ies) in mice, still the most popular model for experimental immunology. The authors are to be gratulated for their efforts and their exceptionally well executed study.

This said, one cannot escape to note that the manuscript is a hard read, especially with the copious use of acronyms, the exquisite detail in the text, and the less than adequate figure legends. I have no easy solution for this, except to say that it may be worth considering focusing on the most interesting aspects in the main text, and to provide a more in-depth discussion in the supplementary material. Otherwise, I fear that the impact of the paper would be less than it deserves, which would be a pity.

We thank the reviewer for the positive feedback and detailed comments.

We fully agree with the note that the manuscript is complex and have embraced the suggestion of simplifying the main text while shifting additional detailed information to the supplementary section. Consequently, we have comprehensively restructured and shortened the manuscript in the following way:

1. We moved technical details on TissueTag and the CMA into two detailed Supplementary Notes 1-2, which allowed us to dedicate more space to explanations on the motivation and mathematical principles behind the tissue annotation and axis framework. These notes also contain simulations and visualisations of data for further clarification and to demonstrate how to implement and modify the approach. Our intention was to make the axis framework and the TissueTag toolkit accessible to the interested reader but avoid excessive detail in the main text.

2. We moved the long sections describing the composition of our single cell atlas to dedicated supplementary notes, in which we show cell annotations and Visium spatial mapping of detailed T lineage stages, myeloid and stromal cells. This way, we hope to focus the text on the CMA-related specific biological findings and technical advancements but keep the information on the generated scRNA-seq and CITE-seq data sets accessible to researchers interested in this atlas part.

3. We merged Imagespot into TissueTag for the sake of simplicity so that all annotation and axis functions and utilities are in one place.

4. We have reduced the number of acronyms in the text as much as possible and have taken care to formally introduce the remaining ones upon first use and define them in the figure legends where

	needed. We hope that the remaining acronyms, which we deem to be frequently used in the field, are familiar to most readers and actually enhance readability by preventing the overuse of long complex terms. 5. We have removed detailed descriptions on methodology and less relevant biological findings from the text and substantially shortened the length of the main text (from above 10000 words down to ~5700). We hope that this has improved the clarity of the manuscript and resulted in a more result-focused revised version.	
3.1	Introduction: The authors begin by introducing some of the well-known features of thymopoiesis (a section that could well be shortened), and then clearly spell out the major gap in our knowledge about thymus function that could not be addressed in the past: the dynamic interaction between a stationary component and the transient haematopoietic cells, nicely put in lines 118-121, and re-emphasized in lines 243-246.	To reduce the length of the manuscript, we have shortened the introduction to only retain the information that is directly relevant for understanding the motivation behind our approach as well as the fundamental biological background of thymic composition and T cell development.
3.2	Results_ part 1: Method development The authors provide, in exquisite detail the rationale of their computational tool development, and how these strategies might allow	We thank the reviewer for this observation. While we are careful to not overstate 3D claims in this manuscript, we do think that the axis is a good approximation. However to fully build a three-dimensional model we would need to generate 3D

	future data integration into this foundational reference set (Fig. 1,2; Extended Data Figures 1-4). I am not an expert in bioinformational tool development and hence cannot critique their strategy in detail, although it would seem to me that it does address the main biological aspects of a tissue as complex as the thymus. Two aspects are notable here. First, annotations into different levels of resolutions are an excellent way of allowing other researchers to input their own data, even if they don't cover all the levels in a single experiment. Second, the idea of the "boundary-centred" approach is a clever way of dealing with the irregular 3D shape of the main tissue compartments.	thymus high resolution imaging datasets from multiple donors to test how well the model would estimate 3D space.
3.3	Fig. 2A/B: What are the ages of the thymi shown? Label: Septum (singular) rather than septa; the term "artefact" is not explained (fat tissue?)	We have updated figure legends for all relevant figures to give details on the age of the thymus samples and adjusted the name for septum. We define "artefact" as regions of holes, tears, freezing or sectioning artefacts by manual assessment and annotation.
3.4	Fig. 2F: typo CMA paediat – r -ic Visium	We have corrected this.
3.5	Fig. 2H: It is unclear what the green columns mean: # total genes/spot vs. # total exp./spot; please clarify Scale bars undefined in B, C, D, missing in F	We have now clarified in the text: "To quantify how the CMA captures biological variance we performed a principal component analysis (PCA) on the full feature space (genes for Visium, channels for IBEX). We then derived the cumulative explained variance and compared this value to the CMA and to the most significant technical factor in Visium (number

		of genes per spot) or IBEX (total signal across all channels per cell)". We have checked and added scale bars to all the microscopy images throughout the manuscript.
3.6	The fine-grained binning analysis is a welcome and major improvement over previous analyses. I note that their first time point is too late with respect to capturing the initial formation of medullary islets and the formation of the fledgling cortico-medullary junction, as observed in the mouse [PMID1742403]). The bud-like appearance of cortical regions may be related phenomenon; it appears that the coalescence of the cortical structure happens after that of fusion the medullary islets into a continuum. In this context: What does the equivalent of Fig. 3A look like in a fetal thymus? This may be relevant to the dissimilar distribution of ETP and DN thymocytes, apart from the developmentally related expansion of DP cells. In fact, it may be a reflection of bi-directional lympho-stromal interactions, as a consequence of the migratory landscape, partly illustrated in Fig. 3F.	We agree that our data is missing a significant portion of human development. This is largely due to the lack of availability of early embryonic and very late gestation fetal samples. In respect to the difference between ETP and DN locations in fetal vs. paediatric thymus we consider that location of vasculature might hold some interesting keys for thymocyte migration. To further explore the lympho-stromal interactions and their potential influence on the migration in both age groups we have added a dotplot indicating cell types producing cytokines which show difference in their spatial patterns across CMA between fetal and paediatric as estimated by low cosine similarity, or significant ANOVA interaction effect between age group and CMA layer (Figure 3f, Extended Data figure 4c). Following the reviewer's comment we also now show examples of the annotations and continuous as well as binned axis on the fetal thymus (Figure 2a,c, Figure 3b).
3.7	I think that Fig. 3F would be even more informative, if the distributions of the highlighted chemokines were correlated to the distributions of the corresponding chemokine receptors, such that – if there is no 1:1 correspondence across bins – one may even be able to infer something	We have further explored the expression gradients of cytokines/chemokines and the corresponding receptors and agree that the combination of both could be informative to infer the thymic regions where these are functional. We provide several examples for relevant thymic chemokines in Revision Figure 1 (at the end of this document).

	about the physiological range of chemokine gradients.	However, we urge caution in the interpretation of these gradients given the fact that these are only representing transcription, not protein expression nor secretion, and that they do not provide information about the 'receiving' and 'sending' cells, both in terms of their identity and their frequency. That being said, we do highlight chemokine/receptor expression along the axis that we can now accurately quantify through paired cosine similarity for several relevant chemokines in Revision Figure 1. The tool that we developed for this purpose will be included in our software package and available for further data exploration.
3.8	Line 305: typo Medullary levels I-III (one - three!); check all figures!!	We have corrected this typo.
3.9	Fig. 3: age of paediatric thymus tissue?; typo: medullar -y.	We have corrected the typo and added age details for the respective thymus sample in the figure legend.
3.10	Fig3: Cell type annotation unclear (missing reference in legend) for Fig. 3 and EDF 5 (ie., T_DP (Q); T_DP(P); etc.)	We have included explanations for these acronyms in the figure legend and introduced them in the main text.
3.11	I may have missed it, but do the authors have an indication about IL7 and FGF7/FGF10 expression (and if so where)?	IL7 does indeed represent an important cytokine for thymic development. However, we were not able to robustly detect IL7 and FGF7/10 transcripts, most likely due to a lack of sensitivity of Visium, and therefore chose not to include these in the figure.
3.12	The authors indicate that they have indications of fibroblast migration or maturation in the fetal thymus. Could one not distinguish between these two (non-mutually exclusive) alternatives by looking at chemokine/chemokine receptor expression patterns, and thereby also	We agree with the reviewer that this is an interesting aspect and should be explored in further studies. In our attempt to make the manuscript more concise we have decided to move the larger descriptions surrounding fibroblast identification and localisation to the Supplementary Note 3. We further anticipate the suggested analysis to be challenging due to the limited number of fibroblasts

	get an idea of potential fibroblast<>other interactions?	we can identify in the fetal thymus and the uncertainty about which chemokines/cytokines would drive migration vs. maturation. We therefore consider this analysis out of scope.
3.13	A rather remarkable finding is the presence of developing B cells in the cortex of the fetal thymus. Is this related to transiently (?) higher expression of B cell-promoting cytokines in the fetal cortex?	We agree that this is an interesting observation. Of note, in our previous study (Suo et al.), which had 3 fetal Visium samples, the pro-B population was also observed in the cortex. An important cytokine supporting pro-B cell maturation is CXCL12, which is known to be expressed in the cortex. However, both CXCL12 and IL7, which also promotes B cell development, are only lowly expressed or difficult to detect in the fetal thymus. Moreover, the low number of detected developing B cells does not allow us to assess this question in more detail.
3.14	EDF 7B: typo: General f-r-ibroblast	We have corrected this typo.
3.15	The major question addressed in this section is about the site of residence of bipotent TEC progenitors; this is indeed an important and (especially for the human situation) largely unresolved issue. This said, the data on the segmentation of cTEC and mTEC populations is a welcome refinement and consolidation of scattered earlier data, and provides important new data on the “mimetic”-type TECs, whose developmental history and position is not yet well known. The most exciting observation of this section, at least in my mind, is the finding that putative precursor TECs are detectable at two sites, the capsular region, and at the CMJ. I could not discern whether the authors also obtain evidence of	We thank the reviewer for this valuable suggestion. We have carried out additional analyses on the mcTECs in the paediatric thymus scRNA-seq data and annotated mcTEC subtypes based on their predicted bias towards c/mTECs. cTEC-biased/primed mcTECs were more enriched in the capsular niche, whereas mTEC-primed mcTECs were more enriched in the CMJ region (based on Visium deconvolution). In addition, both regions contained mcTECs that did not exhibit a clear preference in their priming towards cTEC vs. mTEC. We identified the divergent expression profiles of mcTEC marker genes in the differently-primed mcTECs and were able to confirm the gene profiles when in capsular vs. CMJ-associated mcTECs in the IBEX data. Of note, regardless of priming, mcTECs were clearly distinct from mTECs and cTECs in their expression profiles, confirming that this population is correctly annotated and not contaminated by mature cells.

	compartment-specific progenitors, as it is well possible that the “cortical” mcTECs are cTEC biased, and the “CMJ” mcTECs are mTEC biased. This should actually be possible to test by looking at their individual expression profiles. The overall conclusion (lines 540-542) may be somewhat premature unless this potential bias among the “cortical” mcTECs and the “CMJ” mcTECs is better resolved; note that the early stages of the thymus (at least by inference from mouse studies) might be heavily biased towards cTEC differentiation, in line with positive selection coming first, and that mTEC development occurs later (as it is also known that it is heavily dependent on thymocyte maturation). By contrast, I would tend to agree with the “supply role” of the capsular regions (lines 551-553).	We now present these additional analyses in Supplementary Note 5 and Supplementary Figure 16.
3.16	Another interesting aspect of the present study concerns the characterization of Hassall’s corpuscles, a tell-tale sign of human thymic stroma, but of unclear functional relevance. To do this, the authors change tactics, and instead of using a non-linear anchoring scheme (CMA), they measure features in linear distance from HCs. However, I am somewhat confused by the description of the cellular microenvironment of HCs. First of all, the TEC type of HCs does not appear in the fine-grained TEC subset analysis (or did I miss something?). I think this is important to state, as it would give	The TEC subtype giving rise to Hassall’s corpuscles is mTECIII, also referred to as keratinocytes. This has been confirmed by our proximity analysis (Figure 5c) as well as suggested in prior literature (Kadouri et al, 2020). Of note, as mTECIII turn into HCs, they lose their nuclei and acquire a tough outer cell envelope similar to the process that happens in the outer layer of skin. Hence, they cannot be captured in single-cell and spatial sequencing. Regarding the suggestion for a ‘control analysis’: given the small size of the medulla and absence of 3D information it is difficult to define the ideal control points for such analysis without a morphological landmark to focus on, such as the HC. Our analysis was driven by the biological question for the organisation of the HC vs. the general medullary depth, which we can now untangle in

	(developmental) “meaning” to the spatial and fate-type analyses that follow in this section. I am also unsure about a proper control analysis: Would it be possible to perform the same analysis by starting from a point in the medulla equidistant to HCs? It could well be that there are other, so far hidden, organizing centres in the medulla, the HCs simply a most obvious one to test; the myo-TEC (surprisingly large numbers!) may be the obvious candidate to look at.	Figure 5, where we show cell composition in both CMA and HC proximity. We do think that Figure 5d+f touch on this point: The medullary depth of a HC can be imagined as a vertical line in Figure 5d at the CMA depth of mTECIII (x-intercept = 0.64). This highlights a variety of cells including myo-TECs, Tregs, and naive B cells that are enriched at an equidistant position with respect to the CMA but less associated with HCs. We did consider looking at the distance of cells to myo-TEC, as suggested by the reviewer, but noted two fundamental problems. 1. Since there is no morphological landmark the resolution of the analysis would be limited to Visium spot spacing which is 100 μm, which is too low to be informative in this type of analysis. 2. We can only infer myo-TECs from cell deconvolution and would thus need to choose a certain threshold for labelling an individual Visium spot as “containing myo-TECs”, which cannot justify scientifically. For these reasons we consider this analysis out of scope since it would require single cell resolution, full transcriptome profiling such as visiumHD or StereoSeq to unbiasedly detect alternative centres or hubs in the medulla.
3.17	Again, please check nomenclature, and the presence of scale bars, information on age of thymi etc.	We have taken extra care to provide this information in the figure legends.
3.18	The analysis of canonical T lineage differentiation is a highlight of the manuscript; I think that panel D of EDF 11 belongs to main Fig. 6, to put the presentation into context without having to refer back to the additional presentations. This topographical analysis provides a very strong data set supporting previous sketchy	We thank the reviewers for the suggestions for restructuring Figure 6 and the associated Extended Data Figure. We have moved the relevant parts of Extended Data Figure 11 to the main Figure (Figure 6a).

	data/assumptions and is an important resource for further studies on lympho-stromal interactions, which underlie lineage commitment progression.	
3.19	One such aspect is explored here in the analysis of CD4/CD8 bifurcation (starting from positive selection), focusing on the relationship between location (as a function of migration) and differentiation state. I do think, however, that the data presentation in Fig. 6 is somewhat confusing. First, panel A may well be relocated to EDF, then, once one has gone through the entire figure, one realizes that the colour code in B is not matching that in the remaining panels. Moreover, it seems that panels E and C are the most illuminating, and should perhaps be presented in reverse order?	We have reworked Figure 6 to improve the overall presentation. In this context we have removed panel A and shifted panel D to the Extended Data and instead incorporated Extended Data Figures 11D and 12C into the main figure. The colour scheme has been reworked to improve clarity and match with the remaining CITE-seq data, which has now been moved to Supplementary Note 6. In addition, we have moved all pseudotime panels in Figure 6 into a single column to make it easier to compare relationships between them. Regarding the panel order, original/current panel 6c (pseudotemporal ordering of annotated stages) is the foundation for the subsequent panels including the spatio-temporal mapping in 6E (now 6e-f) and we therefore consider the current order of presentation the most logical.
3.20	The distinct differences of deduced migration patterns of CD4 and CD8 is a major finding, hence it would help to move panel C from EDF 12 to the main Fig. 6 to more easily illustrate the ligand/receptor relationships, and to accompany this presentation with a schematic highlighting the distinct responsiveness to CCL17/22.	Following the reviewer's suggestion we have moved original Extended Data Figure 12C to the main figure (now Figure 6h). We have colour-coded ligands and receptors in Figure 6g+h to illustrate their connection. Of note, we have added an additional panel (c) in the new Extended Data Figure 8, which showcases the cell types with high levels of transcription of the relevant chemokines.
3.21	I was also impressed by the seemingly perfect concordance between IBEX/Visium data sets, providing a showcase for the computational pipeline developed in this paper.	We agree that this concordance in mapping is quite astonishing, especially given the fact that the IBEX antibody panel was originally designed to detect various types of TECs but did not include many T maturation markers.

3.22	Discussion: This section more or less recapitulates the results section. I suggest the authors focus on the differences between fetal and postnatal tissue; perhaps, they can also speculate what happens during thymic involution, and chart a road map for a more focused analysis of “older” thymi and those that are found in patients with inborn errors of stromal and haematopoietic compartments.	We thank the reviewer for the suggestions and have rewritten the discussion to offer an outlook on how our technological advancements could be used in the future. We have included the reviewer’s suggestion of applying this to ageing or disease-related thymus samples.
Reviewer #4:		
Comments	Response	
Yayon et al present their work in mapping reference cell types atlases (using an all-integrated dataset) on spatial data. Multiple types of single cell and spatial datasets were used: scRNA, TCR-seq and CITE-seq for single cell data, and Visium, IBEX (for the most part) for spatial data. First, the authors present their approach to establishing a coordinate system to able to compare their results across different spatial platforms/modalities, while they use previously published approached for cell type calling or deconvolution. Next, they present the results, first on larger macrostructures in the thymus using their binned CMA regions (first on T cells and then on other cell populations), and then they move on to a higher resolution structure (HC) before finally back to mapping finer grained CITE-seq-defined immune T cells subsets across the inferred spatial regions in pediatric samples only.	We thank the reviewer for the careful review of the methods and detailed description of how to improve our work. We respond to their concerns below.	

There are two major highlights in the presented work: attempting to use a CCF for multi-tissue and multi-patient spatial analysis and integrating publicly available and newly generated single cell and spatial datasets from difficult to obtain fetal samples. However, there are two major concerns the reviewer would like to point out:	
4.1	Briefly, one could describe the CCF approach as follows:  1. manually annotate some components in an image (either HE or IF) based on choosing spots with highest marker expression 2. train and refine pixel classifier (basic random forest) to annotate the rest of the tissue section 3. calculate distances between the centroid of a region and the edge of another region or the edge of the tissue 4. bin these distances using user-based cut offs and add user-based labels to these bins marking regions of interest 5. repeat for every image in the dataset While the authors posit the benefit of having a CCF approach (lines 127-143), this is not really what they did - there is not a real common coordinate system here. They used annotated tissue areas to bin their data into categories of observations so that they could compare those between conditions ie. pediatric and fetal; or within a condition ie. pediatric alone across the binned We thank the reviewer for the critical feedback. However, the reviewer’s summary of the CMA construction led us to believe that there was a fundamental misunderstanding, which has probably led to some of the subsequent comments: Importantly, in the third point the reviewer describes a process according to which we selectively calculate distances based on measurements from the centroid of a region (e.g. medulla) to the edge of a different region (e.g. cortex) and then divide the distance from this centroid to the edge to derive the bins, which is incorrect. The reviewer appears to assume that an independent measurement is derived for the capsule and cortex. This would be prone to bias and not consistent enough to be used as a CCF. It would also require huge annotation effort as each medulla or cortical region would need to be defined as “medulla_1,2,n” in order to calculate its centroid. The actual approach we take for calculation of the base OrganAxis function and the CMA is very different: The CMA is a general model that is calculated using the same formula for all the spots in an image based on the annotation of three major landmarks: cortex, medulla, and capsule. The CMA combines minimal distances from all the landmarks to a given spot in a weighted, non-linear, normalised fashion. This allows the derivation of a positional score for each xy location, accounting for local and global

	annotations. As such, the authors cannot argue that they built a completely novel spatial processing and analysis pipeline with a thymus CCF.	environment and enables direct and consistent comparison across developmental stages and modalities while retaining spatial variance within regions. Next, while the bins were selected to better represent compartments of the thymus that are important for the biological processes taking place in this organ, they are drawn consistently across sections based on the same, specific CMA value cut-offs across all sections (Supplementary Table 8). Based on this comment, which has emphasised the need for a more detailed explanation on this matter, we now provide an extended demonstration on the axis construction in the Supplementary Note 2. In this note we elaborate on the mathematical principles behind the OrganAxis construction and the chosen parameters. We have also added simulations and visualisation of data for further clarification and show how to implement and adapt the approach (Supplementary Figures 3-4). Finally, we have generated a dedicated online tutorial (https://organ-axis-tutorial.readthedocs.io/en/latest/index.html#), which we will maintain and update. We would also like to clarify in response to the reviewer's first point, that we do not solely annotate thymic compartments based on marker expression. In our approach only the broad distinction of cortex and medulla and the boundary between them are originally defined automatically based on markers. The rest of the annotations and QC are based on expert guided manual curation to minimise error (See Supplementary Note 1).
4.2	There is no analysis of the power vs. spatial variance in their study. Given that their analysis was not performed using any spatial statistics of Bayesian inference methods, it is hard to put	We thank the reviewer for bringing up this important point. We have now conducted extensive additional analysis on the variance between samples, conditions and regions along the axis to

value into observations from a handful of patients. While the reviewer fully acknowledges these samples are hard to obtain, there needs to be a clear presentation of the statistical power of this study to the readers, while now the results reads as an absolute “spatial thymus atlas”. A suggestion could be to: 1. reanalyse a part of their data (eg. Figure 3 where they had both IBEX and Visium data) with a spatially aware approach to see if they can with confidence detect the differentially expressed genes between regions or conditions:2. report the spatial variance explained by this model alongside their results in Figure 2H as it is clear from results in 2H, that their own results show the important of having “a spatial component” (albeit their is only in binned regions so there is much more space for improvement).3. in case a) and b) yield good results, then a simple statement in the paper saying “these genes that we identify were confirmed with alternative spatially-aware approaches giving us confidence in our own approach” should suffice.	provide an accurate estimation and comparison between conditions through direct statistical tests. Following the reviewer’s suggestions, we have conducted an extension of our exploration of cytokine space reported in Figure 3 in three ways. (Of note, this figure was focused on comparing Visium data from fetal and paediatric thymi and, contrary to the reviewers comment, no IBEX was included here; no cytokines were measured in the IBEX panel): a) We conducted a two-way ANOVA to estimate the divergence or similarity in cytokine expression profiles between conditions (age) and on the axis bins (now shown in Figure 3 and Extended data Figure 4). Briefly, we pooled gene expression of spots per sample and bin to avoid artificially inflating our statistical power. We then derived the ANOVA main and interaction effects for the cytokines and Bonferroni-corrected for multiple comparisons. In Figure 3f we now report the interaction effect and the cosine similarity of the median values along the axis between fetal and paediatric datasets. These two quantitative measurements now give a much better representation of how each cytokine is expressed. We also provide all the results and main effect sizes in Supplementary Table 5. b) To better estimate the correlation of cytokines to the CMA along the entire range using continuous data, we constructed a spline generalised linear model. This approach uses the full CMA continuous space and the bin edges as the spline knots and draws confidence intervals and corrected p-values to the covariates (donor and age) to estimate how the axis is correlated to local segments for a specific gene. We have used this analysis to investigate the association of cytokines and their receptors within an age group. This analysis is presented in the
--	---

		Revision Figure 1 (at the end of this document). Of note: We did not use this approach to compare between age groups since it is not clear how to interpret a divergence only in a specific segment of the axis and we felt the ANOVA analysis is better suited to address that question. c) In addition, we also expanded the analysis in the original Figure 2H and now report the cumulative variance explained by the CMA for each Visium and IBEX sample independently. We show that there is no specific trend in variance explained by CMA depending on age, both within and between age groups, nor a specific sample that forms an outlier (Extended data Figure 3d). We think that the reviewer's comment for an alternative, spatially-aware approach comes from the assumption that the CMA bins could mix up discrete morphological annotations. This is not the case as we show in Supplementary Figure 5d. Specifically, discrete annotation separations into cortex and medulla are not compromised by the bins that we derived. Finally, we consider that building a Bayesian spatial model is outside the scope of this work and would require a separate set of assumptions and computational approaches far more complex than the approach we are offering here.
4.3	The reviewer is aware that there is not much difference between the binned IBEX vs. Visium data (for specific targets) and that the authors can argue they have cross-validated their results already. However, using a small targeted panel of antibodies also proves the point that these targets have already been validated and are not especially novel, while	We would like to clarify that the aim of this study was not to discover novel marker genes or cell types. Regardless, we did find new markers and previously uncharacterised cell types:  1. Our single cell analyses revealed that LMNA is expressed at higher levels in non-T cells compared to the T lineage and can therefore be considered a novel marker which we have added to our IBEX panel. In addition, due to its nuclear expression, it

	the authors went through lengths to create spatial RNA-seq data and integrated it with cell2location with scRNAseq (and CITE-seq) for novel marker discovery in order to make the thymus atlas.	facilitates better detection of stromal cells which are hard to capture by nuclear segmentations due to their “non-circular” morphology. 2. We have also developed a statistical approach to detect “specialisation genes” that are only expressed in a single cell type, an approach we use to directly infer a spatial location of the corresponding TEC sub-population in the medulla (Figure 5). We also discovered a novel mTECIII subtype marked by expression of the specialisation genes CXCL17, MUC4 and PSCA, which we termed mTECIII-muc.
--	--	---

Reviewer Reports on the First Revision:

Referees' comments:

Referee #1 (Remarks to the Author):

The authors have responded thoroughly to all my concerns

Referee #3 (Remarks to the Author):

I am very pleased with the revision of the paper. It is much better structured now, and I particularly like the current emphasis on the biological discoveries; the “technical” aspects are now well summarized in the supplementary material.

Discussion: I suggest that the authors briefly mention in this section that the apparent differential pace of CD4[>]CD8 development (as manifested here in the earlier appearance in the medulla of the former) may also be due to the inherent time difference in interpretation of TCR signalling-triggered lineage decision, a rather remarkable similarity to the analyses recently summarized for the mouse (PMID: 38271641).

Introduction, line 62: The term “seeded by” is misleading, when the thymic primordium is “composed of” the said stromal cell types, as so elegantly demonstrated 5 decades ago (PMID: 239088).

Referee #4 (Remarks to the Author):

The reviewer congratulate the authors on a much improved manuscript. All of the reviewer's technical comments about the method have been adequately addressed.

Referee #4 (Remarks on code availability):

Code is available, well documented and usable.